# 6-Methylsulfinylhexyl isothiocyanate activates carbonic anhydrase-dependent $HCO_3^-/H^+/Na^+/Ca^{2+}$ transport via SLC4As–NHE–NCX–PMCA axis in odontoblasts

Yoshiaki Furusawa[1,2], Maki Kimura[1], Isao Okunishi[3], Takehito Ouchi[1] (ID), Ryuya Kurashima[1], Tomoe Katou-Yamada[3], Hidetaka Kuroda[4] (ID), Makoto Sugita[5] (ID), Masahiro Furusawa[2] (ID) and Yoshiyuki Shibukawa[1] (ID)

[1]*Department of Physiology, Tokyo Dental College, Tokyo, Japan*
[2]*Department of Endodontics, Tokyo Dental College, Tokyo, Japan*
[3]*Kinjirushi Co., Ltd, Aichi, Japan*
[4]*Department of Dental Anesthesiology, Kanagawa Dental University, Kanagawa, Japan*
[5]*Department of Physiology and Oral Physiology, Hiroshima University, Hiroshima, Japan*

Handling Editors: Kim Barrett & Pawel Ferdek

The peer review history is available in the Supporting Information section of this article (https://doi.org/10.1113/JP287809#support-information-section).

Y. Furusawa and M. Kimura contributed equally to this study.

The Journal of Physiology

**Abstract figure legend** 6-Methylsulfinylhexyl isothiocyanate (6-MSITC, hexaraphane), a wasabi sulfinyl compound, activates intracellular carbonic anhydrase (CA)-mediated sequential $HCO_3^-$, $Na^+$, $H^+$ and $Ca^{2+}$ transport through an activation axis involving $HCO_3^-$-transporting solute carrier family 4 (SLC4As), the $Na^+$–$H^+$ exchanger (NHE), the $Na^+$–$Ca^{2+}$ exchanger (NCX) and plasma membrane $Ca^{2+}$-ATPase (PMCA). 6-MSITC exhibits a strong ability to induce reactionary dentin formation via CA-mediated SLC4A–NHE–NCX–PMCA coupling, making it a promising and potent candidate for novel dentin-regenerative therapy.

**Abstract** $Ca^{2+}$-permeable transient receptor potential channels, which play an important role in the developmental/reactionary dentinogenesis by odontoblasts, are sensitive to wasabi sulfinyls. We investigated the effects of the wasabi sulfinyls 6-methylsulfinylhexyl isothiocyanate (6-MSITC) and its eight derivatives on ion transport mechanisms which promote mineralisation by odontoblasts. 6-MSITC significantly increased the mineralisation efficiency in cultured odontoblasts, and we also observed a significant increase in medium pH. Inhibitors of carbonic anhydrase (CA) and plasma membrane $Ca^{2+}$-ATPase (PMCA) significantly reduced 6-MSITC-induced mineralisation. Odontoblasts expressed the $HCO_3^-$-transporting solute carrier family 4 (SLC4A) members SLC4A1, SLC4A2, SLC4A3, SLC4A4, SLC4A8 and SLC4A9, as well as CA I and CA II. 6-MSITC enhanced reactionary dentinogenesis beneath the cavities prepared on rat mandibular first molars. We recorded 6-MSITC-induced outward currents, which were suppressed by inhibitors of CA, $Na^+$–$H^+$ exchanger (NHE), and $Na^+$–$Ca^{2+}$ exchanger (NCX). These results indicated that 6-MSITC has a strong ability to form reactionary dentin by activating or upregulating intracellular CA and electrically neutral $HCO_3^-$/$H^+$ transport via SLC4As/NHE. Exchanging $Na^+$ with $H^+$ using NHE resulted in the reversal of the transmembrane $Na^+$ gradient. This activated the $Ca^{2+}$ influx mode of NCX, and the subsequent accumulation of intracellular $Ca^{2+}$ was then extruded by PMCA activity to produce reactionary dentin. Thus, 6-MSITC activates CA-mediated SLC4As–NHE–NCX–PMCA coupling and is useful in dentin regenerative medicine.

(Received 4 October 2024; accepted after revision 28 January 2026; first published online 20 February 2026)

**Corresponding author** M. Kimura and Y. Shibukawa: Department of Physiology, Tokyo Dental College, Tokyo 101-0061, Japan. Email: yshibuka@tdc.ac.jp; tsumuramaki@tdc.ac.jp

## Key points

- $Ca^{2+}$ signalling in odontoblasts plays an important role not only in developmental/reactionary dentinogenesis, but also in the generation of dentinal (tooth) pain.
- The wasabi sulfinyls including 6-methylsulfinylhexyl isothiocyanate (6-MSITC) promote mineralisation by odontoblasts and significantly increase pH in medium with cultured odontoblasts.
- We showed that 6-MSITC has a strong ability to form reactionary dentin through the upregulation and activation of intracellular carbonic anhydrase (CA) and electrically neutral $HCO_3^-$/$H^+$ transport by members of the $HCO_3^-$-transporting solute carrier family 4 (SLC4As) and $Na^+$-$H^+$ exchanger (NHE).
- $Na^+$ accumulation by NHE activity resulted in the reversal of the transmembrane $Na^+$ gradient. This activated the $Ca^{2+}$ influx mode of the $Na^+$–$Ca^{2+}$ exchanger (NCX), resulting in the accumulation of intracellular $Ca^{2+}$. It was then extruded using plasma membrane $Ca^{2+}$-ATPase (PMCA) to produce reactionary dentin.
- 6-MSITC activates CA-mediated SLC4As–NHE–NCX–PMCA coupling and is useful in dentin regenerative medicine.

## Introduction

During dentinogenesis, odontoblasts play an important role in dentin formation and mineralisation. Odontoblasts can detect mechanical, osmotic, pH-related and thermal stimuli and activate dentinogenesis (Goldberg & Smith, 2004; Tsumura et al., 2012, 2013), which forms 'reactionary dentin' following moderate dentin injury when multiple stimuli are applied to the dentin surface (Charadram et al., 2012; Goldberg & Smith, 2004; Kimura et al., 2016, 2021; Tsumura et al., 2013).

The stimuli applied to the dentin surface are converted into dentinal fluid movements, which activate mechano-sensitive ion channels, such as transient receptor potential (TRP) channel subtypes and Piezo1 channels in odontoblasts as sensory receptor cells, which induce increases in intracellular free $Ca^{2+}$ concentration ($[Ca^{2+}]_i$) through $Ca^{2+}$ influx (Kimura et al., 2018, 2021; Matsunaga et al., 2021; Ohyama et al., 2022; Sato et al., 2013, 2015, 2018; Shibukawa et al., 2015). $Ca^{2+}$ influx in odontoblasts triggers the release of ATP from pannexin channels and glutamate from the volume-sensitive outwardly rectifying anion channels as neuro-/intercellular-transmitters (Nishiyama et al., 2016; Ohyama et al., 2022; Sato et al., 2015, 2018; Shibukawa et al., 2015). The released ATP and glutamate act as sensory signal transduction sequences for dentinal pain by transmitting sensory signals to neurons via activation of the ionotropic ATP receptor subtype 3 ($P2X_3$) in the trigeminal neurons (odontoblast mechano-sensory/hydrodynamic receptor model) (Kimura et al., 2021; Matsunaga et al., 2021; Nishiyama et al., 2016; Ohyama et al., 2022; Sato et al., 2018; Shibukawa et al., 2015). Additionally, ATP, glutamate and ADP, which is hydrolysed from ATP, activate $P2X_{4/7}$ (Inoue et al., 2021), metabotropic glutamate receptors, and G-protein-coupled nucleotide receptors subtype 1 and 12 ($P2Y_{1/12}$) receptors in the surrounding odontoblasts and may promote dentinogenesis through a cluster of odontoblasts via odontoblast–odontoblast intercellular signal communication (Kimura et al., 2018, 2021; Nishiyama et al., 2016; Sato et al., 2015, 2018; Shibukawa et al., 2015). In addition to external stimuli-induced $Ca^{2+}$ entry, odontoblasts possess $Ca^{2+}$ release pathways to the intracellular space from intracellular $Ca^{2+}$ stores, which are mediated by inositol-1,4,5-trisphosphate receptors or ryanodine receptors in response to G-protein-coupled receptor activation or depolarisation, respectively (Kimura et al., 2018; Kojima et al., 2017; Shibukawa & Suzuki, 1997, 2003). Thus, $Ca^{2+}$ signalling in odontoblasts, which mediates dentin formation and/or sensory transduction, comprises two closely related components of the $Ca^{2+}$ mobilisation pathway: external stimuli-evoked $Ca^{2+}$ influx and intracellular $Ca^{2+}$ release from $Ca^{2+}$ stores. Through intracellular $Ca^{2+}$ signalling, $Ca^{2+}$ is extruded to the dentin mineralising front via the $Na^+$–$Ca^{2+}$ exchanger (NCX; $Na^+$–$Ca^{2+}$ antiporter) subtypes 1 and 3 (Kimura et al., 2016; Lundgren & Linde, 1988; Lundquist et al., 2000; Tsumura et al., 2010, 2012) and/or plasma membrane $Ca^{2+}$-ATPase (PMCA) (Kimura et al., 2021; Linde & Lundgren, 2003) in odontoblasts, which is involved not only in maintaining $Ca^{2+}$ homeostasis but also in dentinogenesis.

Among the various TRP channels in odontoblasts, TRP ankyrin subfamily member 1 (TRPA1) channels are sensitive to a high pH extracellular environment, showing external $Ca^{2+}$ dependency, which induces the $Ca^{2+}$ influx pathway (Kimura et al., 2016). TRPA1 channel activation by high-pH stimuli, such as dental materials of mineral trioxide aggregate (MTA) and $Ca(OH)_2$, elicited dentin formation. TRPA1 channels are also involved in the formation of physiological dentin (Kimura et al., 2016, 2021). Furthermore, physiological and high pH-induced dentinogenesis is mediated by PMCA and NCX (Kimura et al., 2016, 2021). It has been reported that TRPA1 channels are activated not only by high pH stimulation but also a variety of isothiocyanate compounds, including allyl-, benzyl-, phenylethyl-, isopropyl- and methyl-isothiocyanate, which constitute the main pungent ingredients in wasabi, yellow mustard, brussels sprouts, nasturtium seeds and capers (Jordt et al., 2004). Among these, allyl iso-thiocyanate (AITC), 6-methylsulfinylhexyl isothiocyanate (6-MSITC; hexaraphane) and 6-methylthiohexyl iso-thiocyanate (6-MTITC) are the main compounds of wasabi (*Eutrema japonicum*) (Bandell et al., 2004; Jordt et al., 2004; Uchida et al., 2012) that activate TRPA1 channels. However, their ability to promote mineralisation

**Yoshiaki Furusawa** is an endodontist at the Hospital of Tokyo Dental College. Under the supervision of Professor Yoshiyuki Shibukawa in the Department of Physiology at Tokyo Dental College, he has investigated the effects of wasabi sulfinyl compounds on ionic transport and currently focuses on their potential in human dentin regeneration, aiming to develop novel dentin-regenerative therapies. **Maki Kimura** is a pharmacist and researcher in the laboratory of Professor Yoshiyuki Shibukawa in the Department of Physiology at Tokyo Dental College. Her research focuses on the coupling between mechanosensitive channel activation and neurotransmitter release in mechano-sensory transduction underlying dentinal sensitivity, contributing to the development of innovative dentin-regenerative therapies.

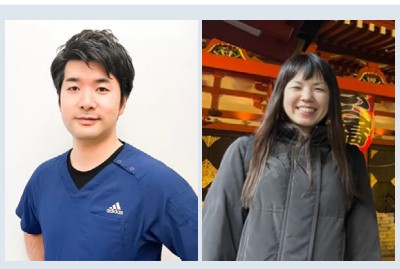

via TRP channel activation remains unclear. Thus, we hypothesised that the wasabi compounds may promote or modify dentinogenesis, with or without TRPA1 channel activity.

Therefore, this study aimed to elucidate whether isothiocyanates and their derivatives can promote dentin formation and to determine their chemical structure(s) for mineralisation. We evaluated 6-MSITC, the main compound from wasabi, as a novel drug to induce tertiary dentin formation and examined the detailed 6-MSITC-induced anion and $Ca^{2+}$ transport pathways underlying its dentin formation and mineralisation abilities *in vitro* and *in vivo*.

## Methods

### Ethical approval

All animals were treated in accordance with the Guiding Principles for the Care and Use of Animals in the Field of Physiological Sciences, approved by the Council of the Physiological Society of Japan and the American Physiological Society. All animal experiments were conducted in accordance with the Animal Research: Reporting of *In Vivo* Experiments guidelines, the guidelines established by the U.S. National Institutes of Health regarding the care and use of animals for experimental procedures, and the UK Animals (Scientific Procedures) Act 1986. This study was approved by the Ethics Committee for Animal Experimentation of Tokyo Dental College (Experimental Animal Plan approval numbers: 200301, 210301 and 220301). The animals used in this study (Wistar rats: Slc:Wistar; $n = 7$; both sexes; 12–13 weeks old; weight 190–210 g; Sankyo Lab Service Co. Inc., Tokyo, Japan) were housed in transparent cages with wood chips under specific pathogen-free conditions and maintained on a light–dark cycle (12:12) in a temperature- and humidity-controlled room (21–23°C and 40–60% relative humidity, respectively). Food pellets and water were provided *ad libitum*. All efforts were made to minimise animal suffering. For the animal experiments, cavities were prepared in the dentin without pulpal exposure on the occlusal surfaces of mandibular first molars in Wistar rats. The cavities were tightly sealed with a dental adhesive resin cement (see '*In vivo* experiments' section) to completely prevent postoperative pain. To prevent abrasion and dislodgement of the restorations due to occlusal contact, the cusps of the maxillary teeth were also drilled. The animals were monitored daily for oral and dental conditions, such as dislodgement of the restorations, and all animals used in this study remained healthy and showed no complications throughout the experimental period.

## Culture of human odontoblasts

A human odontoblast cell (HOB) line was obtained from a healthy third molar and immortalised by transfection with the human telomerase transcriptase gene (Ichikawa et al., 2012; Kimura et al., 2016; Kitagawa et al., 2007; Kojima et al., 2017; Matsunaga et al., 2021). Cells were provided by Dr Masae Kitagawa of Hiroshima University, Japan, and Dr Takashi Muramatsu of Tokyo Dental College, Japan. A previous study reported that HOB cells showed mRNA expression of dentin sialophosphoprotein (DSPP), type 1 collagen, alkaline phosphatase and bone sialoprotein. They are also immunopositive for DSPP, nestin and dentin matrix protein 1 and exhibit nodule formation by alizarin red staining in the mineralising medium (Kitagawa et al., 2007; Matsunaga et al., 2021). Therefore, this cell line exhibited odontoblastic properties. These cells were used for the experiments after approximately 40 passages. HOB cells were cultured in basal medium (pH 7.4) alpha-minimum essential medium containing 10% fetal bovine serum (Thermo Fisher Scientific, Waltham, MA, USA), 100 U/mL penicillin-streptomycin (Thermo Fisher Scientific), and amphotericin B (Sigma-Aldrich, St. Louis, MO, USA) at 37°C in a 5% $CO_2$ incubator for 24 h.

## Solutions and reagents

Dorzolamide (DZ) hydrochloride, a CA inhibitor, was purchased from Sigma-Aldrich. The concentration of DZ was determined according to that described in previous studies (Adijanto et al., 2009; Boyne et al., 2022; Dong et al., 2016). KB-R7943, an NCX inhibitor, was obtained from Tocris Biosciences (Bristol, UK) and dissolved in dimethyl sulfoxide (DMSO; FUJIFILM Wako Pure Chemical Co., Osaka, Japan) at a concentration of 10 mM as a stock solution. Caloxin 1b1, a non-specific plasma membrane $Ca^{2+}$-ATPase inhibitor, was obtained from Karebay Biochem, Inc. (Monmouth Junction, NJ, USA). A stock solution of caloxin 1b1 was prepared using 10% ethanol. The stock solution was diluted to the appropriate concentration using culture medium before use. The concentration of caloxin 1b1 was determined as described previously (Chen et al., 2014; Groten et al., 2016; Kimura et al., 2021; Pande et al., 2006, 2008). Gramicidin was obtained from Sigma-Aldrich and was first dissolved in methanol to a concentration of 10 mg/mL and then diluted in the pipette solution to a final concentration of 200 µg/mL immediately before use. 4,4′-Diisothiocyanatostilbene-2,2′-stilbenedisulfonic acid (DIDS), a non-specific anion transporter inhibitor, was obtained from Sigma-Aldrich and dissolved in DMSO to a concentration of 10 mM for the stock solution. Calcein was obtained from Dojindo Laboratories (Kumamoto, Japan) and dissolved in phosphate-buffered saline (PBS; Thermo Fisher Scientific) containing 20 mg/mL $NaHCO_3$

(pH 7.4) to a final concentration of 10 mg/mL. The $Na^+$–$H^+$ exchanger inhibitor 5-(*N*,*N*-dimethyl) amiloride hydrochloride (DMA) was obtained from Sigma-Aldrich and dissolved in methanol to a concentration of 6 mg/mL for the stock solution.

## Wasabi sulfinyls and their similar compounds

We underscored on 6-MSITC and eight similar compounds of 6-MSITC (Fig. 1). In this study, nine types of compounds used are the following: 6-MSITC (hexaraphane; Kinjirushi Co., Ltd, Aichi, Japan), AITC (FUJIFILM Wako Pure Chemical Co), 4-methylsulfinylhexyl isothiocyanate (4-MSITC; Abcam, Cambridge, UK), 8-methylsulfinylhexyl isothiocyanate (8-MSITC; Abcam), 6-methylthiohexyl isothiocyanate (6-MTITC; Kinjirushi Co., Ltd), 6-methylsulfonylhexyl isothiocyanate (6-MSFITC; Kinjirushi Co., Ltd), *n*-hexyl isothiocyanate (*n*-hexyl ITC; Sigma-Aldrich), phenethyl isothiocyanate (PEITC; Kinjirushi Co., Ltd), and 4-(methylsulfinyl)-1-butylamine (4-MS amine; Santa Cruz Biotechnology, Dallas, TX, USA). A summary of the chemical structure (Fig. 1*A*) and chemical modifications of 6-MSITC (Fig. 1*B*) are shown in Fig. 1. 6-MSITC is composed of a six-carbon chain bonded to isothiocyanate and methylsulfinyl groups. In 4-MSITC and 8-MSITC, the number of C atoms in the chain was four and eight, respectively, compared to 6-MSITC. The oxygen was removed from 6-MTITC using 6-MSITC. In 6-MSFITC, the methylsulfinyl group of 6-MSITC was replaced with a methylsulfonyl group. The methylsulfinyl group of 6-MSITC was replaced with H for *n*-hexyl ITC. PEITC and AITC have a 2C chain with isothiocyanate; however, the methylsulfinyl group was replaced with a phenyl group for PEITC or with $CH_2$- for AITC, unlike 6-MSITC. The 4-MS amine has a 4C chain without isothiocyanate but with a methylsulfinyl group (see Fig. 1).

## Mineralisation assay

HOB cells were grown to full confluency in the basal medium and then grown in a mineralisation medium containing 10 mM $\beta$-glycerophosphate and 100 μg/ml of ascorbic acid (final concentration) in the basal medium at 37°C with 5% $CO_2$ with or without wasabi sulfinyls for 28 days using 12-, 24- and 48-well plates (Sumitomo Bakelite Co., Ltd, Tokyo, Japan). To detect calcium deposits, the cells were fixed with 4% paraformaldehyde phosphate buffer solution (PFA; FUJIFILM Wako Pure Chemical Co.) and subjected to alizarin red and von Kossa staining. Following alizarin red and von Kossa staining, images of the stained samples were captured using a digital camera (Sony Corporation, Tokyo, Japan), and mineralisation efficiencies were measured using ImageJ

software (NIH, Bethesda, MD, USA). The images were converted to 8-bit reversed grayscale, and the regions of interest (ROIs) were determined for each well to measure the mean luminance intensities of the total number of pixels (*I*) in the ROI. Mineralisation efficiencies were normalised and represented as $I/I_0$ units, and the intensities (*I*) of alizarin red and von Kossa staining were normalised to the mean intensity values of the areas without cells ($I_0$) (Kimura et al., 2021; Matsunaga et al., 2021).

## Crystal violet staining

HOB cells were seeded on 24-well adherent plastic plates (Sumitomo Bakelite Co., Ltd) and cultured in basal medium until they reached confluence. Once confluent, increasing concentrations of 6-MSITC (0 μM as the control, 10, 50 and 500 μM) were applied. After 72 h of incubation, the cells were fixed with 4% PFA and stained with 0.5% crystal violet (dissolved in methanol, 038-04862, FUJIFILM Wako Pure Chemical Co.) for 20 min at room temperature. After staining, the crystal violet solution was removed, and cells were gently washed three times with distilled water. Images were captured using a microscope (BZ-X710, Keyence, Osaka, Japan). Stained cells were quantified by manual counting from the images and presented as the number of cells per superficial area of 9,846,995 μm$^2$, denoted as S (cell number/S).

## Immunofluorescence

HOB cells were cultured in 8-well glass chambers (AGC Techno Glass Co., Ltd, Shizuoka, Japan) for 24 h. The cells were fixed with 4% PFA (FUJIFILM Wako Pure Chemical Co.), permeabilised (0.1% Triton X-100 treatment), and washed three times with PBS. After 10 min of incubation with a blocking buffer (Nacalai Tesque, Kyoto, Japan) at room temperature, the primary antibody was added and allowed to react for 7–8 h at 4°C or 1 h at 37°C. The secondary antibody was added and incubated for 1 h at room temperature in the dark. For the negative control, after blocking, the cells were incubated with primary antibodies, followed by the addition of secondary antibodies. The stained samples were mounted in a mounting medium with 4,6-diamidino-2-phenylindole (DAPI) (Abcam). Immunostained samples were analysed under a fluorescence microscope (BZ-X710; Keyence). Primary and secondary antibodies used in this study are listed in Table 1.

## Whole-cell patch-clamp recordings

Krebs solution containing 136 mM NaCl, 5 mM KCl, 2.5 mM $CaCl_2$, 0.5 mM $MgCl_2$, 10 mM HEPES, 10 mM

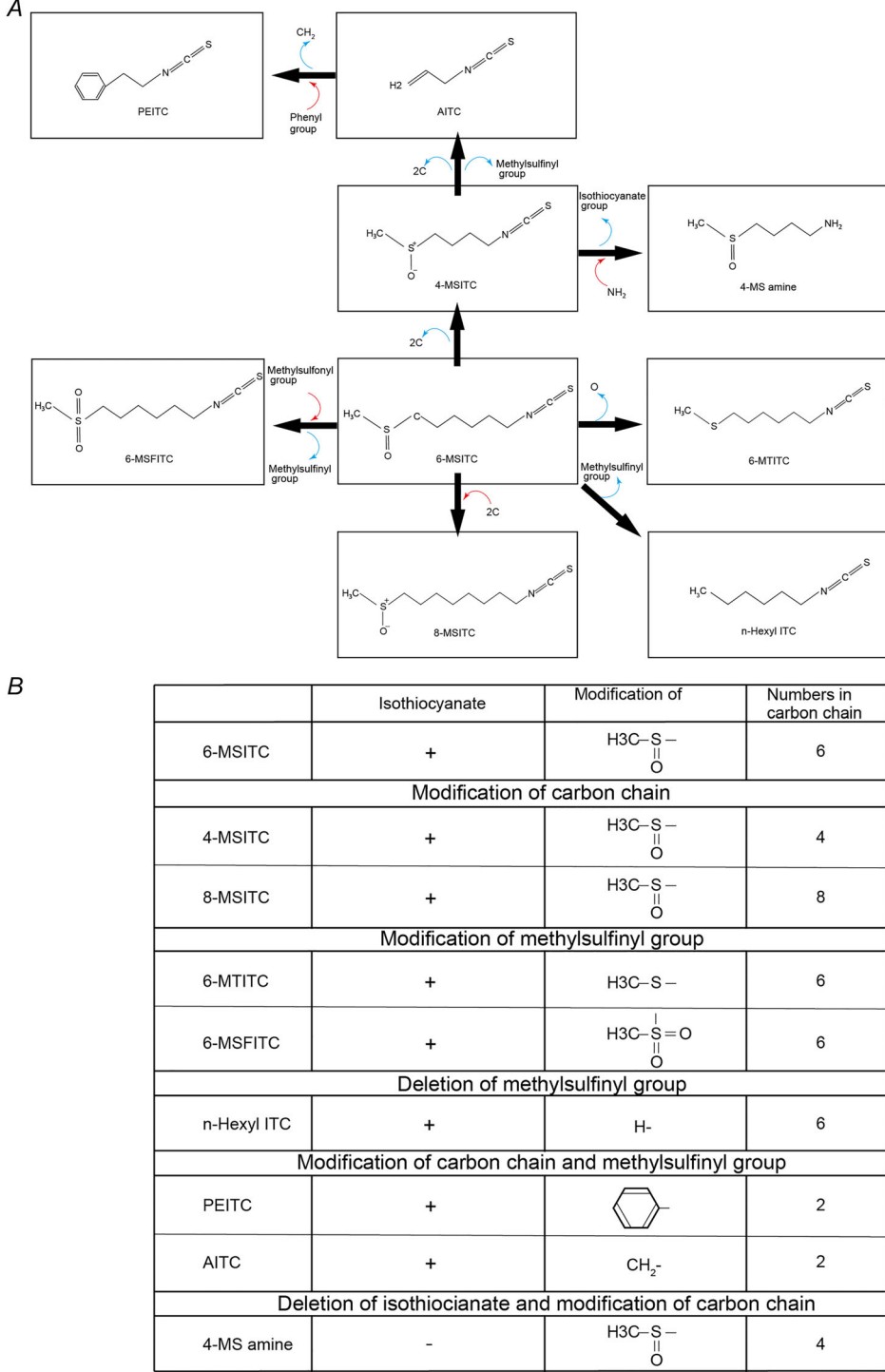

**Figure 1. Chemical structures and modifications of 6-MSITC and its derivatives used in this study**

*A*, schematic chemical structures and modifications of 6-MSITC (hexaraphane) and its eight other derivative compounds. *B*, summary of the modifications of chemicals used in this study. 6-MSITC is composed of a 6-carbon (6C) chain with isothiocyanate and methylsulfinyl groups. 4-MSITC and 8-MSITC have a modified number of C atoms in the carbon chain from 6-MSITC, i.e. 4C and 8C, respectively. For 6-MTITC oxygen is removed from 6-MSITC. For 6-MSFITC, the methylsulfinyl group of 6-MSITC is replaced with a methylsulfonyl group. The methylsulfinyl group is replaced with H for *n*-hexyl ITC. PEITC and AITC have 2C in the chain with isothiocyanate; however, the methylsulfinyl group is replaced with phenyl group for PEITC or with $CH_2$ for AITC. The 4-MS amine has 4C in the chain without isothiocyanate but with a methylsulfinyl group.

**Table 1. Primary and secondary antibodies used in this study**

| Primary antibody | Company | Host | Clonality | Dilution |
|---|---|---|---|---|
| SLC4A1 | Sigma-Aldrich, HPA063911 | Rabbit | Polyclonal | 1:100 |
| SLC4A2 | Sigma-Aldrich, HPA019339 | Rabbit | Polyclonal | 1:100 |
| SLC4A3 | Sigma-Aldrich, HPA063498 | Rabbit | Polyclonal | 1:100 |
| SLC4A4 | Santa Cruz Biotechnology, sc-515 543 | Mouse | Monoclonal | 1:100 |
| SLC4A8 | Santa Cruz Biotechnology, sc-100 672 | Mouse | Monoclonal | 1:100 |
| SLC4A9 | Sigma-Aldrich, HPA051307 | Rabbit | Polyclonal | 1:100 |
| CA I | Santa Cruz Biotechnology, sc-393 490 | Mouse | Monoclonal | 1:100 |
| CA II | Santa Cruz Biotechnology, sc-48 351 | Mouse | Monoclonal | 1:100 |
| Secondary antibody | | | | |
| Donkey anti-mouse/rabbit IgG (Alexafluor® 568/488 conjugated) | Life Technologies Japan Ltd, A10037/A21206 | Donkey | Polyclonal | 1:500 |

Detailed information on the company, host species, clonality and dilution of each primary antibody, as well as each secondary antibody. Life Technologies Japan Ltd, Tokyo, Japan; Santa Cruz Biotechnology, Dallas, TX, USA; Sigma-Aldrich, St Louis, MO, USA. CA, carbonic anhydrase; SLC4A, solute carrier family 4.

glucose and 12 mM $NaHCO_3$ (pH 7.4) was used as the standard extracellular solution (ECS). The intracellular solution (ICS) was composed of 140 mM KCl, 10 mM NaCl and 10 mM HEPES (pH 7.2/Tris) and was used in both conventional and gramicidin-perforated modes for whole-cell patch-clamp recordings. Krebs–Henseleit Ringer (KHR) solution containing 103 mM NaCl, 4.7 mM KCl, 2.56 mM $CaCl_2$, 1.13 mM $MgCl_2$, 25 mM $NaHCO_3$, 1.15 mM $NaH_2PO_4$, 2.8 mM glucose, 4.9 mM sodium pyruvate, 2.7 mM sodium fumarate and 4.9 mM sodium glutamate (buffered with 12.5 mM HEPES at pH 7.4) was used for ECS for both the conventional and gramicidin-perforated patch-clamp recordings. The solution was thoroughly gassed with 95% $O_2$ and 5% $CO_2$ prior to each experiment.

Whole-cell voltage-clamp recordings of the conventional and perforated modes were performed. Patch pipettes with a resistance of 5–10 MΩ were made from glass capillaries (DMZ Universal Puller, Zeitz-Instruments, Martinsried, Germany) and filled with ICS with or without gramicidin. For the perforated mode of patch-clamp recording, the pipette tip was filled with gramicidin-free ICS by brief immersion before backfilling the pipette with gramicidin-containing ICS. Electrical access to the interior of the cell was indicated by a gradual increase in the capacitive transients and a decrease in the series resistance, which occurred within 5–15 min after the formation of the initial gigaohm seal. Gramicidin-perforated patch-clamp recording was initiated after stabilisation of the capacitive currents (∼15–25 min after cell attachment; Ebihara et al., 1995; Sugita et al., 2000, 2004). We applied a voltage ramp protocol at a holding potential of −70 mV, with voltages ranging from −100 mV to +100 mV. Whole-cell currents were measured using a patch-clamp amplifier (MultiClamp 700B; Molecular Devices, San Jose, CA, USA), which compensated for the cell capacitance and series resistance. The currents were monitored and stored in an online computer after the analog signals were digitised through an analog-to-digital converter (Digidata 1440A; Molecular devices) for further analysis using the acquisition/analysis software program (pCLAMP; Molecular devices). Data were analysed using ORIGIN technical graphics/analysis software (OriginLab Corporation, Northampton, MA, USA). All experiments were performed at room temperature (26°C). We expressed the current amplitude in terms of the current density (pA/pF).

### *In vivo* experiments

Cavities were prepared inside the dentin without pulpal exposure at the occlusal surface on both sides of the mandibular first molars in Wistar rats of both sexes (12–13 weeks old). MedGel (Nitta Gelatin Inc., Osaka, Japan) containing 500 μM 6-MSITC or saline (as the control, Otsuka Pharmaceutical Co., Ltd, Tokyo, Japan) was put in place, before tightly sealing with dental adhesive resin cement (Super-Bond; Sun Medical Co., Ltd, Shiga Japan). Before cavity preparation, the rats were anaesthetised with a mixture of three agents prepared in saline: midazolam (8%; Sandoz KK, Yamagata, Japan), medetomidine hydrochloride (7.5%; Domitor; Nippon Zenyaku Kogyo Co., Ltd, Fukushima, Japan), and butorphanol tartrate (10%; Vetraphale; Meiji Co., Ltd, Tokyo, Japan). The mixture was administered intraperitoneally at 5.0 mL/kg. We prepared 1-mm-deep cavities using a round-type 0.5-mm-diameter dental

**Table 2. Primer sequences for real-time RT-PCR used in this study**

| Name | 5′-Sequence-3 | | GenBank number |
|---|---|---|---|
| GAPDH | Forward | GCACCGTCAAGGCTGAGAAC | NM_0 02046.7 |
| | Reverse | TGGTGAAGACGCCAGTGGA | |
| SLC4A1 | Forward | CTTGTGCTCATGGCCGGTA | NM_000342.4 |
| | Reverse | CAGGACCATGATCAGGATGGA | |
| SLC4A2 | Forward | TCATCTTCATGGAGACACAGATCAC | NM_0 011 99692.3 |
| | Reverse | ACGATGAGCAGCAGGTCCAG | |
| SLC4A3 | Forward | GTCCCAGCGTTTGTTGCTCA | NM_0 013 26559.2 |
| | Reverse | ATGCATCCGCCACGTCTTC | |
| SLC4A4 | Forward | AGCCCAGCCATGACCCATAG | NM_0 03759.4 |
| | Reverse | ACGTTGGAAGCTTCTGCATCAC | |
| SLC4A8 | Forward | CAGCCAGCTGGACCACCTTA | NM_0 012 58402.2 |
| | Reverse | GTCAGGTTAGCCCAGTGGACTTC | |
| SLC4A9 | Forward | CATCCTTACAGGAGCCTCCA | NM_0 012 58426.2 |
| | Reverse | TGGCATCAACAACAGCTTCAC | |
| CA1 | Forward | TTCACGTAGCTCACTGGAATTCTG | NM_0 012 91968.2 |
| | Reverse | AGCTTTGGGTTGGCCTCAC | |
| CA2 | Forward | GATGGACTGGCCGTTCTAGGTA | NM_0 012 93675.2 |
| | Reverse | TGGAATCCAGCACATCAACA | |

The conditions for real-time RT-PCR were as follows: 1 cycle at 42°C for 5 min, followed by 1 cycle at 95°C for 10 s, 40 cycles at 95°C for 5 s, and then 60°C for 30 s. The conditions for the dissociation curve analysis were as follows: 1 cycle at 95°C for 15 s, 1 cycle at 60°C for 30 s, and 1 cycle at 95°C for 15 s. CA, carbonic anhydrase; GAPDH, glyceraldehyde-3-phosphate dehydrogenase; SLC4A, solute carrier family 4.

diamond bur (Mokuda Dental Co., Ltd, Kobe, Japan). To prevent abrasion on both sides, the cusps of the maxillary teeth were drilled to prevent them from biting together with the lower teeth. To test whether the application of 6-MSITC promoted reactionary dentin formation, we attempted to detect newly calcified dentin using calcein labelling (Li et al., 2018; Zhao et al., 2021). For calcein labelling, rats were injected intraperitoneally with 10 mg/kg calcein 24 h before cavity preparation and 24 h before sacrifice. Two weeks after cavity preparation, the rats were sacrificed using isoflurane (3%). Sections, including the mandible, were prepared, and mineralisation/dentin formation was evaluated using haematoxylin–eosin (H-E) staining, Masson's trichrome staining, and calcein labelling. For H-E and Masson's trichrome staining, samples were decalcified using 10% EDTA.2Na solution (Muto Pure Chemicals Co., Ltd, Tokyo, Japan) for 2 months at 4°C, and paraffin sections were prepared (4 μm-thick). For calcein labelling, samples were fixed with 4% PFA (FUJIFILM Wako Pure Chemical Co.) at 4°C and subsequently immersed in 30% sucrose solution for 6 h for cryoprotection. Cryosections (6-μm-thick) were prepared using Kawamoto's film method (Kawamoto & Shimizu, 2000). Staining was observed and analysed as 8-bit TIFF images using a fluorescence microscope (BZ-X710; Keyence). The application of 6-MSITC was performed using a single-blind method.

## Real-time reverse-transcription polymerase chain reaction

Total RNA from human odontoblasts (HOB cells) was extracted using a modified acid guanidinium–phenol chloroform method. The One-Step SYBR Primescript RT-PCR Kit with Thermal Cycler Dice (TaKaRa-Bio, Shiga, Japan) was used for semi-quantitative real-time RT-PCR to conduct reverse-transcription (RT), complementary DNA amplification, and polymerase chain reaction (PCR). Primer sets and PCR conditions are shown in Table 2. The comparative threshold ($2^{-\Delta\Delta C_t}$) method ($C_t$ as cycle threshold) was used for the quantification of the real-time RT-PCR results. Relative mRNA detection level of interest was normalised against that of glyceraldehyde-3-phosphate dehydrogenase (GAPDH). The average $C_t$ value for each primer set was measured, and the change in the $C_t$ was then calculated as the difference between the average $C_t$ for the target gene and GAPDH, which showed the control for the total baseline RNA quantity. The $\Delta\Delta C_t$ was calculated to evaluate the fold change in mRNA levels relative to GAPDH mRNA levels.

## Statistics and offline analysis

In the Results section, data are expressed and/or described as the means ± standard deviation (SD)

of $n$ observations, where $n$ represents the number of independent experiments or cells tested. Data are plotted in figures using box and whisker plots, with all points showing the lower inner fence, 25% percentile, median, 75% percentile, and upper inner fence. Note that all examined data points are plotted throughout the figure. The Shapiro–Wilk test or Kolmogorov–Smirnov test was used to test for normality. The Kruskal–Wallis test with Dunn's *post hoc* test, Friedman's test with Dunn's *post hoc* test, or Wilcoxon's test was used to determine non-parametric statistical significance. Parametric statistical significance was determined using ordinary one-way analysis of variance (ANOVA) with Tukey's *post hoc* test, RM one-way ANOVA with Tukey's *post hoc* test, or Student's unpaired *t*-test. Statistical significance was set at $P < 0.05$. The sample size was estimated using data obtained from our pilot study. The difference of mineralising efficiencies represented as $I/I_0$ units (see above) between control (without application of 6-MSITC to the cells) and tested (with application of 40 μM 6-MSITC to the cells) group was 0.50, and the SD was set at 0.01. Four samples were required to detect a difference in the experiment with a type I error of 0.05 and a power of 0.95 for a two-tailed unpaired *t*-test. A total of 4–40 experiments were included to account for the possibility of withdrawal from the study. Statistical analyses were performed using GraphPad Prism 8 software (GraphPad Software, La Jolla, CA, USA).

# Results

## 6-MSITC promoted mineralisation by odontoblasts

We examined the effects of 6-MSITC and similar compounds of 6-MSITC on mineralisation efficiency using alizarin red and von Kossa staining (Fig. 2). In both alizarin red (Fig. 2*A* and *C*) and von Kossa (Fig. 2*B* and *D*) staining, 6-MSITC significantly increased mineralisation efficiency at concentrations of 40 μM and 100 μM, while at concentrations ranging from 4 μM to 20 μM, 6-MSITC decreased mineralisation efficiency.

## Quantification of cell numbers following 6-MSITC treatment

To monitor cell numbers during the application of 6-MSITC, cell counts were performed using crystal violet staining (Fig. 2). Stained cells were quantified by manual counting and presented as the number of cells per superficial area of 9,846,995 μm$^2$, denoted as $S$ (cell number/$S$). Treatment of human odontoblasts with 6-MSITC resulted in a significant decrease in cell numbers at concentrations of 10 μM and 50 μM (4475.33 ± 341.35 cell number/$S$, $n = 6$, $P < 0.0001$ at 10 μM; 1803.33 ± 211.50 cell number/$S$,

$n = 6$, $P < 0.0001$ at 50 μM; Fig. 2*E2*, *E3*, and *F*). However, cell numbers recovered at a concentration of 500 μM 6-MSITC (3143.83 ± 175.89 cell number/$S$ $n = 6$, $P < 0.0001$ at 500 μM; Fig. 2*E4* and 2*F*), compared with the control condition (no 6-MSITC application: 6502.33 ± 404.00 cell number/$S$, $n = 6$; Fig. 2*E1* and *F*).

## AITC, a TRPA1 channel agonist, inhibited mineralisation by odontoblasts

AITC, a TRPA1 channel agonist, significantly decreased mineralisation efficiency of alizarin red staining (Fig. 3*A* and *B*), showing a half-maximal inhibitory concentration (IC$_{50}$) of 0.06 μM in concentrations ranging from 0.1 μM to 100 μM.

## Mineralisation-promoting abilities of 6-MSITC and its similar compound

We compared the effects of 6-MSITC, 4-MSITC, 8-MSITC, 6-MTITC, 6-MSFITC, *n*-hexyl ITC, PEITC and 4-MS amines on mineralisation efficiency induced by human odontoblasts. At a concentration of 50 μM, we observed significant increases in the mineralisation efficiency of both alizarin red (Fig. 4*A*) and von Kossa staining (Fig. 4*B*) by 6-MSITC (2.41 ± 0.64 $I/I_0$ unit, $n = 6$, $P = 0.000246$ with alizarin red and 1.35 ± 0.09 $I/I_0$ unit, $n = 6$, $P = 0.000336$ with von Kossa staining) and 6-MSFITC (2.02 ± 0.214 $I/I_0$ unit, $n = 6$, $P = 0.000449$ with alizarin red and 1.28 ± 0.101 $I/I_0$ unit, $n = 6$, $P = 0.000336$ with von Kossa staining), compared with that observed in the absence of 6-MSITC and 6-MSFITC (as control; 1.28 ± 0.25 $I/I_0$ unit, $n = 40$, with alizarin red and 1.07 ± 0.05 $I/I_0$ unit, $n = 40$, with von Kossa staining).

## 6-MSITC increased pH of odontoblast culture medium

Administration of 6-MSITC to the odontoblast culture medium resulted in a significant (at 6-MSITC concentration of 40 μM and 400 μM) and concentration-dependent increase in the pH of the medium (Fig. 5), showing a half-maximal effective concentration (EC$_{50}$) of 29.271 μM in concentrations ranging from 0.4 μM to 400 μM.

## Intracellular HCO$_3^-$ production and plasma membrane Ca$^{2+}$ extrusion participates in 6-MSITC-induced mineralisation by odontoblasts

DZ, a CA inhibitor, significantly and concentration-dependently decreased the 6-MSITC-induced mineralisation efficiency in odontoblasts, as determined by alizarin red staining

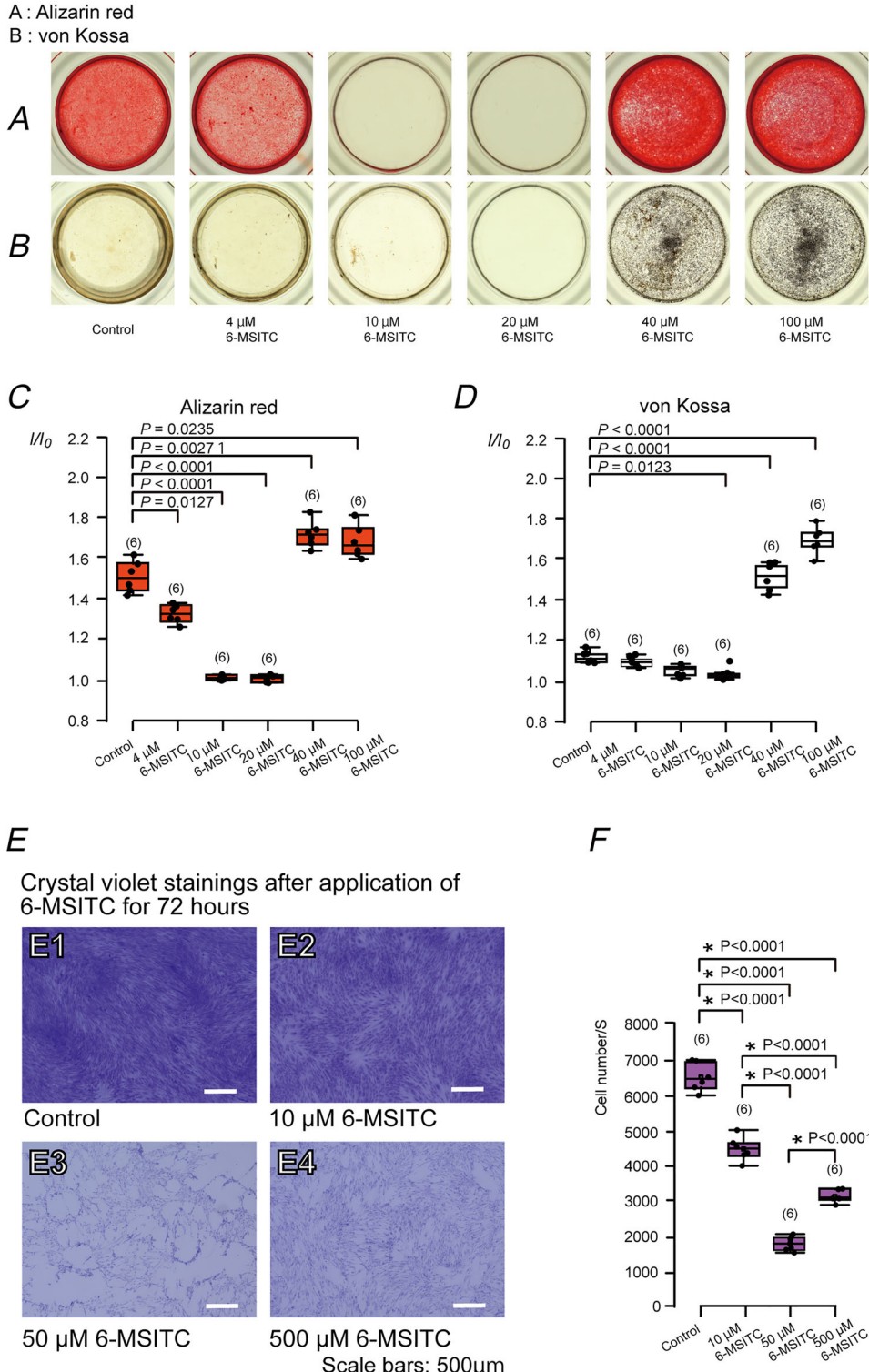

**Figure 2. Concentration-dependent effects of 6-MSITC on mineralisation and cell proliferation**
*A* and *B*, we cultured human odontoblast cells for 28 days in a mineralisation medium without (control) or with 4, 10, 20, 40 or 100 µM 6-MSITC. Mineralisation levels without (control) or with 4, 10, 20, 40 or 100 µM 6-MSITC were evaluated using alizarin red (*A*) and von Kossa staining (*B*). *C* and *D*, each box and whisker represent lower inner fence, 25% percentile, median, 75% percentile, and upper inner fence for mineralisation levels (as $I/I_0$ values) assessed using alizarin red (*C*) and von Kossa staining (*D*). *E1–E4*, human odontoblast cells were cultured for 72 h in the presence of 0 µM (control), 10 µM, 50 µM or 500 µM 6-MSITC after reaching confluence. After crystal

(1.45 ± 0.04 $I/I_0$ unit by 0.5 mM DZ; 1.35 ± 0.07 $I/I_0$ unit, $P = 0.000283$ by 1 mM DZ; 1.12 ± 0.04 $I/I_0$ unit, $P < 0.0001$ by 10 mM DZ; $n = 7$ each; Fig. 6*A* and *C*) and von Kossa staining (1.45 ± 0.05 $I/I_0$ unit by 0.5 mM DZ; 1.34 ± 0.06 $I/I_0$ unit, $P = 0.00161$ by 1 mM DZ; 1.10 ± 0.05 $I/I_0$ unit, $P < 0.0001$ by 10 mM DZ; $n = 7$ each; Fig. 6*B* and *C*), compared with those of the control values without DZ (1.51 ± 0.08 for alizarin red and 1.49 ± 0.06 for von Kossa staining, $n = 7$ each). To confirm the relationship between the increase in 6-MSITC-induced mineralisation efficiency and $HCO_3^-/Ca^{2+}$ transport, we examined the effects of DZ, the $Na^+$–$Ca^{2+}$ exchanger inhibitor KB-R7943, and the plasma membrane $Ca^{2+}$-ATPase (PMCA) inhibitor caloxin 1b1 on the mineralisation. 6-MSITC-induced mineralisation was significantly reduced by the simultaneous application of caloxin 1b1 (100 μM) or DZ (1 mM) with alizarin red (Fig. 7*A* and *C*) and von Kossa staining (Fig. 7*B* and *D*). Although KB-R7943 (10 μM) did not have any effects

on 6-MSITC-induced mineralisation in either staining, simultaneous application of KB-R7943 and DZ reduced mineralisation with alizarin red staining. Additionally, the simultaneous application of caloxin 1b1 and DZ significantly abolished 6-MSITC-induced mineralisation in both strains.

The application of 100 μM DIDS (a solute carrier family 4 [SLC4A] transporter and $Cl^-$ channel inhibitor) completely inhibited mineralisation by odontoblasts cultured in mineralisation medium, as shown by both alizarin red staining (Fig. 8*A* and *C*) and von Kossa staining (Fig. 8*B* and *D*). Furthermore, DIDS significantly reduced mineralisation efficiency in odontoblasts even in the presence of 50 μM 6-MSITC in the mineralisation medium, as evidenced by both alizarin red (Fig. 8*A* and *C*) and von Kossa staining (Fig. 8*B* and *D*).

## Immunofluorescence analyses

To examine the expression patterns of plasma membrane $HCO_3^-$ transporters and CA I and II in odontoblasts,

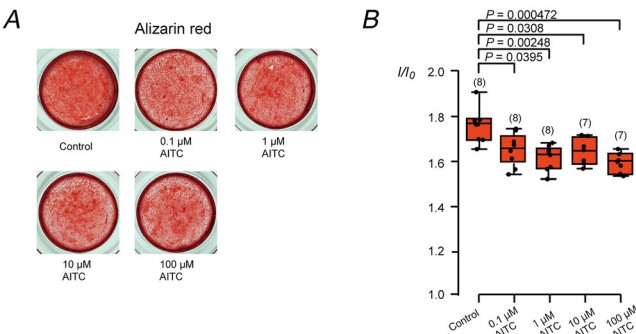

**Figure 3. AITC, a TRPA1 agonist, inhibited mineralisation elicited by odontoblasts**
*A*, we cultured human odontoblast cells for 28 days in a mineralisation medium without (control) or with 0.1, 1.0, 10 or 100 μM AITC. Mineralisation levels without (control) or with 0.1, 1.0, 10 or 100 μM AITC were evaluated using alizarin red staining. *B*, each box and whisker represent lower inner fence, 25% percentile, median, 75% percentile, and upper inner fence for mineralisation levels (as $I/I_0$ values) assessed using alizarin red. The number of each experiment is shown in parentheses. *P*-values showing statistically significant differences between columns (shown by solid lines) are illustrated. The dependence of changes in mineralisation level ($I/I_0$) on the concentrations of the extracellular AITC applied was obtained by fitting the data to the following function: $A = A_{max} - A_{min}/1 + ([x]_o/K_D)p + A_{min}$, where $K_D$ was the half maximal (50%) inhibitory concentration (IC$_{50}$) of the AITC (0.06 μM), $[x]_o$ indicates concentrations of extracellular AITC applied (0.1 μM to 100 μM), and $A_{max}$ and $A_{min}$ are maximal and minimal $I/I_0$ as mineralising levels, respectively. The Hill coefficient (*p*) was 2.0.

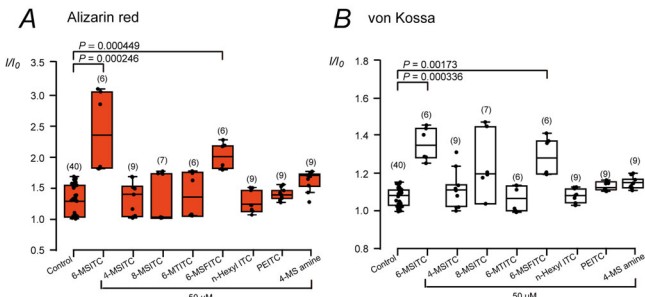

**Figure 4. Mineralisation efficacies of 6-MSITC and its derivatives**
*A* and *B*, we cultured human odontoblast cells for 28 days in a mineralisation medium without (control in *A* and *B*) and with 50 μM 6-MSITC, 4-MSITC, 8-MSITC, 6-MTITC, 6-MSFITC, *n*-Hexyl ITC, PEITC or 4-MS amine. Mineralisation levels (as $I/I_0$ values) were assessed through alizarin red (*A*) or von Kossa staining (*B*). Each box and whisker represent lower inner fence, 25% percentile, median, 75% percentile, and upper inner fence for mineralisation levels. The number of each experiment is shown in parentheses. *P*-values showing statistically significant differences between columns (shown by solid lines) are shown. Note that, for 6-MSFITC, a significant increase in mineralisation efficiency was observed at concentrations of 50 μM compared with that without 6-MSFITC, but not at that of 5 μM (1.157 ± 0.0788, $n = 11$, $P > 0.999$ with alizarin red and 1.044 ± 0.0278, $n = 7$, $P = 0.993$ with von Kossa staining; not shown).

immunofluorescence analysis was performed. We observed the expression of $HCO_3^-$-transporting solute carrier family 4 (SLC4A) members SLC4A1 (Fig. 9*A*), SLC4A2 (Fig. 9*B*), SLC4A3 (Fig. 9*C*), SLC4A4 (Fig. 9*D*; electrogenic $Na^+/HCO_3^-$ cotransporter-1: NBCe1), SLC4A8 (Fig. 9*E*; $Na^+$-driven $Cl^-/HCO_3^-$ exchanger-1: NDCBe1) and SLC4A9 (Fig. 9*F*) in human odontoblasts. The results revealed that SLC4A1, SLC4A2, SLC4A3, SLC4A4, SLC4A8 and SLC4A9 were constitutively expressed in the plasma membrane of human odontoblasts. Constitutive immunoreactivities of CA I (Fig. 10*A*) and II (Fig. 10*B*) were also observed.

## Gramicidin-perforated patch-clamp recording

Gramicidin-perforated whole-cell patch-clamp recordings were performed to evaluate the plasma

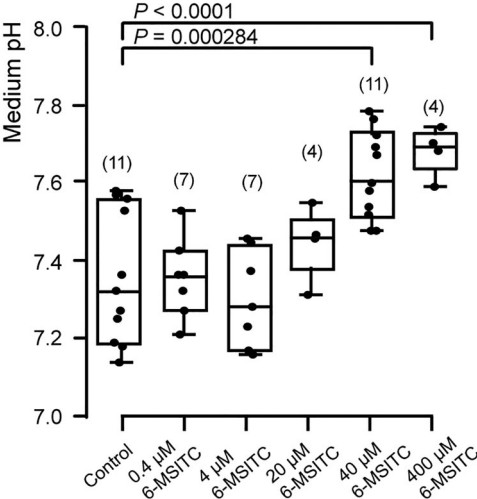

**Figure 5. 6-MSITC concentration-dependently increases the pH of the culture medium of human odontoblasts**
We cultured human odontoblast cells for 28 days in a mineralisation medium without (control) or with 0.4, 4.0, 20, 40 and 400 µM 6-MSITC. The culture medium was changed every 3 days using a mineralisation medium with or without a concentration of 6-MSITC. We collected culture medium (200 µL) from a well and measured the pH of the collected sample. The pH measurements were conducted three or four times for each well during the culture period from 14 to 28 days, and the values represent the grand averages of these measurements from 4 to 13 wells. Each box and whisker represents the lower inner limit, 25% percentile, median, 75% percentile, and upper inner limit for the pH values of the medium. The number of measurements is shown in parentheses. *P*-values indicating statistically significant differences between columns (solid lines) are shown. The relationship between changes in medium pH and the concentration of extracellular 6-MSITC applied was determined by fitting the data to the equation $A = A_{max} - A_{min}/1 + ([x]_o/K_D)p + A_{min}$, where $K_D$ was the half maximal (50%) effective concentration (EC$_{50}$) of the 6-MSITC (29.271 µM), $[x]_o$ indicates the concentration of extracellular 6-MSITC applied (0.4 µM to 400 µM), and $A_{max}$ and $A_{min}$ are maximal and minimal changes in medium pH, respectively. The Hill coefficient (*p*) was 1.0.

membrane $Ca^{2+}$ and/or $HCO_3^-$ transport machinery underlying dentin mineralisation and formation induced by 6-MSITC. The gramicidin-perforated patch-clamp recording allowed us to conserve intra-cellular circumstances intact during the recording, which enabled us to obtain the currents carried by anions, including $HCO_3^-$. We applied a voltage ramp protocol at a holding potential of $-70$ mV, with voltages ranging from $-100$ mV to $+100$ mV to obtain whole-cell current–voltage (*I–V*) relationships (insets in Fig. 11*A* and *C*). In the ECS with the KHR solution as a control, the voltage ramp protocol elicited inward (negative deflections, representing cation influx or anion efflux) and outward (positive deflections, representing cation efflux or anion influx) currents in odontoblasts. When we applied 5 µM 6-MSITC to odontoblasts, we observed significant increases in outward currents in the perforated patch-clamp recording (Fig. 11*A* and *B*). 6-MSITC significantly enhanced the outward current by 200.1 ± 138.7% at +100 mV. The 6-MSITC-induced increase in the inward and outward currents was not observed when we applied the conventional whole-cell patch-clamp recording mode (Fig. 11*C* and *D*). Application of 10 mM DZ significantly inhibited 6-MSITC (5 µM)-induced outward currents at +100 mV, but not inward currents at the hyperpolarised membrane potential of $-100$ mV (Fig. 12*A* and *B*). DZ (10 mM) significantly suppressed the 6-MSITC-induced outward current by 36.1 ± 15.5% at a membrane potential of +100 mV. DIDS, a non-selective SLC4A inhibitor, did not demonstrate any inhibitory effects on either the outward or inward currents elicited by 5 µM 6-MSITC (Fig. 12*C* and *D*), suggesting that DIDS-sensitive current components did not contribute to the whole-cell currents, but the currents were CA-sensitive. The application of 20 µM DMA significantly decreased 6-MSITC (5 µM)-induced outward currents by 52.1 ± 5.9% at a depolarised membrane potential of +100 mV (Fig. 12*E* and *F*). Application of 10 µM KB-R7943 significantly decreased 6-MSITC (5 µM)-induced outwards currents by 38.0 ± 4.3% at a depolarised membrane potential of +100 mV (Fig. 13*A* and *B*).

## 6-MSITC promoted dentin formation of rats *in vivo*

We found that 6-MSITC enhanced the mineralisation efficiency of odontoblasts. To evaluate whether 6-MSITC could promote reactionary dentin formation, we applied MedGel containing 500 µM 6-MSITC or saline to dentin cavities prepared in rats for 2 weeks. 6-MSITC significantly increased reactionary dentin formation (H-E staining, Fig. 14*A* and *B*), collagen production (Masson's trichrome staining, Fig. 14*C* and *D*), and calcium deposition (calcein labelling, Fig. 14*E* and *F*) in

rat dentin (Fig. 14). As expected, the calcein labelling experiment demonstrated that reactionary dentin formation was significantly accelerated by the application of 6-MSITC (calcein labelling; Fig. 14*E* and *F*). No significant differences were observed in dentin formation, collagen production, or calcium deposition between the male and female rats (data not shown).

## 6-MSITC application modulated expression of SLC4As, CA1 and CA2 mRNA in human odontoblasts

Figure 15 shows the expression of mRNA encoding SLC4A1 (Fig. 15*A*), SLC4A2 (Fig. 15*B*), SLC4A3 (Fig. 15*C*), SLC4A4 (Fig. 15*D*), SLC4A8 (Fig. 15*E*), and SLC4A9 (Fig. 15*F*), as well as CA1 (Fig. 15*G*) and CA2 (Fig. 15*H*) in human odontoblast, without (control) or with 12 h (each middle) and 24 h application (each most right) of 100 μM 6-MSITC. The expression of SLC4A1 (Fig. 15*A*; $P < 0.0001$, $n = 6$) and SLC4A2 (Fig. 15*B*; $P < 0.0001$, $n = 7$), as well as CA II (Fig. 15*H*; $P < 0.000900$, $n = 6$) mRNAs was significantly upregulated by a 12 h application of 100 μM 6-MSITC, compared with

levels without 6-MSITC application (control condition, $n = 18$ for each). In addition, CA I mRNA expression (Fig. 15*G*) was significantly higher than that of CA II (Fig. 15*H*) in the control (no exposure to 6-MSITC, $P < 0.0001$, $n = 17$ for CA I and 18 for CA II, not shown). Together the results from immunofluorescence analyses (Figs 9 and 10) indicate that SLC4A1, SLC4A2, SLC4A3, SLC4A4, SLC4A8 and SLC4A9, as well as CA I and CA II, were constitutively expressed in human odontoblasts.

## Discussion

The application of 40 μM 6-MSITC led to a significant increase in mineralisation efficiency. The 6-MSITC concentration of 40 μM, which promotes mineralisation, aligns with the $EC_{50}$ value (29.271 μM) for the concentration-dependent pH increase induced by 6-MSITC. 6-MSITC showed significant inhibition and a significant increase in mineralisation efficiency, depending on its concentration. Treatment of human odontoblasts with 6-MSITC resulted in a significant decrease in cell numbers at concentrations of 10 and

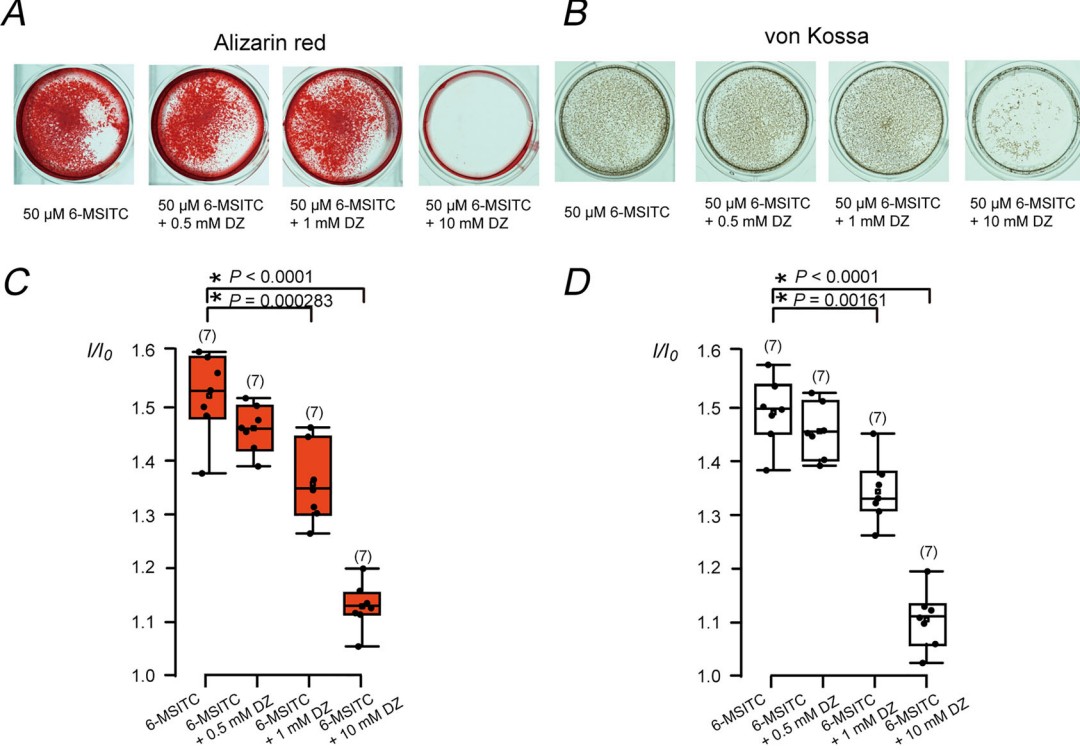

**Figure 6. The carbonic anhydrase inhibitor, dorzolamide, suppressed mineralisation by human odontoblasts**

*A* and *B*, we cultured human odontoblast cells for 28 days in a mineralisation medium containing 50 μM 6-MSITC without or with 0.5, 1.0 or 10 mM dorzolamide (DZ). Mineralisation levels were assessed using alizarin red (*A*) and von Kossa staining (*B*). *C* and *D*, each box and whisker represent the lower inner fence, 25% percentile, median, 75% percentile, and upper inner fence for mineralisation levels (as $I/I_0$ values) assessed by alizarin red (*C*) and von Kossa staining (*D*). The number of each experiment is shown in parentheses. *P*-values showing statistically significant differences between columns (shown by solid lines) are shown.

50 µM, whereas the cell numbers recovered at 500 µM. These results suggest that 6-MSITC exerted a bimodal effect, acting reciprocally on mineralisation and cell proliferation. At lower concentrations of 6-MSITC (10 and 20 µM), the observed decrease in mineralisation was likely attributable to a reduction in cell number, resulting from decreased cell proliferation rather than a direct reduction in mineralisation. At higher concentrations, both cell number and mineralisation capacity recovered, with mineralisation and cell proliferation specifically

enhanced at 100 µM and 500 µM, respectively. The detailed mechanism of the bimodal behaviour exerted on mineralisation and proliferation remains unclear, however. Although one possible explanation for the bimodal effects of 6-MSITC on mineralisation might be the overlapping influence of activation, inactivation and/or desensitisation on the intracellular 6-MSITC signalling pathway (i.e. CA activation) (Alpizar et al., 2013; Lee et al., 2008; Meents et al., 2019; Talavera et al., 2009) and its downstream ionic transporters

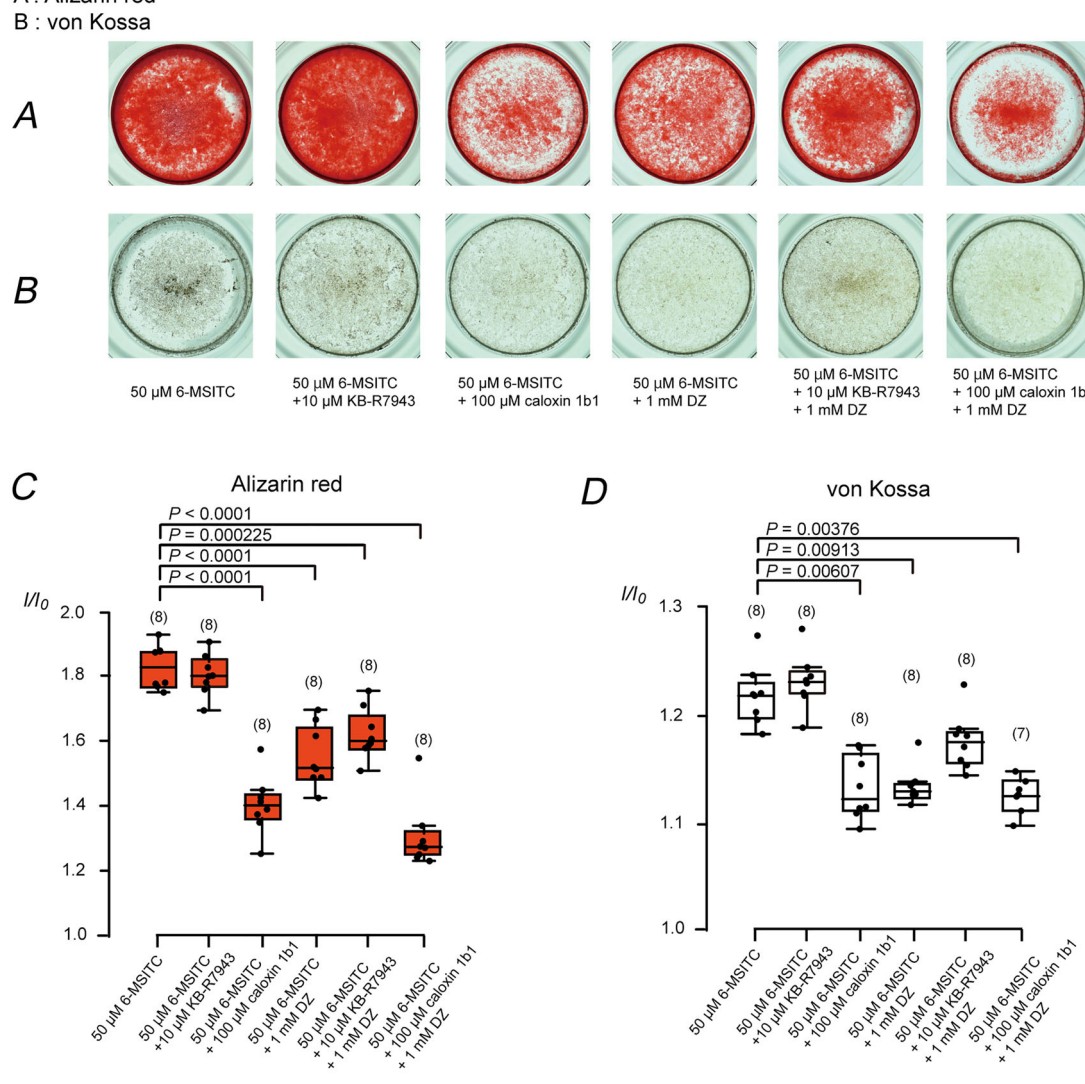

**Figure 7. Carbonic anhydrase inhibitor dorzolamide and plasma membrane Ca²⁺-ATPase (PMCA) inhibitor caloxin 1b1, but not Na⁺–Ca²⁺ exchanger inhibitor KB-R7943, suppressed mineralisation by human odontoblasts**

*A* and *B*, we cultured human odontoblast cells for 28 days in a mineralisation medium containing 50 µM 6-MSITC without or with 10 µM KB-R7943, 100 µM caloxin 1b1, 1 mM dorzolamide (DZ), 10 µM KB-R7943 with 1 mM DZ, or 100 µM caloxin 1b1 with 1 mM DZ. Mineralisation levels were assessed using alizarin red (*A*) and von Kossa staining (*B*). *C* and *D*, each box and whisker represent lower inner fence, 25% percentile, median, 75% percentile, and upper inner fence for mineralisation levels (as $I/I_0$ values) assessed using alizarin red (*C*) and von Kossa staining (*D*). The number of each experiment is shown in parentheses. *P*-values showing statistically significant differences between columns (shown by solid lines) are reflected.

(i.e. the SLC4As-NHE-NCX-PMCA axis; see below), further studies are required to elucidate the detailed mechanism underlying the bimodal effect of 6-MSITC on mineralisation/proliferation. Additionally, in the present study, AITC, a TRPA1 channel agonist, significantly decreased the mineralisation efficiency in odontoblasts. Although the 6-MSITC-induced decrease in mineralisation at concentrations up to 20 μM may involve a minor contribution from TRPA1 activation, the primary cause is likely a direct effect of 6-MSITC itself. These results also suggest that the 6-MSITC-induced increase in mineralisation occurs independently of TRPA1 channel activation.

6-MSITC and 6-MSFITC (in which the methylsulfinyl group of 6-MSITC was replaced with a methylsulfonyl group) showed increased mineralisation efficiency in odontoblasts. However, other compounds (4-MSITC, 8-MSITC, 6-MTITC, *n*-hexyl ITC, PEITC and 4-MS amine) did not affect mineralisation efficiency. Among these chemicals, 6-MSITC showed the highest mineralisation potency. When we compared the mineralisation efficiencies of the nine types of compounds, including 6-MSITC and its derivatives, isothiocyanate and methylsulfonyl or methylsulfinyl groups with six carbon chains were necessary to promote mineralisation. Interestingly, the mineralisation efficacy of 6-MSITC was higher than that of 6-MSFITC, indicating that the methylsulfinyl group was more potent in eliciting mineralisation than the methylsulfonyl group.

We also observed an increase in the pH of the medium when 6-MSITC was added to the odontoblast culture medium, suggesting that intracellular pH metabolism, $H^+/HCO_3^-$ production, and/or transport of cations/anions to the extracellular medium may play important roles in the enhancement of mineralisation by 6-MSITC. Simultaneous application of 6-MSITC with DZ and/or caloxin 1b1 to the culture medium resulted in a significant decrease in mineralisation efficiency, suggesting that transmembrane $H^+/HCO_3^-$ and $Ca^{2+}$ transport is involved in the promotion of mineralisation induced by 6-MSITC. These results are also in line with our previous report that $Ca^{2+}$ extrusion via PMCA plays a critical role in physiological dentin formation, as well as in pathological tertiary (reactionary) dentin formation,

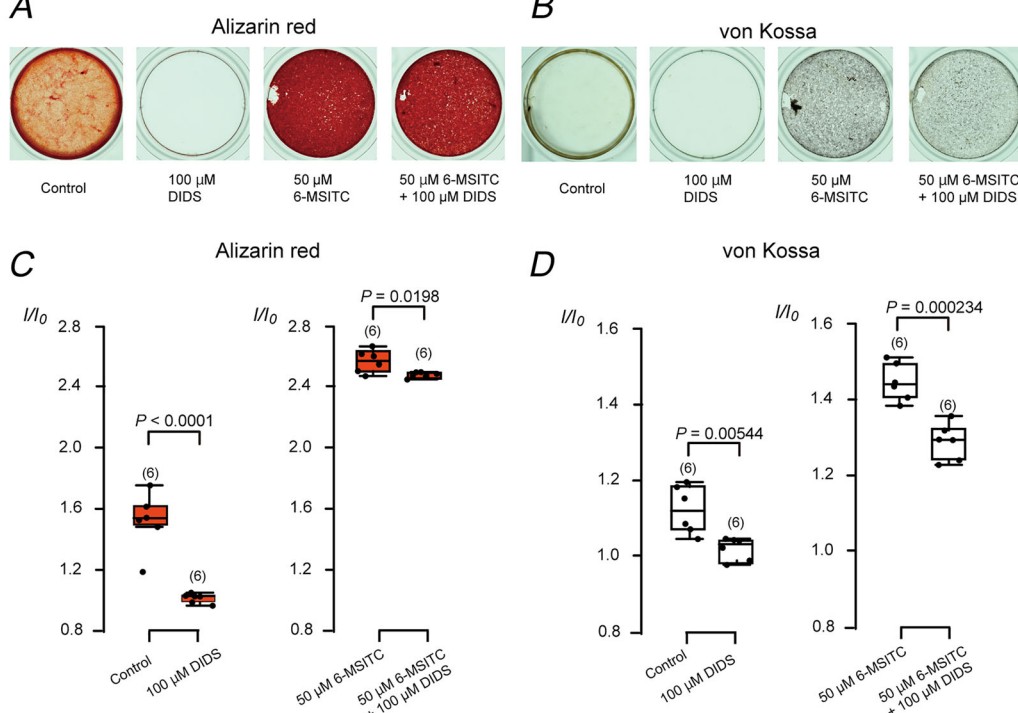

**Figure 8. Solute carrier family 4 (SLC4A) transporter/Cl⁻ channel inhibitor, 4,4′-diisothiocyanatostilbene-2,2′-stilbenedisulfonic acid (DIDS), suppressed both physiological- and 6-MSITC-induced mineralisation by human odontoblasts**
*A* and *B*, we cultured human odontoblast cells for 28 days in a mineralisation mediumwithout or with 50 μM 6-MSITC, 100 μM DIDS, or 50 μM 6-MSITC with 100 μM DIDS. Mineralisation levels were evaluated using alizarin red (*A*) and von Kossa staining (*B*). *C* and *D*, each box and whisker represent lower inner fence, 25% percentile, median, 75% percentile, and upper inner fence for mineralisation levels (as $I/I_0$ values) assessed using alizarin red (*C*) and von Kossa staining (*D*). The number of each experiment is shown in parentheses. *P*-values showing statistically significant differences between columns (shown by solid lines) are shown.

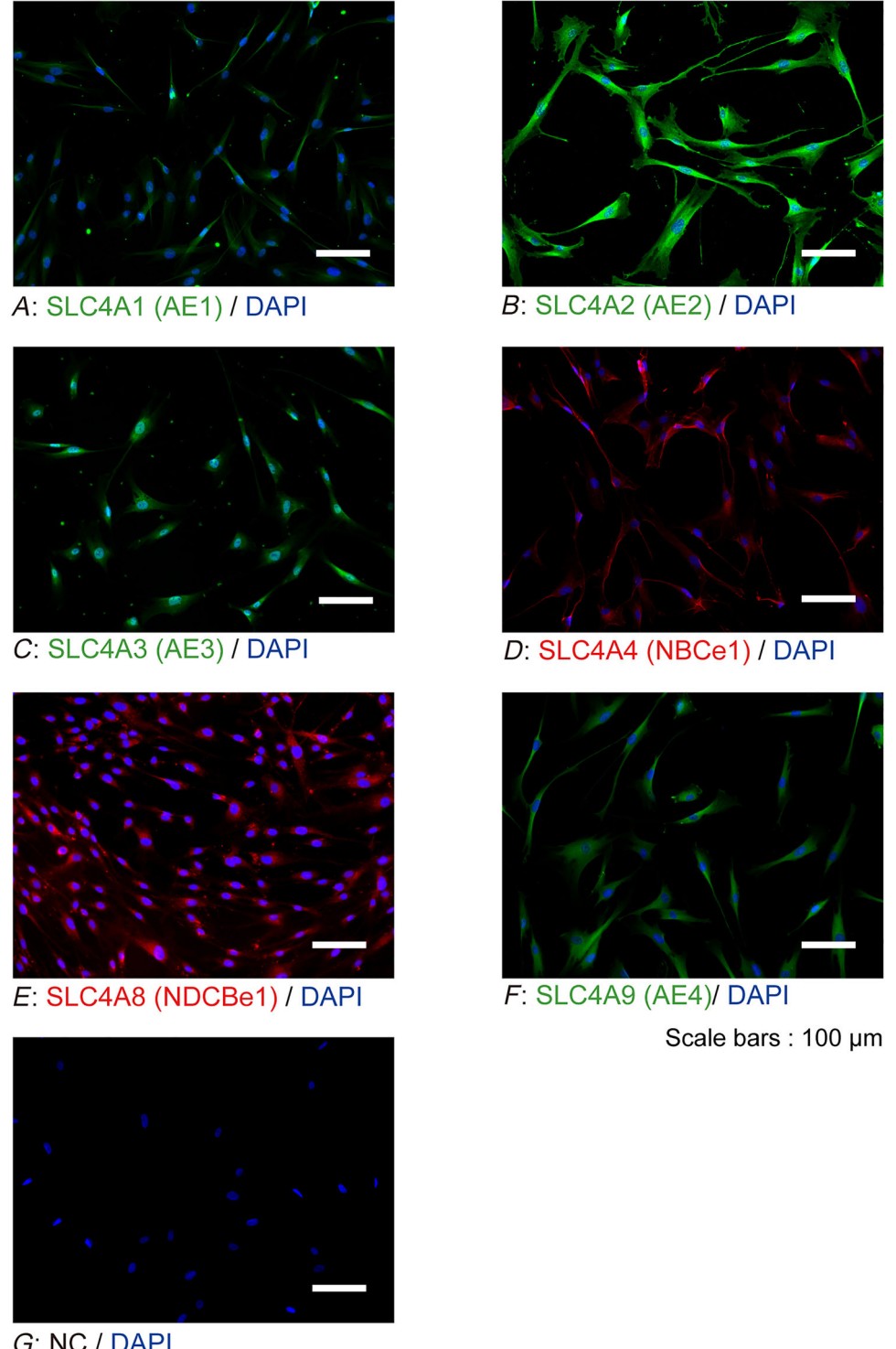

*A*: SLC4A1 (AE1) / DAPI

*B*: SLC4A2 (AE2) / DAPI

*C*: SLC4A3 (AE3) / DAPI

*D*: SLC4A4 (NBCe1) / DAPI

*E*: SLC4A8 (NDCBe1) / DAPI

*F*: SLC4A9 (AE4)/ DAPI

Scale bars : 100 μm

*G*: NC / DAPI

**Figure 9. Human odontoblasts constitutively express $HCO_3^-$ transporters SLC4A1 (anion exchanger 1; AE1), SLC4A2 (AE2), SLC4A3 (AE3), SLC4A4 (electrogenic $Na^+/HCO_3^-$ cotransporter-1; NBCe1), SLC4A8 ($Na^+$-driven $Cl^-/HCO_3^-$ exchanger-1: NDCBe1) and SLC4A9 (AE4)**

*A–G*, human odontoblasts showing positive immunoreactivity to SLC4A1 (green in *A*), SLC4A2 (green in *B*), SLC4A3 (green in *C*), SLC4A4 (red in *D*), SLC4A8 (red in *E*), and SLC4A9 (green in *F*). Nuclei (DAPI) are shown in blue. Scale bars: 100 μm. No fluorescence was detected in the negative controls (*G*).

which is induced by multiple external stimuli applied to the dentin surface, including high-pH treatments that mimic the effects of alkaline dental materials (Kimura et al., 2021).

SLC4A1 (anion exchanger-1: AE1), SLC4A2 (AE2), SLC4A3 (AE3), SLC4A4 (NBCe1), SLC4A8 (NDCBe1) and SLC4A9 (AE4) were constitutively expressed in human odontoblasts. SLC4A1–3 act as $Na^+$-independent DIDS-sensitive $Cl^-$–$HCO_3^-$ exchangers (Choi, 2012; Nguyen et al., 2004; Romero et al., 2013). SLC4A4, an electrogenic $Na^+$–$HCO_3^-$ cotransporter-1 (NBCe1), is also DIDS-sensitive, cotransports one $Na^+$ ion with two to three $HCO_3^-$ ions, and requires an $Na^+$ gradient across the plasma membrane (Choi, 2012; Jalali et al., 2014). SLC4A4 expression has also been reported in odontoblasts (Lacruz et al., 2010). $SLC4A4^{-/-}$ ($NBCe1^{-/-}$) animals have a significantly softer dentin (Lacruz et al., 2010). SLC4A9 (AE4) is functionally characterised as a DIDS-sensitive $Cl^-$–$HCO_3^-$ exchanger; however, it

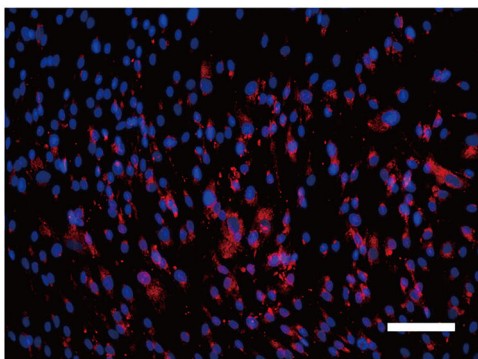

*A*: CA I / DAPI

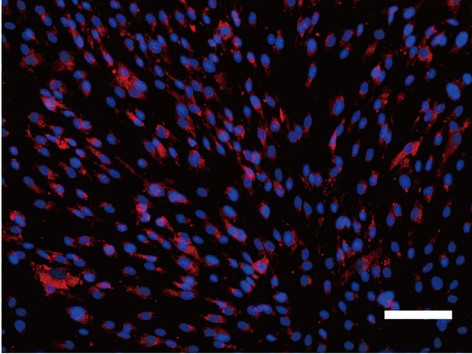

*B*: CA II / DAPI
Scale bars : 100 µm

**Figure 10. Expression of carbonic anhydrase I (CA I) and CA II in odontoblasts**
*A* and *B*, human odontoblasts constitutively express CA I (red in *A*) and CA II (red in *B*). Nuclei (DAPI) are shown in blue. Scale bars: 100 µm. No fluorescence was detected in the negative controls (not shown).

has also been suggested to be an electroneutral NBC without $Cl^-$–$HCO_3^-$ exchange activity (Choi, 2012). SLC4A8 (NDCBe1), a DIDS-sensitive $Na^+$-driven $Cl^-$–$HCO_3^-$ exchanger, is predominantly expressed in neurons (Choi, 2012). SLC4-family members are integral membrane proteins (Romero et al., 2013). Mutations in the SLC4A2, SLC4A4 and SLC4A9 transporters induce enamel anomalies because these proteins play important roles in pH regulation during enamel development by enamel-forming ameloblasts (Jalali et al., 2014; Lacruz et al., 2010; Lyaruu et al., 2008; Reibring et al., 2014; Yin & Paine, 2017). Members of the SLC4A family of odontoblasts, which are the dentin-forming cells, are likely involved in regulating extracellular pH by mediating the exclusion of $HCO_3^-$ during dentin mineralisation and/or formation (see below).

Additionally, human odontoblasts expressed CA I and CA II. In previous studies, CA activity has been detected histochemically in preodontoblasts and odontoblasts (Dogterom & Bronckers, 1983; Reibring et al., 2014; Sugimoto et al., 1988). CA II, VI and XIII have also been detected in the preodontoblasts and dental papilla mesenchyme of mice (Reibring et al., 2014). CA regulates intracellular and extracellular volumes by determining $[H^+]$ and $[HCO_3^-]$, maintaining intracellular and extracellular pH, and is involved in the processing of excess $CO_2$ (as $HCO_3^-$) produced by high metabolic activity in cells expressing $HCO_3^-$ transporters (Reibring et al., 2014). Our results revealed that extracellular pH changes in odontoblast medium, the CA inhibitor-induced reduction of mineralisation efficiencies by 6-MSITC, and the expression of transmembrane $HCO_3^-$ transporters of the SLC4A family are closely involved in promoting mineralisation. Additionally, the results showing the inhibitory effect of $Ca^{2+}$ extrusion by PMCA inhibitors on mineralisation efficiency suggests that they are involved in 6-MSITC-induced dentin mineralisation (see below).

In this study, we used gramicidin-perforated patch-clamp recordings to confirm the relationship between increased mineralisation efficiency and ionic transport. In both conventional and gramicidin-perforated whole-cell patch-clamp recordings, macroscopic transmembrane currents were observed using extracellular $Na^+$-rich and intracellular $K^+$-rich solutions. These currents in human odontoblasts (i.e. the same cell line used in this study) were carried by intermediate-conductance $Ca^{2+}$-activated $K^+$ channels, as well as by voltage-dependent $K^+$ channel α-subunits (Kv) 1.1 and Kv1.2. Notably, these Kv channels were consistently expressed in both human and rat odontoblasts (Ichikawa et al., 2012; Kojima et al., 2017). In the gramicidin-perforated mode of whole-cell patch-clamp recording, the application of 5 µM 6-MSITC significantly increased outwards currents in human

## Gramicidin-perforated patch-clamp recording

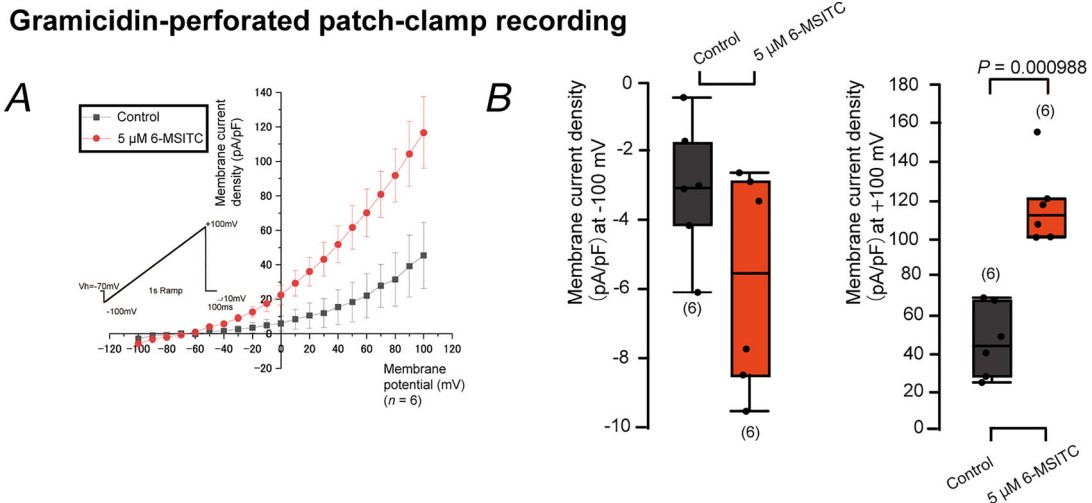

## Conventional patch-clamp recording

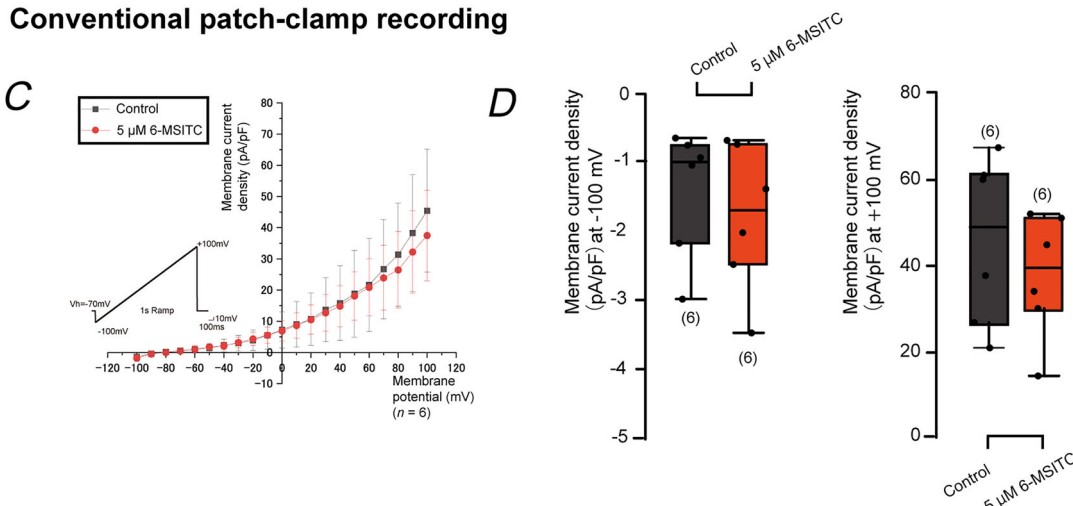

**Figure 11. 6-MSITC induced inward and outward currents were observed in gramicidin-perforated mode, but not in conventional mode of whole-cell patch-clamp recordings**

*A* and *C*, in the gramicidin-perforated mode (*A*) or conventional mode (*C*) of whole-cell patch-clamp recordings, ramp currents elicited by the voltage ramp protocol (inset in *A* and *C*) from −100 mV to +100 mV at a $V_h$ of −70 mV were recorded with an extracellular solution of KHR solution and intracellular solution (ICS). Current–voltage (*I–V*) relationships were obtained from ramp currents measuring current densities (pA/pF) every 10 mV from −100 mV to +100 mV without (black circles) or with 5 μM 6-MSITC (red circles). Each point indicates a mean ± SD of six experiments for *A* and *C*. *B* and *D*, each box and whisker represents the lower inner fence, 25% percentile, median, 75% percentile, and upper inner fence for values of current densities (pA/pF) at membrane potentials of −100 mV (left in *B* and *D*) and +100 mV (right in *B* and *D*) in gramicidin-perforated mode (*B*) or conventional mode (*D*) of whole-cell patch-clamp recordings. In the gramicidin-perforated mode, the values of the current densities were −2.995 ± 1.963 pA/pF without and −5.716 ± 3.154 pA/pF with 5 μM 6-MSITC at −100 mV (each *n* = 6), and +45.505 ± 19.175 pA/pF without and +116.673 ± 20.725 pA/pF with 5 μM 6-MSITC at +100 mV (*P* = 0.000988, each *n* = 6). In the conventional mode, the values of the current densities were −1.402 ± 0.942 pA/pF without and −1.777 ± 1.0825 pA/pF with 5 μM 6-MSITC at −100 mV (each *n* = 6), and +45.433 ± 19.732 pA/pF without and +35.512 ± 14.484 pA/pF with 5 μM 6-MSITC at +100 mV (each *n* = 6). The numbers for each experiment from a cell are shown in parentheses. *P*-values indicating statistically significant differences between columns (solid lines) are shown.

**Effect of DZ**
**on 6-MSITC-induced current: Gramicidin-perforated mode**

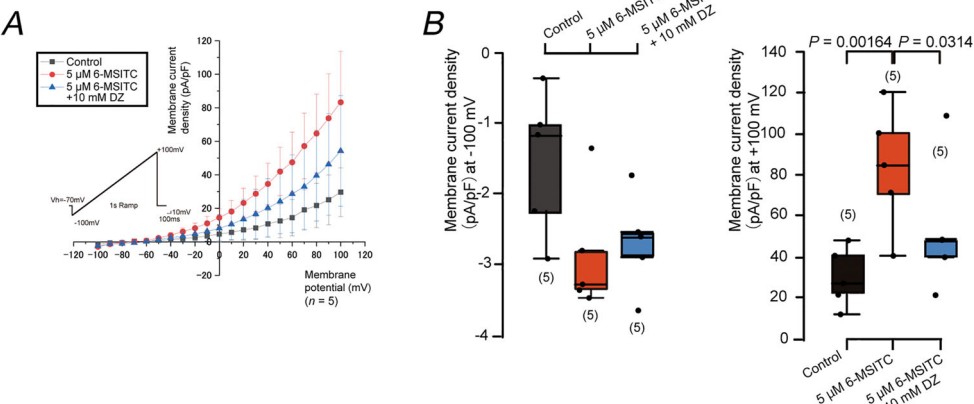

**Effect of DIDS**
**on 6-MSITC-induced current: Gramicidin-perforated mode**

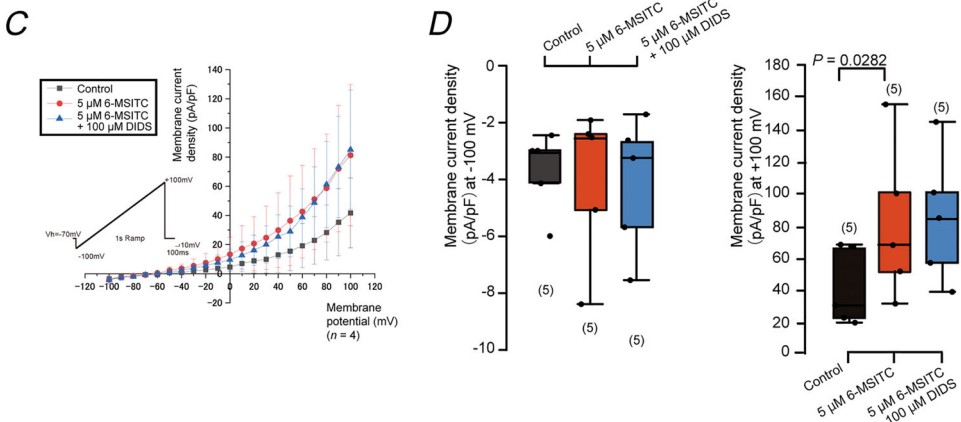

**Effect of DMA**
**on 6-MSITC-induced current: Gramicidin-perforated mode**

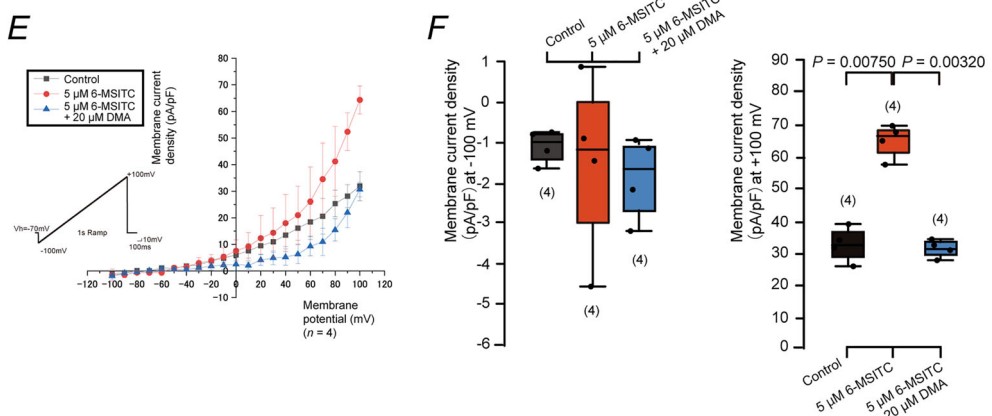

**Figure 12. 6-MSITC-induced outward currents are sensitive to the carbonic anhydrase inhibitor dorzolamide (DZ) and Na$^+$–H$^+$ exchanger inhibitor 5-(*N*,*N*-dimethyl) amiloride hydrochloride (DMA), but not to the solute carrier family 4 (SLC4A) transporter/Cl$^-$ channel inhibitor 4,4′-diisothiocyanatostilbene-2,2′-stilbenedisulfonic acid (DIDS)**

*A*, *C* and *E*, in the gramicidin-perforated mode of whole-cell patch-clamp recordings, we recorded ramp currents elicited by a voltage ramp protocol (inset in *A*, *C* and *E*) from −100 mV to +100 mV at a $V_h$ of −70 mV with an extracellular solution of KHR solution and intracellular solution (ICS). Current–voltage (*I–V*) relationships in *A*

were obtained from ramp currents measuring current densities (pA/pF) every 10 mV from $-100$ mV to $+100$ mV without (black rectangles), with 5 μM 6-MSITC (red circles) or with 5 μM 6-MSITC and 10 mM DZ (blue triangles). *I–V* relationships were also obtained by measuring current densities (pA/pF) without (black rectangles in *C* and *E*), with 5 μM 6-MSITC (red circles in *C* and *E*), with 5 μM 6-MSITC and 100 μM DIDS (blue triangles in *C*), or with 5 μM 6-MSITC and 20 μM DMA (blue triangles in *E*). Each point indicates a mean ± SD of five experiments for *A* and four experiments for *C* and E, respectively. *B*, *D* and *F*, each box and whisker represents the lower inner fence, 25% percentile, median, 75% percentile, and upper inner fence for values of current densities (pA/pF) at membrane potentials of $-100$ mV (left in *B*, *D* and *F*) and $+100$ mV (right in *B*, *D* and *F*) under the gramicidin-perforated mode of whole-cell patch-clamp recordings. In *B*, at the membrane potential of $-100$ mV, the values of current density were $-1.528 \pm 1.027$ pA/pF under the control condition, $-2.847 \pm 0.881$ pA/pF with 5 μM 6-MSITC, and $-2.671 \pm 0.687$ pA/pF 5 μM 6-MSITC plus 10 mM DZ (each *n* = 6). At the membrane potential of $+100$ mV, the values of the current density were $+29.647 \pm 14.563$ pA/pF under the control condition, $+83.338 \pm 30.225$ pA/pF with 5 μM 6-MSITC ($P = 0.00164$ compared with control), and $+54.246 \pm 32.993$ pA/pF 5 μM 6-MSITC plus 10 mM DZ ($P = 0.00314$ compared with the value exposed to 5 μM 6-MSITC) (each *n* = 6). In *D*, at the membrane potential of $-100$ mV, the values of the current density were $-3.732 \pm 1.423$ pA/pF under the control condition, $-4.081 \pm 2.730$ pA/pF with 5 μM 6-MSITC, and $-4.186 \pm 2.411$ pA/pF 5 μM 6-MSITC plus 100 μM DIDS (each *n* = 5). At the membrane potential of $+100$ mV, the values of the current density were $+41.6732 \pm 23.979$ pA/pF under the control condition, $+81.309 \pm 48.380$ pA/pF with 5 μM 6-MSITC ($P = 0.0282$ compared with control), and $+85.127 \pm 40.891$ pA/pF 5 μM 6-MSITC plus 100 μM DIDS (each *n* = 5). In *F*, at the membrane potential of $-100$ mV, the values of the current density were $-1.092 \pm 0.417$ pA/pF under the control condition, $-1.506 \pm 2.264$ pA/pF with 5 μM 6-MSITC, and $-1.855 \pm 1.039$ pA/pF 5 μM 6-MSITC plus 20 μM DMA (each *n* = 4). At the membrane potential of $+100$ mV, the values of the current density were $+31.953 \pm 5.484$ pA/pF under the control condition, $+64.361 \pm 5.307$ pA/pF with 5 μM 6-MSITC ($P = 0.00750$ compared with control), and $+30.675 \pm 2.783$ pA/pF 5 μM 6-MSITC plus 20 μM DMA ($P = 0.00320$ compared with the value exposed to 5 μM 6-MSITC) (each *n* = 4). The numbers for each experiment from a cell are shown in parentheses. *P*-values indicating statistically significant differences between columns (solid lines) are shown.

odontoblasts, whereas 6-MSITC-induced whole-cell currents could not be observed in the conventional mode, indicating that 6-MSITC activates ionic permeability that originates from intact intracellular space and/or from the ions produced by 6-MSITC action, but not $Cl^-$ or $K^+$. The linear pentadecapeptide antibiotic gramicidin is a naturally occurring product of *Bacillus bravis* and is known to form monovalent cation-selective channels in both synthetic and natural membranes (Andersen & Koeppe, 1992; Mueller & Rudin, 1967; Sugita et al., 2004). Although the structure of the active gramicidin channels remains controversial, gramicidin in the ICS inside the patch pipette forms a pore in the patch membrane that is completely impermeable to anions, with electrical access to the cell's interior (Allen et al., 2003; Burkhart et al., 1998; Kovacs et al., 1999; Sugita et al., 2004). This indicates that the intracellular environment remained intact during gramicidin-perforated patch-clamp recordings, whereas in the conventional mode, the ICS fully replaced the intracellular ionic environment. We showed that 6-MSITC did not affect the amplitudes of whole-cell currents in the conventional mode but increased the amplitudes of the current in the perforated mode. This indicated that 6-MSITC activates the permeability driven by cations/anion situated in the purely intact intracellular environment. Notably, in both modes of patch-clamp recording, we could not record any recovery effects of 6-MSITC (i.e. the current amplitudes induced by the application of 6-MSITC did not return to the control level after removing 6-MSITC from the ECS). This may be because of its high fat solubility.

In the perforated mode of the patch-clamp recordings, application of the CA inhibitor DZ resulted in a decrease in the 6-MSITC-induced outward currents. This phenomenon was due to a decrease in intracellular $H^+$ and $HCO_3^-$ concentrations. SLC4As, except SLC4A7, are DIDS-sensitive; however, DIDS application did not affect the whole-cell currents elicited by 6-MSITC. Although patch-clamp recordings can detect trans-membrane charge movements such as those caused by electrogenic transporters, electroneutral $HCO_3^-$ transport (i.e. SLC4A1, SLC4A2, SLC4A3, SLC4A8 and SLC4A9) activities cannot be recorded by patch-clamp recording. This might also explain why DIDS (a non-specific $HCO_3^-$ transporting SLC4A inhibitor) did not affect the 6-MSITC-induced currents, while DIDS significantly suppressed mineralisation efficiency. Among the constitutively expressed SLC4A family members observed in odontoblasts through immunofluorescence analysis, namely, SLC4A1, SLC4A2, SLC4A3, SLC4A4, SLC4A8 and SLC4A9, only SLC4A4 was identified as a DIDS-sensitive electrogenic transporter. These findings suggest that SLC4A4 is unlikely to contribute to the 6-MSITC-induced outward currents and mineralisation. Furthermore, after a 12-h exposure to 6-MSITC treatment in odontoblasts, the mRNA expression of SLC4A1, SLC4A2 and CA II was upregulated. Thus, during 6-MSITC exposure, the differential expression of electroneutral SLC4A family members and CA plays a role in modulating dentinogenesis by mediating the production and exclusion of extracellular $HCO_3^-$.

The inhibitory effect of DZ on 6-MSITC-induced currents and mineralisation aligns with the significant constitutive expression of CA at mRNA and protein levels, suggesting that CA plays a crucial role in $HCO_3^-$ production, not only during 6-MSITC application but also throughout aerobic ATP synthesis in odontoblasts.

In addition to $HCO_3^-$, CA produces $H^+$, and it has been reported that intracellular $H^+$ is excluded by the $Na^+$ gradient through the $Na^+$–$H^+$ exchanger (NHE) (Bers et al., 2003). DMA, an NHE inhibitor, suppresses the 6-MSITC-induced current; however, NHE is also electrically neutral, suggesting that another cation efflux, besides NHE activity, mediates the current conductance elicited by 6-MSITC in odontoblasts.

Plasma membrane NCX are bidirectional transporters, or antiporters, that catalyse the electrogenic exchange of three or four $Na^+$ for one $Ca^{2+}$ and are tightly involved in dentin mineralisation. NCX carries out either $Ca^{2+}$ efflux or influx ($Ca^{2+}$ efflux with $Na^+$ influx by forward exchange or $Ca^{2+}$ influx with $Na^+$ efflux by reverse exchange) depending on the direction of the trans-membrane $Na^+$ gradient (Kimura et al., 2016, 2021; Tsumura et al., 2010, 2012). $Na^+$ incorporation via the transmembrane $Na^+$ gradient, which couples with $Ca^{2+}$ extrusion to the dentin-forming/mineralising front (i.e. to the predentin region) via NCX in odontoblasts, results in a lower concentration of $Na^+$ at the mineralising front of the predentin region (Tsumura et al., 2010). In the present study, we showed that 6-MSITC induces the activation of electrically neutral NHE activity to incorporate $Na^+$ ($Na^+$ influx). $Na^+$ influx induced by NHE may 'further' reverse the transmembrane $Na^+$ gradient in the odontoblast microenvironment. Increased intracellular $Na^+$ drove NCX in the reverse mode, and the subsequent $Na^+$ efflux was observed as 6-MSITC-induced outward currents. Thus, we observed that KB-R7943, the NCX inhibitor, suppressed 6-MSITC-induced outward currents. KB-R7943, however, did not affect 6-MSITC-induced mineralisation efficiency, as 6-MSITC subsequently activated the $Ca^{2+}$ influx mode of the reverse exchange of NCX, but not $Ca^{2+}$ extrusion to the mineralising front. Notably, the application of the PMCA inhibitor with

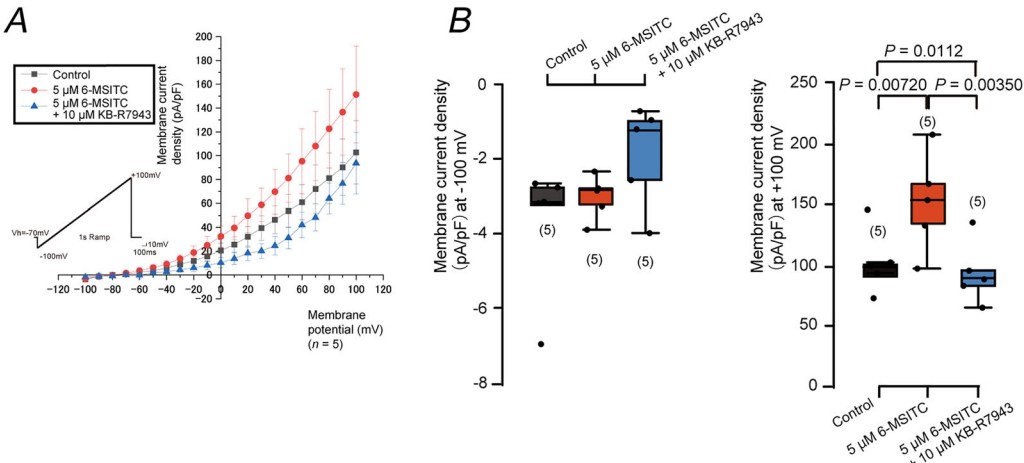

**Figure 13. 6-MSITC-induced outward currents are sensitive to the $Na^+$–$Ca^{2+}$ exchanger inhibitor KB-R7943**

*A*, in the gramicidin-perforated mode of whole-cell patch-clamp recordings, we recorded ramp currents elicited by the voltage ramp protocol (inset in *A*) from −100 mV to +100 mV at a $V_h$ of −70 mV with an extracellular solution of KHR and intracellular solution (ICS). Current–voltage (*I–V*) relationships were obtained from ramp currents measuring current densities (pA/pF) every 10 mV from −100 mV to +100 mV without (black rectangles), with 5 μM 6-MSITC (red circles), or with 5 μM 6-MSITC and 10 μM KB-R7943 (blue triangles). Each point indicates a mean ± SD of five experiments. *B*, each box and whisker represents the lower inner fence, 25% percentile, median, 75% percentile, and upper inner limit for values of current densities (pA/pF) at membrane potentials of −100 mV (left) and +100 mV (right) under the gramicidin-perforated mode of whole-cell patch-clamp recordings. In *B*, at the membrane potential of −100 mV, the values of the current density were −3.728 ± 1.811 pA/pF under the control condition, −3.012 ± 0.585 pA/pF with 5 μM 6-MSITC, and −1.873 ± 1.371 pA/pF 5 μM 6-MSITC plus 10 μM KB-R7943 (each *n* = 5). At the membrane potential of +100 mV, the values of the current density were +102.802 ± 26.519 pA/pF under the control condition, +151.458 ± 40.679 pA/pF with 5 μM 6-MSITC (*P* = 0.00720 compared with control), and +93.611 ± 25.885 pA/pF with 5 μM 6-MSITC plus 10 μM KB-R7943 (*P* = 0.00350 compared with the value exposed to 5 μM 6-MSITC) (each *n* = 5). The numbers for each experiment from a cell are shown in parentheses. *P*-values indicating statistically significant differences between columns (solid lines) are shown.

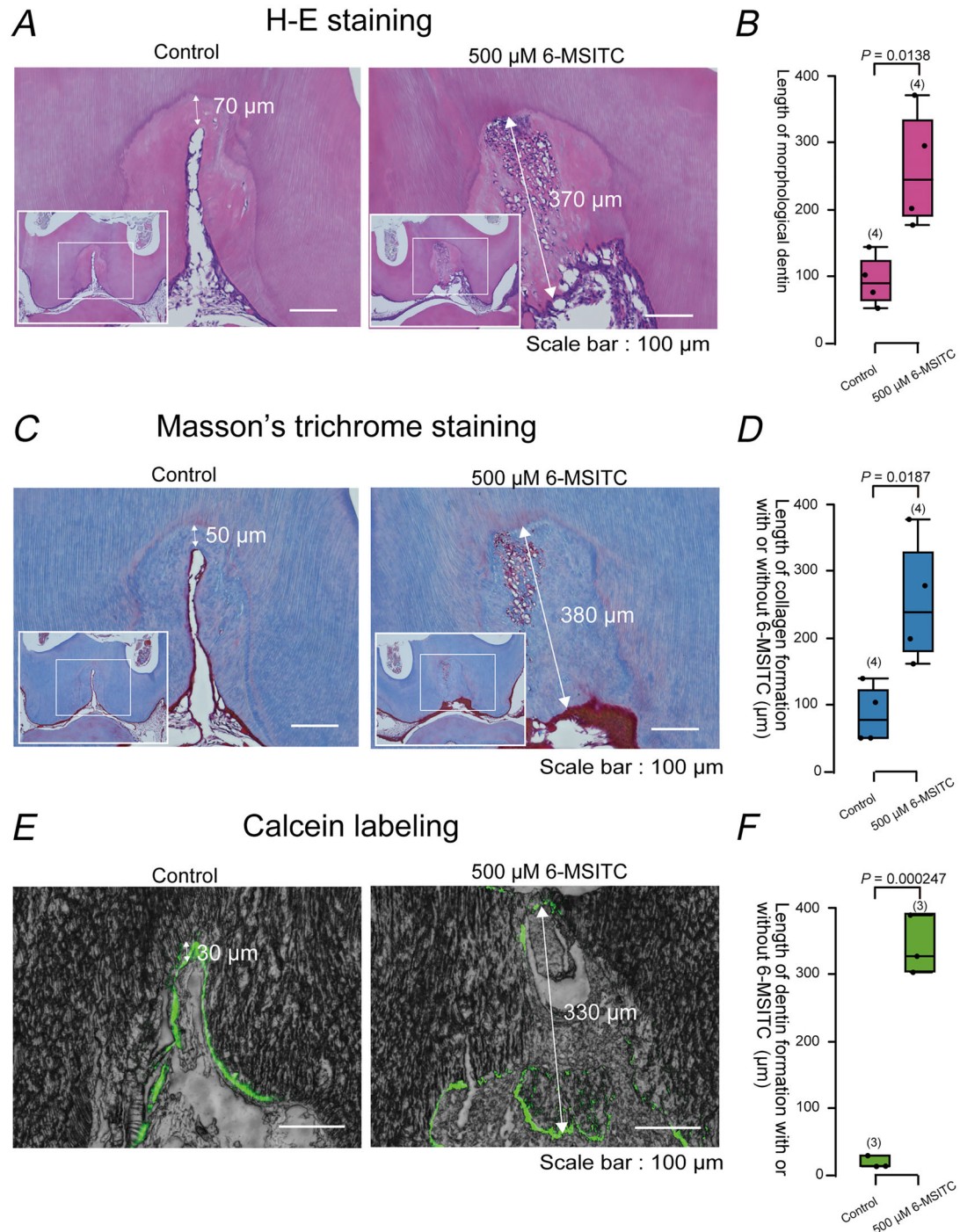

**Figure 14. 6-MSITC induces reactionary dentinogenesis as a novel and potent dentin regenerative drug**
*A*, *C* and *E*, we prepared cavities inside the dentin without pulpal exposure at the occlusal surface on both sides of the mandibular first molars in Wistar rats. Morphological observations demonstrated significant increase in reactionary dentin formation, when we placed MedGel containing 500 µM 6-MSITC (right panels in *A*, *C* and *E*), compared to that containing saline (left panels in *A*, *C* and *E* as control). Morphological changes were observed by haematoxylin and eosin staining (*A*), collagen production by Masson's trichrome staining (*C*), and calcium deposition by calcein labelling (green in *E*) during 6-MSITC-induced reactionary dentin formation. Large inset (large white boxes) at the bottom left of *A* and *C* shows low magnification images, while white small boxes inside the large inset show the observation area for high magnification of *A* and *C*. Double arrows in *A*, *C* and *E* indicate the increased width of reactionary dentin formation without (as control; left panels in *A*, *C* and *E*) or with 500 µM 6-MSITC (right panels in *A*, *C* and *E*). Scale bars: 100 µm. *B*, *D* and *F*, each box and whisker represent lower inner fence, 25% percentile, median, 75% percentile, and upper inner fence for increased width of reactionary

dentin formation without (control in *B*, *D* and *F*) or with 500 μM 6-MSITC (right in *B*, *D* and *F*) assessed by H-E staining (*B*), Masson's trichrome staining (*D*), or calcein labelling (*F*). 6-MSITC significantly increased reactionary dentin formation to 259.75 ± 89.593 μm, compared with the control condition (91.5 ± 39.373 μm, each *n* = 4, *P* = 0.0138) in H-E staining, 255.5 ± 96.476 μm, compared with the control condition (86.0 ± 44.091 μm, each *n* = 4, *P* = 0.0187) in Masson's trichrome staining, and 340.0 ± 44.099 μm, compared with the control condition (18.9 ± 9.185 μm, each *n* = 3, *P* = 0.000247) in calcein labelling. The numbers for each experiment are shown in parentheses. *P*-values indicating statistically significant differences between columns (solid lines) are shown.

and without the CA inhibitor significantly suppressed 6-MSITC-induced mineralisation, indicating that intracellular $Ca^{2+}$ accumulation via the $Ca^{2+}$ influx-reverse mode of NCX activity was extruded by PMCA to promote reactionary/regenerative dentinogenesis.

In summary, for the *in vitro* experiments (Fig. 16), 6-MSITC activated intracellular CA activity to produce $H^+$ and $HCO_3^-$. SLC4As (SLC4A1, SLC4A2, SLC4A3, SLC4A8 and SLC4A9) play important roles in the electro-neutral extrusion of $HCO_3^-$. NHE excludes $H^+$ by

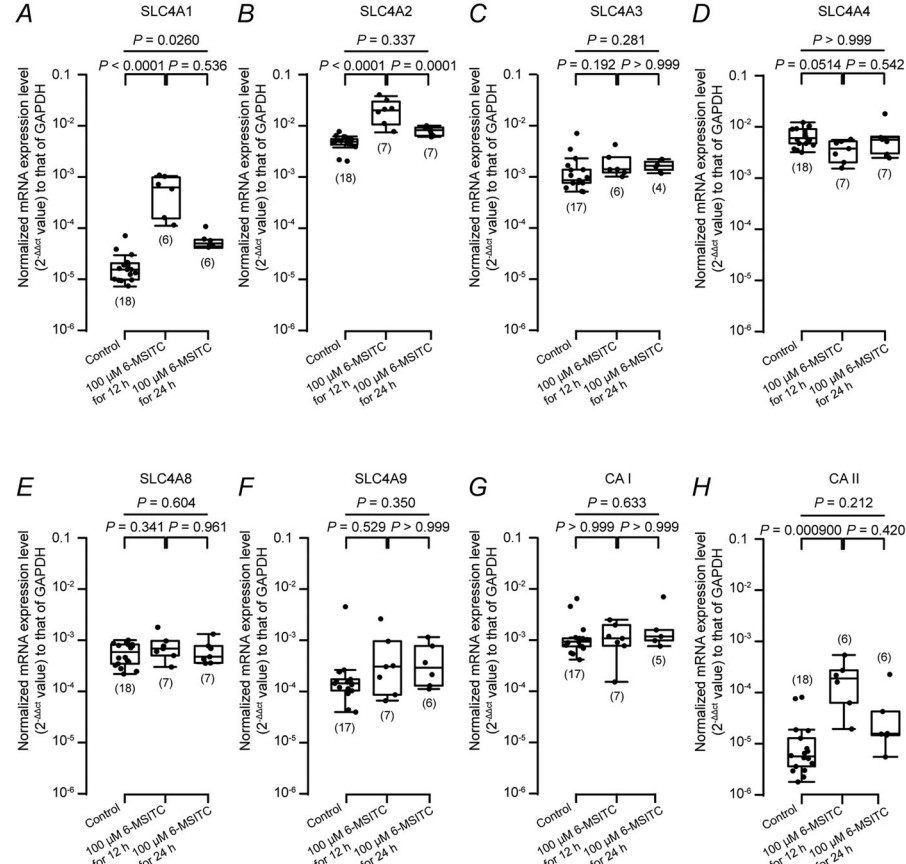

**Figure 15. Wasabi ingredient, 6-methylsulfinylhexyl (6-MSITC), modulates mRNA expression of solute carrier family 4 members (SLC4As) and carbonic anhydrase (CA) in human odontoblast cells**
*A* to *H*, bar graphs show normalised mRNA expression of SLC4A1 (*A*), SLC4A2 (*B*), SLC4A3 (*C*), SLC4A4 (*D*), SLC4A8 (*E*), and SLC4A9 (*F*), as well as CA I (*G*) and CA II (*H*), without (control) or with 12 h (each middle) and 24 h application (each most right) of 100 μM 6-MSITC in human odontoblasts. Real-time RT-PCR findings show mRNAs levels quantified by measuring the increase in fluorescence induced by the binding of SYBR green dye to double-stranded DNA. Data were analysed using the $2^{-\Delta\Delta C_t}$ method, with GAPDH as an internal control, which was positive in all samples. Expression of each mRNA was normalised to the GAPDH mRNA level. The number of independent experiments is shown in parentheses. *P*-values indicating statistically significant differences between columns (solid lines) are shown. Note that, among the subfamily of SLC4As, the detected level of mRNA showed significant differences between SLC4A1 and SLC4A2 (*P* < 0.0001), SLC4A1 and SLC4A3 (*P* < 0.0001), SLC4A1 and SLC4A4 (*P* < 0.0001), SLC4A1 and SLC4A8 (*P* = 0.00430), SLC4A2 and SLC4A8 (*P* = 0.00430), SLC4A2 and SLC4A9 (*P* < 0.0001), and SLC4A3 and SLC4A4 (*P* = 0.0449). No significant differences were observed between SLC4A1 and SLC4A9 (*P* = 0.763), SLC4A2 and SLC4A3 (*P* = 0.281), SLC4A2 and SLC4A4 (*P* > 0.999), SLC4A3 and SLC4A8 (*P* > 0.999), SLC4A3 and SLC4A9 (*P* = 0.0759), and SLC4A8 and SLC4A9 (*P* > 0.999).

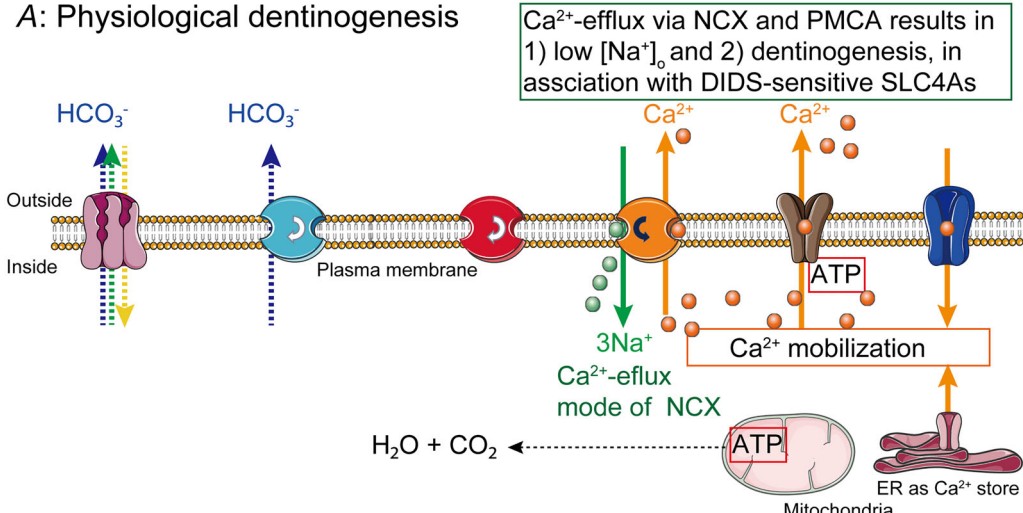

**A: Physiological dentinogenesis**

Ca²⁺-efflux via NCX and PMCA results in 1) low [Na⁺]ₒ and 2) dentinogenesis, in asscciation with DIDS-sensitive SLC4As

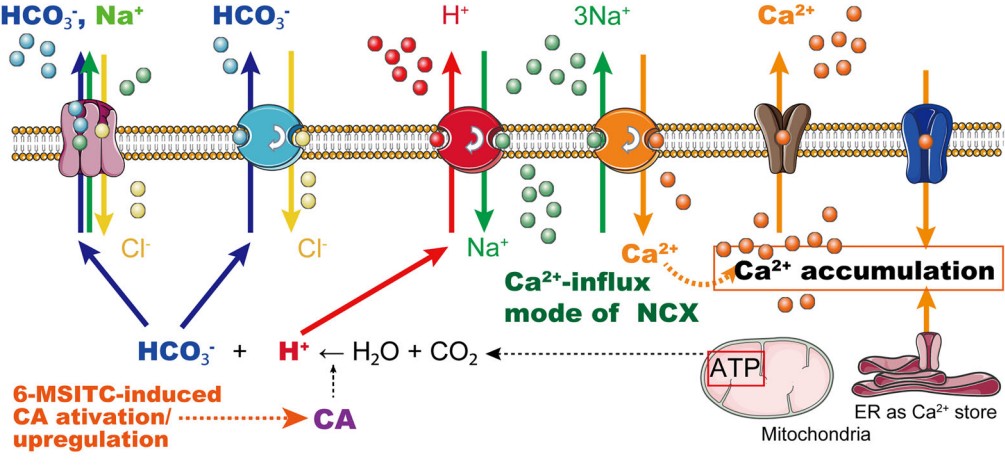

**B: 6-MSITC-induced dentin regeneration**

6-MSITC upregulates CA and enhancement of HCO₃⁻ extrusion, which results in [pH]ₒ-elevation and activation of NHE activity.

[Na⁺]ₒ being low by NHE activity reverses NCX into Ca²⁺-inflflux mode, and accumulated [Ca²⁺]ᵢ is extruded via PMCA to regenerate dentin under high [pH]ₒ and [Ca²⁺]ₒ environment

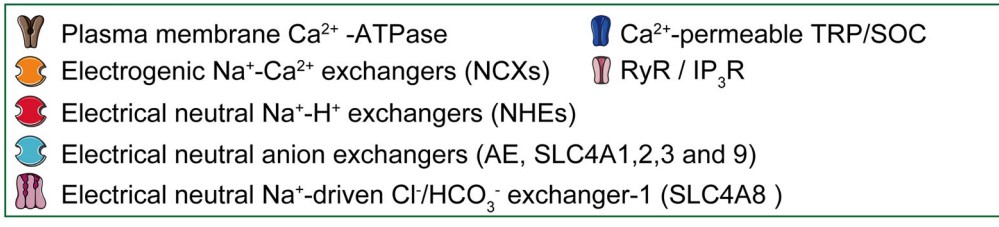

Plasma membrane Ca²⁺ -ATPase    Ca²⁺-permeable TRP/SOC
Electrogenic Na⁺-Ca²⁺ exchangers (NCXs)    RyR / IP₃R
Electrical neutral Na⁺-H⁺ exchangers (NHEs)
Electrical neutral anion exchangers (AE, SLC4A1,2,3 and 9)
Electrical neutral Na⁺-driven Cl⁻/HCO₃⁻ exchanger-1 (SLC4A8 )

**Figure 16. The wasabi ingredient, wasabi sulfinyl 6-methylsulfinylhexyl isothiocyanate, activates carbonic anhydrase-dependent transmembrane $HCO_3{}^-$-$H^+$-$Na^+$-$Ca^{2+}$ transport via SLC4As–NHE–NCX–PMCA axis**

*A*, during physiological dentin formation, $Ca^{2+}$ signalling is mediated by two closely related components: $Ca^{2+}$ influx from the extracellular medium and $Ca^{2+}$ release from intracellular $Ca^{2+}$ stores. $Ca^{2+}$ influx is caused by the activation of transient receptor potential (TRP) channels, Piezo channels and/or store-operated $Ca^{2+}$ channels, while $Ca^{2+}$ release from intracellular $Ca^{2+}$ stores is mediated by inositol-1,4,5-trisphosphate receptors or ryanodine receptors in response to G-protein coupled receptor activation or depolarisation (see Introduction). This mobilised intracellular $Ca^{2+}$ is extruded to the extracellular medium via the subtypes of NCX and $Ca^{2+}$-ATPase (PMCA) in

the distal end of plasma membrane in odontoblasts (i.e. dentin-mineralising front) to drive dentinogenesis (see also Kimura et al., 2016, 2018, 2021). Note that the extracellular concentration of $Ca^{2+}$ is 2–3 times higher in the predentin region than in the dental pulp, while $Na^+$ concentrations are 3–5 times lower in the dentinal fluid at the dentin-forming site than in serum due to the $Na^+$ incorporation mechanism that couples with $Ca^{2+}$ extrusion via $Na^+$–$Ca^{2+}$ exchange in odontoblasts. This system is caused by $Ca^{2+}$ extrusion via NCX using the transmembrane $Na^+$ gradient, resulting in a lower concentration of $Na^+$ at the mineralising front of the predentin region (see also Tsumura et al., 2010). Our preliminary results showed that the NCX inhibitor KB-R7943 and the carbonic anhydrase inhibitor dorzolamide both inhibited mineralisation, suggesting that the $Ca^{2+}$-extrusion activity of NCX and carbonic anhydrase were involved in physiological dentin formation. Together with our present findings, these results suggest that DIDS-sensitive SLC4A transporters (SLC4A1, SLC4A2, SLC4A3, SLC4A8 and SLC4A9), along with $Ca^{2+}$-ATPase, NCX and carbonic anhydrase activity, also contribute to physiological dentin formation. *B*, 6-MSITC induced enhancement of reactionary dentinogenesis, as dentin regeneration. 6-MSITC applied to the dentin surface penetrates to the odontoblasts, reaches and passes through the cell membrane, and acts on the intracellular signalling pathway. The 6-MSITC activates intracellular CA to produce and transport $H^+$ and $HCO_3^-$ extracellularly. Furthermore, SLC4As (SLC4A1, SLC4A2, SLC4A3, SLC4A8 and SLC4A9 but not SLC4A4; see Discussion) play an important role in excluding $HCO_3^-$ in an electrically neutral way. This $HCO_3^-$ transport results in extracellular pH elevation. NHE excludes $H^+$ by incorporating $Na^+$ into the cell resulting in further decreases in extracellular $Na^+$ concentration (see above); it reverses the transmembrane $Na^+$ gradient to drive $Na^+$-$Ca^{2+}$ exchange. In this situation, NCX operates in the $Ca^{2+}$ influx mode (reverse mode of NCX activity). Subsequently, intracellularly accumulated $Ca^{2+}$ through NCX activity together with $Ca^2$ from the $Ca^{2+}$ mobilisation pathway is extruded by PMCA into the extracellular space, as a dentin mineralisation front. Extruded $Ca^{2+}$ easily precipitates at the dentin mineralising front, where there is an extracellularly high pH environment, to produce reactionary/regenerative dentin.

incorporating $Na^+$ into the cell, reversing the transmembrane $Na^+$ gradient and driving the $Ca^{2+}$ influx mode (reverse mode) of NCX activity. Subsequently, intracellularly accumulated $Ca^{2+}$ is extruded into the extracellular space by PMCA. Note that the PMCA couples $Ca^{2+}$ efflux to $H^+$ uptake (Thomas, 2009; 2011). The $H^+$ incorporated into the intracellular medium via PMCA activity may subsequently be extruded by the NHE, which further reverses the transmembrane $Na^+$ gradient. This reversal can enhance $Ca^{2+}$ influx through NCX activity and promote $Ca^{2+}$ extrusion via the PMCA. Thus, SLC4As–NHE–NCX–PMCA signal coupling mediates 6-MSITC-induced dentin mineralisation and formation.

Two weeks after cavity preparation of the lower first molars in rats, we observed a significant increase in dentin formation, including collagen formation and $Ca^{2+}$ deposition, with 6-MSITC. Reactionary dentin formation is promoted by the activation of various ion channels by various stimuli applied to the dentin surface, such as mechanical, thermal, pH-related, osmotic and chemical stimuli (Goldberg & Smith, 2004; Kimura et al., 2016, 2018; Tsumura et al., 2013). These stimuli also elicited dentinal pain (as an odontoblast mechano-sensory/hydrodynamic receptor model); however, we did not evaluate pain in rats during 6-MSITC application. Intraplantar injection of high concentrations (10–30 mM) of 6-MSITC into the hind paw causes licking or biting behaviours in mice (Uchida et al., 2012). 6-MSITC also produces a few pungent sensations in humans (Masuda, 2008; Uchida et al., 2012). Further studies are required to determine whether application of 6-MSITC to the dentin surface induces dentinal pain *in vivo*.

In conclusion, wasabi sulfinyl derivative 6-MSITC exhibited strong potency in enhancing reactionary dentin formation through the activation of transmembrane $HCO_3^-$/$H^+$/$Na^+$/$Ca^{2+}$ transport via the SLC4As–NHE–NCX–PMCA axis, which promotes an extracellular environment with a high pH and high $Ca^{2+}$ concentration. Therefore, 6-MSITC is a novel and potent pharmacological candidate as a dentin regenerative drug. Our immediate research focus was to assess the effectiveness of 6-MSITC on human dentin regeneration. Clinical studies on the utilisation of 6-MSITC are currently being conducted.

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

## Additional information

### Data availability statement

The original contributions of this study are included in the article. Further enquiries can be directed to the corresponding author.

### Competing interests

The authors declare that this study received funding and reagents from Kinjirushi Co. Ltd, Aichi (Japan). The funder was not involved in the study design; collection, analysis, interpretation of data; writing of this article; or the decision to submit it for publication. The 6-MSITC patent is pending in Japan, the United States, the People's Republic of China, and the Republic of China.

### Author contributions

Conception or design of the work, Y.F., M.K., I.O., T.O., R.K., T.K-Y., H.K., M.S., M.F. and Y.S.; Acquisition, analysis or interpretation of data for the work, Y.F., M.K., I.O., T.O., R.K., T.K-Y., H.K., M.S., M.F. and Y.S.; Drafting the work or revising it critically for important intellectual content, Y.F., M.K., I.O., T.O., R.K., T.K-Y., H.K., M.S., M.F. and Y.S.; Final approval of the version to be published, Y.F., M.K., I.O., T.O., R.K., T.K-Y., H.K., M.S., M.F. and Y.S.; Agreement to be accountable for all aspects of the work, Y.F., M.K., I.O., T.O., R.K., T.K-Y., H.K., M.S., M.F. and Y.S. All authors have read and approved the final version of the manuscript and agreed to be accountable for all aspects of the work in ensuring that questions related to the accuracy or integrity of any part of the work are appropriately investigated and resolved. All persons designated as authors qualify for authorship, and all those who qualify for authorship are listed.

### Funding

This research was supported by JSPS KAKENHI Grant Number 19K10117/19H03833/22K09972/22K17025/24K12953, and a Private University Research Branding Project from MEXT of Japan (Multidisciplinary Research Centre for Jaw Disease (MRCJD), Achieving Longevity and Sustainability by Comprehensive Reconstruction of Oral and Maxillofacial Functions). This research was also supported by a 'Tokyo Dental College Research Grant (Well-Being Project)'.

### Acknowledgements

The selected artwork (cell membranes, mitochondria, and transporters) shown in the figure was adapted from pictures provided by Servier Medical Art (Servier; https://smart.servier.com/), licenced under Creative Commons Attribution 4.0 Unported License.

### Keywords

dentin, odontoblasts, regeneration, tooth, wasabi sulfinyls

## Supporting information

Additional supporting information can be found online in the Supporting Information section at the end of the HTML view of the article. Supporting information files available:

**Peer Review History**

## Translational perspective

We demonstrated that one of the main compound of wasabi (*Eutrema japonicum*), 6-methylsulfinylhexyl iso-thiocyanate (6-MSITC), has a strong ability to promote reactionary dentin formation by activating trans-membrane $HCO_3^-/H^+/Na^+/Ca^{2+}$ transport mediated through intracellular carbonic anhydrase activity. We have initiated clinical studies on the use of 6-MSITC as a dentin-regenerative therapy.

