## [Peer Review History · The Journal of Physiology]

6-Methylsulfinylhexyl isothiocyanate activates carbonic anhydrase-dependent $\text{HCO}_3^-/\text{H}^+/\text{Na}^+/\text{Ca}^{2+}$ transport via SLC4As-NHE-NCX-PMCA axis in odontoblasts

Yoshiaki Furusawa, Maki Kimura, Isao Okunishi, Takehito Ouchi, Ryuya Kurashima, Tomoe Katou-Yamada, Hidetaka Kuroda, Makoto Sugita, Masahiro Furusawa, and Yoshiyuki Shibukawa
DOI: 10.1113/JP287809

Corresponding author(s): Yoshiyuki Shibukawa (yshibuka@tdc.ac.jp)

The following individual(s) involved in review of this submission have agreed to reveal their identity: Jason I.E. Bruce (Referee #2)

Review Timeline:

Submission Date:	04-Oct-2024
Editorial Decision:	14-Nov-2024
Revision Received:	26-Nov-2025
Editorial Decision:	06-Jan-2026
Revision Received:	22-Jan-2026
Accepted:	28-Jan-2026

Senior Editor: Kim Barrett

Reviewing Editor: Pawel Ferdek

Transaction Report:

Dear Dr Shibukawa,

Re: JP-RP-2024-287809 "6-Methylsulfinylhexyl isothiocyanate activates carbonic anhydrase-dependent HCO₃⁻/H⁺/Na⁺/Ca²⁺ transport via SLC4As-NHE-NCX-PMCA axis in odontoblasts" by Yoshiaki Furusawa, Maki Kimura, Isao Okunishi, Takehito Ouchi, Ryuya Kurashima, Tomoe Katou-Yamada, Hidetaka Kuroda, Makoto Sugita, Masahiro Furusawa, and Yoshiyuki Shibukawa

Thank you for submitting your manuscript to The Journal of Physiology. It has been assessed by a Reviewing Editor and by 2 expert referees and we are pleased to tell you that it is acceptable for publication following satisfactory revision.

REVISION CHECKLIST:

We look forward to receiving your revised submission.

Yours sincerely,

Kim Barrett
Senior Editor
The Journal of Physiology

EDITOR COMMENTS

Reviewing Editor:

Comments to the Author:

Thank you for submitting your work to The Journal of Physiology. Both Referees find the manuscript potentially interesting and impactful within the field. However, they have noted a number of areas that require further clarification to strengthen the paper's conclusions and improve overall clarity. Please also ensure that the origin of the animals used in this study is clearly stated.

REFEREE COMMENTS

Referee #1:

The paper of Furusawa and co-authors "6-Methylsulfinylhexyl isothiocyanate activates carbonic anhydrase-dependent HCO₃⁻/H⁺/Na⁺/Ca²⁺ transport via SLC4As-NHE-NCX-PMCA axis in odontoblasts" is an ambitious effort of putting together the transmembrane transport in the odontoblasts and dentinogenesis, dentin regeneration and remineralisation processes. The bright breakthroughs of this impressive research, although are having some major and minor downsides consideration of which is going to significantly strengthen this conceptual study.

Major points.

- 1) P25 (390-391) It would be great if the authors provided IC₅₀ value for AITC (TRPA1 channel agonist) on alizarin red staining.
- 2) P25-26 (397-398) "...6-MSITC and 6-MSFITC significantly increased the mineralisation efficiency in both alizarin red and (Figure 4A) and von Kossa staining (Figure 4B)." How much significantly? It is so much inconvenient to go to the figures to see P-values, please provide the values in the form of mean{plus minus}SD or SEM and give n-numbers. It would be great if the authors explained why they used 50 µM for both compounds, have they, or other groups, done the concentration-response for both 6-MSITC and 6-MSFITC in the previous studies? If yes, please cite these papers.
- 3) P26 (408 and below) "DZ, a CA inhibitor, significantly and dose-dependently decreased the 6-MSITC-induced mineralisation efficiency in odontoblasts, as determined by alizarin red staining (Figures 6A and 6C) and von Kossa staining (Figures 6B and 6C)." Would it be possible for the authors to specify the level of significance, i.e. mean{plus minus}SD or SEM and give n-numbers and the P-values? Please have these values in the text body, not just in the figures. Also, here, above (line 403-404) and in the rest of the text please change "dose-dependent", which refers to the in vivo studies, for "concentration-dependent", which refers to the in vitro experiments.
- 4) P26-27 (415-417) "...mineralisation was significantly reduced by the simultaneous application of caloxin 1b1 (100 µM) or DZ (1 mM) with alizarin red (Figures 7A and 7C) and von Kossa staining (Figures 7B and 7D)." The chosen concentrations (100 µM and 1 mM) were rather high and might have nonspecific effect on other pathways. It would be great if the authors could verify why they used these huge concentrations for these compounds.
- 5) P27 (423-426) "The application of 500 µM DIDS (solute carrier family 4 [SLC4A]423 transporter/Cl⁻ channel inhibitor) completely inhibited mineralisation efficiency in 424 odontoblasts with mineralisation medium with alizarin red (Figures 8A

and 8C) and von Kossa staining (Figures 8B and 8D)." 500 μ M of DIDS is another industrial concentration, which could have nonspecific effects. DIDS inhibits Cl⁻ conductance with IC₅₀ ~ 150 nM, what is a point of using it in 500 μ M. Also, it is not clear for how long was DIDS administrated on the cells?

6) P27-28 (431 and below) Immunofluorescence Analyses. It seems like the authors are having too much trust in the antibodies they used for the staining to assess expression of SLC4A subtypes. Such confidence should be supported with RNA-seq or qPCR experiments, also, employment of additional antibodies is highly recommended. For the posttranslational quantification of the staining it would be great if the authors employed Westerns or at least FIJI-program semi-quantification.

7) P28 (446 and below) Gramicidin-Perforated Patch-Clamp Recording. It would be less confusing if the authors represent the transmembrane currents in the text simply as current densities (pA/pF), same as they did that in the Figures 12-14 without %. Also, it would be good if the n of experiments is provided in the text as well. Overall, and taking into account the composition of bath and pipette solutions the authors used, the macroscopic transmembrane current seems to be of voltage-gated K⁺ outward nature, however the authors did not verify the nature of the current pharmacologically by using TEA, 4-AP or other generic K⁺ current inhibitors applied from in the bath, or by using Cs⁺ in the pipette solution. Since there are no experimental evidences of the transmembrane current nature, the authors should not use term "outward and inward", but rather use terms "positive and negative". This is because if the significant component of the transmembrane current is anion (Cl⁻) not cation (K⁺ or Na⁺/Ca²⁺), what looks as outward could be inward and vice versa.

8) P30 (470) "The application of 10 mM DMA..." again, this is gigantic concentration, the specificity of its effect is highly doubtful. It would be helpful to know whether the authors equimolarly substituted 10mM of NaCl? If not, the solution was hypertonic and might cause shrinkage of the cells, with consequent nonspecific effects.

9) P30 (480-481) we applied 6-MSITC to dentin cavities prepared in rats. 6-MSITC significantly increased reactionary dentin formation..." It would be beneficial if the authors provided mean{plus minus}SD or SEM, give n-numbers and the P-values. Also, it should be specified in which form and concentration 6-MSITC was applied and for how long.

10) P35 (551 and below) "Mutations in the SLC4A2, SLC4A4, and SLC4A9 transporters induce enamel anomalies because these proteins play important roles in pH regulation during enamel development (Lyaruu et al., 2008; Lacruz et al., 2010; Jalali et al., 2014; Reibring et al., 2014; Yin & Paine, 2017). Members of the SLC4A family of odontoblasts are likely involved in regulating extracellular pH by mediating the exclusion of HCO₃⁻ during dentin mineralisation and/or formation." This must be ameloblast story, not odontoblast, it would be good if the authors clarified that in the text.

Minor points

The Latin must be Italic in the text:

P13 (175) In Vivo

P13 (184 respectively). ad libitum

P26 "...and dose-dependent increase in the pH of the medium (Figure 5)" please correct for "concentration-dependent". The same in "DZ, a CA inhibitor, significantly and dose-dependently decreased the 6-MSITC", change for "concentration-dependent".

P30 (477) "In vivo"

P33 "... (i.e. CA activation) and its downstream ionic transporters (i.e.)"

P38 (608) - "... (i.e. the current those caused by electrogenic transporters, electroneutral HCO₃⁻ transport (i.e.)"

P40 (641) "dentin-forming/mineralizing front (i.e. to the predentin region).."

P41 (658) "In summary, for the in vitro experiments (Figure 16)..."

P40 (634) "NCX are bidirectional transporters" normally they are called antiporters.

Figure 3: Why in Fig 3 P= and in Fig 4 P

Figure 5: why P= and P< in the same figure? Please make uniform and it would be good if the authors plot it as a concentration-response to figure out EC₅₀

Figure 5: why P= and P< in the same figure? Did the authors ever apply DZ independently of 50 μ M 6-MSITC? Why the authors used here only one concentration (50 μ M) of 6-MSITC?

Figure 7: in C why $P <$ and $P =$, why is that? Please make uniform. Why not to test 10 μM KB-R7943, 100 μM caloxin 1b1, and 1 mM dorzolamide (DZ), on their own without 50 μM 6-MSITC?

Figure 8: Why the effects of 6-MSITC and DIDS are so much heterogenous in alizarin red in A and von Kossa staining in B?

Figure 9: Not quite convincing immunostaining to support the idea on the expression. Did the authors try RNA-seq or q-PCR?

Referee #2:

This manuscript provides a very detailed mechanistic description for mineralisation of odontoblasts in response to the active ingredients of wasabi sulfinyls (6-MSITS). Based on the data, this seems to involve a functional coupling between carbonic anhydrase (CA), HCO_3^- -transporters (SLC4A), Na-H exchangers (NHE), plasma membrane Ca ATPase (PMCA)/Na-Ca exchange (NCX) and the TRPA1.

Overall, this is an excellent manuscript that is very impactful for the oral biology/dentistry field, and I believe will be extensively cited. The manuscript is very well written, with nicely presented data that are extensively and appropriately analyzed. All data seem to be internally consistent with the authors hypotheses. However, I have a few minor corrections/suggestions for improvement:

-Could the authors explain whether any increase/decrease in mineralization (Alizarin red or von Kossa stain) observed with any of the drugs might be influenced by effects on cell proliferation (increase) or cell death (decrease). Could these be confounding factors or do the authors have any additional data or could cite previous papers validating this technique for assessing mineralization? Should/could the Alizarin/von Kossa stain be normalized to live cell number?

-What is the mechanism for the biphasic response of 6-MSITC on mineralization? In the Discussion, the authors suggest that the effect of 6-MSITC-induced mineralization is independent of TRPA1 activation. However, the initial 6-MSITC-induced decrease in mineralization may be due to activation of TRPA1, but the secondary increase may involve some other mechanism. What might that be? Could the authors speculate as to the potential bimodal effect on mineralization (i.e. both the decrease and increase)?

-Related to the above point, did any of the other analogues of 6-MSITC have a similar bimodal effect or were lower concentrations not tested? Could the "lack of potency" of some of these compounds be due to a rightward shift in their bimodal response on mineralization?

-Its also worth noting in the Discussion that extracellular acidification will enhance and alkalization will inhibit PMCA activity because PMCA is in fact an electroneutral ATP-driven $\text{H}^+/\text{Ca}^{2+}$ -exchanger.

-Why is there clustering of cells (or stain) in the center of the wells in Figure 6?

-There is no control in Figure 6. Is this because the mean value for 6-MSITC (approx. 1.5) is very close to the control value (1.3, instead of 2.3 in Fig 4)? If this is the case, then why. Moreover, in this experiments the CA inhibitor (dorzolamide) appears to reduce the values to less than the control values (without 6-MSITC). Why is this?

-MTA not defined on page 11

END OF COMMENTS

Response to Reviewers

MS ID#: JP-RP-2024-287809R1

MS Title: 6-Methylsulfinylhexyl isothiocyanate activates carbonic
anhydrase-dependent $\text{HCO}_3^-/\text{H}^+/\text{Na}^+/\text{Ca}^{2+}$ transport via
SLC4As-NHE-NCX-PMCA axis in odontoblasts

Format: Research Article

Authors: Yoshiaki Furusawa, Maki Kimura, Isao Okunishi,
Takehito Ouchi, Ryuya Kurashima, Tomoe Katou-Yamada, Hidetaka
Kuroda, Makoto Sugita, Masahiro Furusawa, Yoshiyuki Shibukawa

Response to Reviewing Editor

Thank you for submitting your work to The Journal of Physiology. Both Referees find the manuscript potentially interesting and impactful within the field. However, they have noted a number of areas that require further clarification to strengthen the paper's conclusions and improve overall clarity. Please also ensure that the origin of the animals used in this study is clearly stated.

Response:

We would like to sincerely thank the Reviewer for the valuable comments. We greatly appreciate the feedback and apologize for the delay in resubmitting the revised manuscript, as additional results were required to address the Reviewers' concerns.

Furthermore, during the revision process, the corresponding author (YS) underwent surgery for stomach cancer, was hospitalized, and is currently undergoing chemotherapy. We deeply apologize once again for the delay in resubmitting the revised manuscript.

Response to Reviewer #1

The paper of Furusawa and co-authors "6-Methylsulfinylhexyl isothiocyanate activates carbonic anhydrase-dependent HCO₃⁻/H⁺/Na⁺/Ca²⁺ transport via SLC4As-NHE-NCX-PMCA axis in odontoblasts" is an ambitious effort of putting together the transmembrane transport in the odontoblasts and dentinogenesis, dentin regeneration and remineralisation processes. The bright breakthroughs of this impressive research, although are having some major and minor downsides consideration of which is going to significantly strengthen this conceptual study.

Response:

We would like to sincerely thank the Reviewer for the valuable comments. We greatly appreciate the feedback and apologize for the delay in resubmitting the revised manuscript, as additional results were required to address the Reviewers' concerns.

Furthermore, during the revision process, the corresponding author (YS) underwent surgery for stomach cancer, was hospitalized for several weeks, and is currently undergoing chemotherapy. We deeply apologize once again for the delay in resubmitting the revised manuscript.

1) P25 (390-391) It would be great if the authors provided IC₅₀ value for AITC (TRPA1 channel agonist) on alizarin red staining.

Response: Thank you for your suggestion. The half-maximal inhibitory concentration (IC₅₀) of AITC for mineralization efficiency in alizarin red staining was 0.06 μM (Figure 3). The effects of AITC on the mineralising efficiencies were evaluated by fitting the data according to the function:

$$A = (A_{max} - A_{min}) / (1 + ([x]_o / K_D)^p) + A_{min} \quad (1)$$

where K_D was the half maximal (50%) inhibitory concentration (IC₅₀) of the AITC (0.06 μM), [x]_o indicates concentrations of extracellular AITC applied (0.1 μM to 100 μM), and A_{max} and A_{min} are the maximal and minimal I/I_o as mineralising efficiencies (see Materials and Methods), respectively. The Hill coefficient (p) was 2.0.

Accordingly, we have added following sentences to the revised manuscript.

Results section (Page 27; Lines 424-427): "AITC, a TRPA1 channel agonist, significantly decreased mineralisation efficiency of alizarin red staining (Figures 3A and 3B), showing a half-maximal inhibitory concentration (IC₅₀) of 0.06 μM in concentrations ranging from 0.1 μM to 100 μM.

"

Figure legend for Figure 3 (From page 62, line 1067 to page 63, line 1074): “The dependence of changes in mineralisation level (I/I_0) on the concentrations of the extracellular AITC applied was obtained by fitting the data to the following function:

$$A = (A_{max} - A_{min}) / (1 + ([x]_0 / K_D)^p) + A_{min} \quad \text{equation 1}$$

where K_D was the half maximal (50%) inhibitory concentration (IC_{50}) of the AITC (0.06 μ M), $[x]_0$ indicates concentrations of extracellular AITC applied (0.1 μ M to 100 μ M), and A_{max} and A_{min} are maximal and minimal I/I_0 as mineralising levels, respectively. Hill coefficient (p) was 2.0.”

2) P25-26 (397-398) "...6-MSITC and 6- MSFITC significantly increased the mineralisation efficiency in both alizarin red and (Figure 4A) and von Kossa staining (Figure 4B). How much significantly? It is so much inconvenient to go to the figures to see P-values, please provide the values in the form of mean{plus minus}SD or SEM and give n-numbers. It would be great if the authors explained why they used 50 μ M for both compounds, have they, or other groups, done the concentration-response for both 6-MSITC and 6- MSFITC in the previous studies? If yes, please cite these papers.

Response: According to the comment, we have revised the relevant sentence, by adding the mean values of mineralisation efficiency as I/I_0 units along with the SD values and n as the number of independent experiments tested, as follows:

Page 28; Lines 432-440: "At a concentration of 50 μ M, we observed significant increases in the mineralisation efficiency of both alizarin red (Figure 4A) and von Kossa staining (Figure 4B) by 6-MSITC ($2.41 \pm 0.64 I/I_0$ unit, $n = 6$, $P = 0.000246$ with alizarin red and $1.35 \pm 0.09 I/I_0$ unit, $n = 6$, $P = 0.000336$ with von Kossa staining) and 6-MSFITC ($2.02 \pm 0.214 I/I_0$ unit, $n = 6$, $P = 0.000449$ with alizarin red and $1.28 \pm 0.101 I/I_0$ unit, $n = 6$, $P = 0.000336$ with von Kossa staining), compared with that observed in the absence of 6-MSITC and 6-MSFITC (as control; $1.28 \pm 0.25 I/I_0$ unit, $n = 40$, with alizarin red and $1.07 \pm 0.05 I/I_0$ unit, $n = 40$, with von Kossa staining)."

As we addressed in our response to comment #15 raised by this reviewer, the half-maximal effective concentration (EC_{50}) of 6-MSITC for changes in medium pH was 29.271 μ M (see Figure 5). This value was obtained by fitting the data to Equation 1 (as explained in our response to comment #1). Therefore, in Figure 4, we compared the effects of 6-MSITC and its derivatives at a concentration of 50 μ M.

In addition, as we have described in the text, for both alizarin red (Figures 2A and 2C) and von Kossa (Figures 2B and 2D) staining, 6-MSITC significantly increased mineralisation efficiency at concentrations of 40 μ M and 100 μ M, and no significant differences were observed in the staining efficiency between 40 μ M and 100 μ M 6-MSITC. In our data (in preparation for

publication in another journal), for 6-MSFITC, we observed a significant increase in the mineralisation efficiency with the application of the concentrations of 50 μM (2.02 ± 0.214 , $n = 6$, $P = 0.0056$ with alizarin red and 1.28 ± 0.101 , $n = 6$, $P < 0.0001$ with von Kossa staining, see also above) and 500 μM (2.45 ± 0.340 , $n = 7$, $P = 0.0003$ with alizarin red and 1.44 ± 0.0495 , $n = 7$, $P < 0.0001$ with von Kossa staining), compared to the values obtained without 6-MSFITC. However, we did NOT observe any significant increase in mineralisation efficiency at a concentration of 5 μM 6-MSFITC (1.157 ± 0.0788 , $n = 11$, $P > 0.999$ with alizarin red and 1.044 ± 0.0278 , $n = 7$, $P = 0.9927$ with von Kossa staining), compared with that obtained in absence of 6-MSFITC. Thus, we compared the effects of 6-MSITC and its derivatives at a concentration of 50 μM in Figure 4. The authors would greatly appreciate it if the Reviewer takes these points into consideration.

Accordingly, we have added following sentence to the Figure Legend of Figure 4: "Note that, for 6-MSFITC, a significant increase in mineralisation efficiency was observed at concentrations of 50 μM compared with that without 6-MSFITC, but not at that of 5 μM (1.157 ± 0.0788 , $n = 11$, $P > 0.999$ with alizarin red and 1.044 ± 0.0278 , $n = 7$, $P = 0.9927$ with von Kossa staining; not shown)."

To our knowledge, we found one report describing the biological effects of 6-MSFITC on chemical mediator release from rat basophilic leukaemia cells, where 50 μM of 6-MSITC were used. We have added this report as a reference.

"Yamada-Kato T, Nagai M, Ohnishi M, Yoshida K. Inhibitory effects of wasabi isothiocyanates on chemical mediator release in RBL-2H3 rat basophilic leukemia cells. J Nutr Sci Vitaminol (Tokyo). 58: 303-307, 2012. doi: 10.3177/jnsv.58.303. PMID: 23132316."

3) P26 (408 and below) "DZ, a CA inhibitor, significantly and dose-dependently decreased the 6-MSITC-induced mineralisation efficiency in odontoblasts, as determined by alizarin red staining (Figures 6A and 6C) and von Kossa staining (Figures 6B and 6C)." Would it be possible for the authors to specify the level of significance, i.e. mean{plus minus}SD or SEM and give n-numbers and the P-values? Please have these values in the text body, not just in the figures. Also, here, above (line 403-404) and in the rest of the text please change "dose-dependent", which refers to the in vivo studies, for "concentration-dependent", which refers to the in vitro experiments.

Response: According to your comment, we have revised the relevant sentence by adding the mean values of mineralisation efficiency as I/I_0 units along with SD values and n as the number of independent experiments tested, as indicated below. Additionally, we have replaced the term "dose-dependent" with "concentration-dependent".

Page 29; Lines 451-459: "DZ, a CA inhibitor, significantly and concentration-dependently decreased the 6-MSITC-induced mineralisation efficiency in odontoblasts, as determined by alizarin red staining (1.45 ± 0.04 I/I_0 unit by 0.5 mM DZ; 1.35 ± 0.07 I/I_0 unit, $P = 0.000283$ by 1 mM DZ; 1.12 ± 0.04 I/I_0 unit, $P < 0.0001$ by 10 mM DZ; $n = 7$ each; Figures 6A and 6C) and von Kossa staining (1.45 ± 0.05 I/I_0 unit by 0.5 mM DZ; 1.34 ± 0.06 I/I_0 unit, $P = 0.00161$ by 1 mM DZ; 1.10 ± 0.05 I/I_0 unit, $P < 0.0001$ by 10 mM DZ; $n = 7$ each; Figures 6B and 6C), compared with those of the control values without DZ (1.51 ± 0.08 for alizarin red and 1.49 ± 0.06 for von Kossa staining, $n = 7$ each)."

4) P26-27 (415-417) "...mineralisation was significantly reduced by the simultaneous application of caloxin 1b1 (100 μ M) or DZ (1 mM) with alizarin red (Figures 7A and 7C) and von Kossa staining (Figures 7B and 7D)." The chosen concentrations (100 μ M and 1 mM) were rather high and might have nonspecific effect on other pathways. It would be great if the authors could verify why they used these huge concentrations for these compounds.

Response: As we have already described in the manuscript (lines 225-227; page 15), the concentrations of caloxin 1b1 (100 μ M) used in this study were determined by Pande *et al.*, 2006, 2008, Chen *et al.*, 2014, and Groten *et al.*, 2016, as well as by our previous study (Kimura *et al.*, 2021). In our previous study, the Ca^{2+} extrusion efficiency during the hypotonic or alkaline solution-induced intracellular free Ca^{2+} concentration ($[Ca^{2+}]_i$) increase was decreased by 100 μ M caloxin 1b1 in human odontoblasts. Alizarin red and von Kossa staining showed that PMCA inhibition by 100 μ M caloxin 1b1 suppressed mineralisation. Thus, we determined the concentration of caloxin 1b1 (100 μ M) used in this study.

With regard to the concentration of carbonic anhydrase inhibitor, dorzolamide (DZ), Dong *et al.* (2016) used the 3 mM concentration to examine changes in ciliary artery isometric tension and intracellular free Ca^{2+} concentration, whereas Adijanto *et al.* (2009) determined that the most appropriate concentration was 0.25 mM for experiments relative to intracellular pH measurement and transepithelial potential in the retinal pigment epithelium. Furthermore, as Boyne *et al.* (2022) used a concentration of 0.015 mM in experiments investigating cellular cholesterol accumulation. Although there are variations of DZ concentration in previous studies, as described above (from 0.015 mM to 3 mM), the concentrations for DZ in this study were determined as 1 mM based on these previous reports.

Accordingly, we have added following sentence to the revised manuscript:

Page 15; Lines 217-219: "The concentration of DZ was determined according to that described in previous studies (Dong *et al.*, 2016; Adijanto *et al.*, 2009; Boyne *et al.*, 2022)."

We have also added the following additional references to the References section:

1) Dong Y, Sawada Y, Cui J, Hayakawa M, Ogino D, Ishikawa M & Yoshitomi T (2016).

Dorzolamide-induced relaxation of isolated rabbit ciliary arteries mediated by inhibition of extracellular calcium influx. *Jpn J Ophthalmol* **60**, 103–110.

2) Adijanto J, Banzon T, Jalickee S, Wang NS, & Miller SS (2009). CO₂-induced ion and fluid transport in human retinal pigment epithelium. *J Gen Physiol* **133**, 603–622.

3) Boyne K, Corey DA, Zhao P, Lu B, Boron WF, Moss FJ & Kelley TJ (2022). Carbonic anhydrase and soluble adenylylase regulation of cystic fibrosis cellular phenotypes. *Am J Physiol Lung Cell Mol Physiol* **322**: L333–L347.

5) P27 (423-426) "The application of 500 µM DIDS (solute carrier family 4 [SLC4A]423 transporter/Cl⁻ channel inhibitor) completely inhibited mineralisation efficiency in 424 odontoblasts with mineralisation medium with alizarin red (Figures 8A and 8C) and von Kossa staining (Figures 8B and 8D)." 500 µM of DIDS is another industrial concentration, which could have nonspecific effects. DIDS inhibits Cl⁻ conductance with IC₅₀ ~ 150 nM, what is a point of using it in 500 µM. Also, it is not clear for how long was DIDS administrated on the cells?

Response: Thank you for your comment, and we sincerely apologize for the oversight. The concentration of DIDS used in this study was 100 µM. We have revised the manuscript throughout, accordingly, as well as Figures 8 and 13, by replacing “500 µM DIDS” with “100 µM DIDS”. We believe that the concentration of 100 µM DIDS is suitable to inhibit anion channels/transporters, based on previous reports.

6) P27-28 (431 and below) Immunofluorescence Analyses. It seems like the authors are having too much trust in the antibodies they used for the staining to assess expression of SLC4A subtypes. Such confidence should be supported with RNA-seq or qPCR experiments, also, employment of additional antibodies is highly recommended. For the posttranslational quantification of the staining it would be great if the authors employed Westerns or at least FIJI-program semi-quantification.

Response: In response to the reviewer’s comments, we have re-analysed the immunofluorescence data (revised Figures 9 and 10) and added new results showing mRNA expression level of SLC4A family members and carbonic anhydrase (CA) using real-time RT-PCR (additional Figure 16). Our analysis revealed that SLC4A1, SLC4A2, SLC4A3, SLC4A4, SLC4A8, and SLC4A9 are constitutively expressed in human odontoblasts (Figure 9). We also observed constitutive immunoreactivity for CA I and CA II (Figure 10). Furthermore, mRNA expression of SLC4A1, SLC4A2, SLC4A3, SLC4A4, SLC4A8, and SLC4A9, as well as CA I and CA II, was detected in human odontoblasts under control conditions and following 12- and 24-h treatment with 100 µM 6-MSITC (Figure 16). Notably, expression of SLC4A1, SLC4A2, and CA II mRNA was significantly upregulated after 12 h of 6-MSITC treatment.

Accordingly, we have revised the manuscript and updated the figures throughout to

incorporate the newly obtained and re-analysed data.

1) Figures 9 and 10 have been revised and a new figure (Figure 16) has been added to present the real-time RT-PCR results. The figure legends for Figures 9, 10, and 16 have also been updated, accordingly.

2) A list of primer sequences used for real-time RT-PCR has been included as Table 2.

3) Page 24; Lines 378–392: We have added the experimental protocol for real-time RT-PCR in the Materials and Methods section under the new subheading “*Real-time reverse-transcription polymerase chain reaction.*”

4) From Page 30, line 480 to page 31, line 490: We have revised the relevant text in the Results section corresponding to the immunofluorescence data presented in Figures 9 and 10 as follows:

“Immunofluorescence Analyses To examine the expression patterns of plasma membrane HCO_3^- transporters and CA I and II in odontoblasts, immunofluorescence analysis was performed. We observed the expression of HCO_3^- -transporting solute carrier family 4 member 1 (SLC4A1; A), SLC4A2 (B), SLC4A3 (C), SLC4A4 (D; electrogenic $\text{Na}^+/\text{HCO}_3^-$ cotransporter-1: NBCe1), SLC4A8 (E; Na^+ -driven $\text{Cl}^-/\text{HCO}_3^-$ exchanger-1: NDCBe1), and SLC4A9 (F) in human odontoblasts (Figure 9). The results revealed that SLC4A1, SLC4A2, SLC4A3, SLC4A4, SLC4A8, and SLC4A9 were constitutively expressed in the plasma membrane of human odontoblasts. Constitutive immunoreactivities of CA I (Figure 10A) and II (Figure 10B) were also observed.”

5) From Page 34, line 552 to page 35, line 566: We have added the relevant text to the Results section corresponding to the real-time RT-PCR data presented in Figure 16 as follows:

“6-MSITC Application Modulated Expression of SLC4As, CA1 and CA2 mRNA in Human Odontoblasts Figure 16 shows the expression of mRNA encoding SLC4A1 (A), SLC4A2 (B), SLC4A3 (C), SLC4A4 (D), SLC4A8 (E), and SLC4A9 (F), as well as CA1 (G) and CA2 (H) in human odontoblast, without (control) or with 12 h (each middle) and 24 h application (each most right) of 100 μM 6-MSITC. The expression of SLC4A1 (A; $P < 0.0001$, $n = 6$) and SLC4A2 (B; $P < 0.0001$, $n = 7$), as well as CA II (H; $P < 0.0009$, $n = 6$) mRNAs was significantly upregulated by a 12 h application of 100 μM 6-MSITC, compared with levels without 6-MSITC application (control condition, $n = 18$ for each). In addition, CA I mRNA expression (G) was significantly higher than that of CA II (H) in the control (no exposure to 6-MSITC, $P < 0.0001$, $n =$

17 for CA I and 18 for CA II, not shown). Together the results from immunofluorescence analyses (Figures 9 and 10) indicate that SLC4A1, SLC4A2, SLC4A3, SLC4A4, SLC4A8, and SLC4A9, as well as CA I and CA II, were constitutively expressed in human odontoblasts."

6) We have added the relevant text to the Discussion section corresponding to the immunofluorescence and real-time RT-PCR results as follows:

From Page 38, line 621 to page 39, line 623: "SLC4A1 (anion exchanger-1: AE1), SLC4A2 (AE2), SLC4A3 (AE3), SLC4A4 (NBCe1), SLC4A8 (NDCBe1), and SLC4A9 (AE4) were constitutively expressed in human odontoblasts."

From Page 43, line 707–711: "Furthermore, after a 12-h exposure to 6-MSITC treatment in odontoblasts, the mRNA expression of SLC4A1, SLC4A2, and CA II was upregulated. Thus, during 6-MSITC exposure, the differential expression of electroneutral SLC4A family members and CA plays a role in modulating dentinogenesis by mediating the production and exclusion of extracellular HCO₃⁻."

The inhibitory effect of DZ on 6-MSITC-induced currents and mineralisation aligns with the significant constitutive expression of CA at mRNA and protein levels, suggesting that CA plays a crucial role in HCO₃⁻ production, not only during 6-MSITC application but also throughout aerobic ATP synthesis in odontoblasts."

From Page 45, line 747 to page 46, line 749: "SLC4As (SLC4A1, SLC4A2, SLC4A3, SLC4A8, and SLC4A9) play important roles in the electroneutral extrusion of HCO₃⁻."

7) P28 (446 and below) Gramicidin-Perforated Patch-Clamp Recording. It would be less confusing if the authors represent the transmembrane currents in the text simply as current densities (pA/pF), same as they did that in the Figures 12-14 without %. Also, it would be good if the n of experiments is provided in the text as well. Overall, and taking into account the composition of bath and pipette solutions the authors used, the macroscopic transmembrane current seems to be of voltage-gated K⁺ outward nature, however the authors did not verify the nature of the current pharmacologically by using TEA, 4-AP or other generic K⁺ current inhibitors applied from in the bath, or by using Cs⁺ in the pipette solution. Since there are no experimental evidences of the transmembrane current nature, the authors should not use term "outward and inward", but rather use terms "positive and negative". This is because if the significant component of the transmembrane current is anion (Cl⁻) not cation (K⁺ or Na⁺/Ca²⁺), what looks as outward could be inward and vice versa.

Response: According to the Reviewer's comment, we have revised the relevant sentence in the Figure legend for Figures 12 to 14 by adding the mean values of current density along with the respective SD values and *n* as the number of independent experiments tested as follows.

Legend for Figure 12; “In the gramicidin-perforated mode, the values of the current densities were -2.995 ± 1.963 pA/pF without and -5.716 ± 3.154 pA/pF with $5 \mu\text{M}$ 6-MSITC at -100 mV (each $n = 6$), and $+45.505 \pm 19.175$ pA/pF without and $+116.673 \pm 20.725$ pA/pF with $5 \mu\text{M}$ 6-MSITC at $+100$ mV ($P = 0.000988$, each $n = 6$). In the conventional mode, the values of the current densities were -1.402 ± 0.942 pA/pF without and -1.777 ± 1.0825 pA/pF with $5 \mu\text{M}$ 6-MSITC at -100 mV (each $n = 6$), and $+45.433 \pm 19.732$ pA/pF without and $+35.512 \pm 14.484$ pA/pF with $5 \mu\text{M}$ 6-MSITC at $+100$ mV (each $n = 6$).”

Legend for Figure 13; “In B, at the membrane potential of -100 mV, the values of current density were -1.528 ± 1.027 pA/pF under the control condition, -2.847 ± 0.881 pA/pF with $5 \mu\text{M}$ 6-MSITC, and -2.671 ± 0.687 pA/pF $5 \mu\text{M}$ 6-MSITC plus 10 mM DZ (each $n = 6$). At the membrane potential of $+100$ mV, the values of the current density were $+29.647 \pm 14.563$ pA/pF under the control condition, $+83.338 \pm 30.225$ pA/pF with $5 \mu\text{M}$ 6-MSITC ($P = 0.00164$ compared with control), and $+54.246 \pm 32.993$ pA/pF $5 \mu\text{M}$ 6-MSITC plus 10 mM DZ ($P = 0.00314$ compared with the value exposed to $5 \mu\text{M}$ 6-MSITC) (each $n = 6$). In D, at the membrane potential of -100 mV, the values of the current density were -3.732 ± 1.423 pA/pF under the control condition, -4.081 ± 2.730 pA/pF with $5 \mu\text{M}$ 6-MSITC, and -4.186 ± 2.411 pA/pF $5 \mu\text{M}$ 6-MSITC plus $100 \mu\text{M}$ DIDS (each $n = 5$). At the membrane potential of $+100$ mV, the values of the current density were $+41.6732 \pm 23.979$ pA/pF under the control condition, $+81.309 \pm 48.380$ pA/pF with $5 \mu\text{M}$ 6-MSITC ($P = 0.0282$ compared with control), and $+85.127 \pm 40.891$ pA/pF $5 \mu\text{M}$ 6-MSITC plus $100 \mu\text{M}$ DIDS (each $n = 5$). In F, at the membrane potential of -100 mV, the values of the current density were -1.092 ± 0.417 pA/pF under the control condition, -1.506 ± 2.264 pA/pF with $5 \mu\text{M}$ 6-MSITC, and -1.855 ± 1.039 pA/pF $5 \mu\text{M}$ 6-MSITC plus $20 \mu\text{M}$ DMA (each $n = 4$). At the membrane potential of $+100$ mV, the values of the current density were $+31.953 \pm 5.484$ pA/pF under the control condition, $+64.361 \pm 5.307$ pA/pF with $5 \mu\text{M}$ 6-MSITC ($P = 0.00750$ compared with control), and $+30.675 \pm 2.783$ pA/pF $5 \mu\text{M}$ 6-MSITC plus $20 \mu\text{M}$ DMA ($P = 0.00320$ compared with the value exposed to $5 \mu\text{M}$ 6-MSITC) (each $n = 4$).”

Legend for Figure 14; “In B, at the membrane potential of -100 mV, the values of the current density were -3.728 ± 1.811 pA/pF under the control condition, -3.012 ± 0.585 pA/pF with $5 \mu\text{M}$ 6-MSITC, and -1.873 ± 1.371 pA/pF $5 \mu\text{M}$ 6-MSITC plus $10 \mu\text{M}$ KB-R7943 (each $n = 5$). At the membrane potential of $+100$ mV, the values of the current density were $+102.802 \pm 26.519$ pA/pF under the control condition, $+151.458 \pm 40.679$ pA/pF with $5 \mu\text{M}$ 6-MSITC ($P = 0.00720$ compared with control), and $+93.611 \pm 25.885$ pA/pF with $5 \mu\text{M}$ 6-MSITC plus $10 \mu\text{M}$ KB-R7943 ($P = 0.00350$ compared with the value exposed to $5 \mu\text{M}$ 6-MSITC) (each $n = 5$).”

Regarding the macroscopic transmembrane current shown in Figures 12 to 14, recorded using extracellular Na^+ -rich and intracellular K^+ -rich solutions, we have previously

identified their properties, which were published in "Ichikawa et al., Voltage-dependent Sodium Channels and Calcium-activated Potassium Channels in Human Odontoblasts In Vitro, *Journal of Endodontics*, 38: 1355-1362, 2012". Human odontoblasts, which are the same cell line used in this study, express intermediate-conductance Ca^{2+} -activated K^+ channels, as well as voltage-dependent K^+ channel α subunits (Kv) 1.1 and Kv1.2. These channels are conservatively expressed in both human and rat odontoblasts, as reported by "Kojima et al., Potassium Currents Activated by Depolarization in Odontoblasts, *Frontiers in Physiology*, 8: 1078, 2017."

Although these studies have been cited in the original manuscript, we have expanded on this information in the Discussion section as follows:

Page 41; Lines 660–667: "In both conventional and gramicidin-perforated whole-cell patch-clamp recordings, macroscopic transmembrane currents were observed using extracellular Na^+ -rich and intracellular K^+ -rich solutions. These currents in human odontoblasts (i.e. the same cell line used in this study) were carried by intermediate-conductance Ca^{2+} -activated K^+ channels, as well as by voltage-dependent K^+ channel α -subunits (Kv) 1.1 and Kv1.2. Notably, these Kv channels were consistently expressed in both human and rat odontoblasts (Ichikawa et al., 2012; Kojima et al., 2017)."

In response to the Reviewer's comment, we have revised the Results section to include a description clarifying the terminology of "outward" and "inward" currents as follows:

Page 32; Lines 515–518: "In the ECS with the KHR solution as a control, the voltage ramp protocol elicited inward (negative deflections, representing cation influx or anion efflux) and outward (positive deflections, representing cation efflux or anion influx) currents in odontoblasts."

8) P30 (470) "The application of 10 mM DMA..." again, this is gigantic concentration, the specificity of its effect is highly doubtful. It would be helpful to know whether the authors equimolarly substituted 10mM of NaCl? If not, the solution was hypertonic and might cause shrinkage of the cells, with consequent nonspecific effects.

Response: Thank you for your comment, and we deeply apologize our oversight. The concentration of DMA used in this study was 20 μM . Accordingly, we have revised the manuscript throughout, as well as Figure 13, by replacing "10 mM DMA" with "20 μM DMA". Based on the previous reports, we believe that the concentration of 20 μM DMA is suitable to inhibit the $\text{Na}^+ - \text{H}^+$ exchanger.

9) P30 (480-481) we applied 6-MSITC to dentin cavities prepared in rats. 6-MSITC significantly increased reactionary dentin formation..." It would be beneficial if the authors provided mean{plus minus}SD or SEM, give n-numbers and the P-values. Also, it

should be specified in which form and concentration 6-MSITC was applied and for how long.

Response: Accordingly, we have revised the relevant sentence, by adding the mean values along with SD values, n as the number of independent experiments tested, and the P -values, as follows:

Legend for Figure 15; “6-MSITC significantly increased reactionary dentin formation to $259.75 \pm 89.593 \mu\text{m}$, compared with the control condition ($91.5 \pm 39.373 \mu\text{m}$, each $n = 4$, $P = 0.0138$) in H&E staining, $255.5 \pm 96.476 \mu\text{m}$, compared with the control condition ($86.0 \pm 44.091 \mu\text{m}$, each $n = 4$, $P = 0.0187$) in Masson’s trichrome staining, and $340.0 \pm 44.099 \mu\text{m}$, compared with the control condition ($18.9 \pm 9.185 \mu\text{m}$, each $n = 3$, $P = 0.000247$) in calcein labelling.”

Details regarding the concentration, carrier compound, and application timing of 6-MSITC are provided in the Materials and Methods section (From Page 22, line 345 to page 24, line 376). In response to the reviewer’s comments, we have now included a brief description of these aspects in the Results section as follows:

Page 34; Lines 540–542; “To evaluate whether 6-MSITC could promote reactionary dentin formation, we applied MedGel containing $500 \mu\text{M}$ 6-MSITC or saline to dentin cavities prepared in rats for 2 weeks.”

10) P35 (551 and below) "Mutations in the SLC4A2, SLC4A4, and SLC4A9 transporters induce enamel anomalies because these proteins play important roles in pH regulation during enamel development (Lyaruu et al., 2008; Lacruz et al., 2010; Jalali et al., 2014; Reibring et al., 2014; Yin & Paine, 2017). Members of the SLC4A family of odontoblasts are likely involved in regulating extracellular pH by mediating the exclusion of HCO_3^- during dentin mineralisation and/or formation." This must be ameloblast story, not odontoblast, it would be good if the authors clarified that in the text.

Response: We have revised the relevant sentences in accordance with the Reviewer’s comments as follows:

From Page 39, line 635 to page 40, line 642; “Mutations in the SLC4A2, SLC4A4, and SLC4A9 transporters induce enamel anomalies because these proteins play important roles in pH regulation during enamel development **by enamel-forming ameloblasts** (Lyaruu et al., 2008; Lacruz et al., 2010; Jalali et al., 2014; Reibring et al., 2014; Yin & Paine, 2017). Members of the SLC4A family of odontoblasts, **which are the dentin-forming cells**, are likely involved in regulating extracellular pH by mediating the exclusion of HCO_3^- during dentin mineralisation and/or formation (see below).”

11) The Latin must be Italic in the text:**P13 (175) In Vivo****P13 (184 respectively). ad libitum****P30 (477) "In vivo"****P33 "... (i.e. CA activation) and its downstream ionic transporters (i.e.)"****P38 (608) - "... (i.e. the current those caused by electrogenic transporters, electroneutral HCO₃⁻ transport (i.e.)"****P40 (641) "dentin-forming/mineralizing front (i.e. to the predentin region).."****P41 (658) "In summary, for the in vitro experiments (Figure 16)..."**

Response: Thank you for your kind note. We have revised all the Latin as Italic form, throughout. In addition, we have revised "N" and "P" with "*n*" and "*P*", respectively, throughout the text and in the Figures.

12) P26 "...and dose-dependent increase in the pH of the medium (Figure 5)" please correct for "concentration-dependent". The same in "DZ, a CA inhibitor, significantly and dose-dependently decreased the 6-MSITC", change for "concentration-dependent".

Response: Accordingly, we have replaced the term "dose-dependent" with "concentration-dependent" throughout the text.

13) P40 (634) "NCX are bidirectional transporters" normally they are called antiporters.

Response: Thank you for your comment. Yes, NCX is an antiporter. Accordingly, we have added the term "antiporter" to the relevant text, as well as in the Introduction section, as follows:

Page 44; Lines 723-724: "Plasma membrane NCX are bidirectional transporters, or antiporters, that catalyse the electrogenic exchange..."

Page 11; Lines 145-147: "Through intracellular Ca²⁺ signalling, Ca²⁺ is extruded to the dentin mineralising front via the Na⁺-Ca²⁺ exchanger (NCX; Na⁺-Ca²⁺ antiporters) subtypes 1 and 3..."

14) Figure 3: Why in Fig 3 P= and in Fig 4 P<? Please make uniform

Response: With regard to the comments #14 to 17 raised by this Reviewer, the "Information for Authors" for the *Journal of Physiology* states "the exact *P* values must be stated to three

significant figures (not decimal places)...” and “The only exception to this is if p is less than 0.001, in which case '<' is permitted.” Therefore, the “P = ” and P < “ have been adjusted accordingly in the Figures. The authors greatly appreciate it if the reviewer understands this point.

15) Figure 5: why P= and P< in the same figure? Please make uniform and it would be good if the authors plot it as a concentration-response to figure out EC50

Response: Please see our response to the comment #14 above raised by this Reviewer, regarding the expression of P-values in the text and figures.

The half-maximal effective concentration (EC₅₀) of 6-MSITC for the changes in medium pH was 29.271 μM (Figure 5), evaluated by fitting the data according to equation 1:

$$A = (A_{max} - A_{min}) / (1 + ([x]_o / K_D)^p) + A_{min} \quad (1)$$

where K_D was the half maximal (50%) effective concentration (EC₅₀) of the 6-MSITC (29.271 μM), [x]_o indicates concentrations of extracellular 6-MSITC applied (0.4 μM to 400 μM), and A_{max} and A_{min} are the maximal and minimal medium pH, respectively. The Hill coefficient (p) was 1.0. Accordingly, we have added the following sentences to the revised manuscript.

Results section (Page 28; Lines 442-447): “Administration of 6-MSITC to the odontoblast culture medium resulted in a significant (at 6-MSITC concentration of 40 μM and 400 μM) and concentration-dependent increase in the pH of the medium (Figure 5), showing a half-maximal effective concentration (EC₅₀) of 29.271 μM in concentrations ranging from 0.4 μM to 400 μM.”

Figure legend for Figure 5 (Page 67; Lines 1108-1114): “The relationship between changes in medium pH and the concentration of extracellular 6-MSITC applied was determined by fitting the data to equation 1 (see legend of Figure 3), where K_D was the half maximal (50%) effective concentration (EC₅₀) of the 6-MSITC (29.271 μM), [x]_o indicates the concentration of extracellular 6-MSITC applied (0.4 μM to 400 μM), and A_{max} and A_{min} are maximal and minimal changes in medium pH, respectively. Hill coefficient (p) was 1.0.”

Discussion section (Page 36; Lines 571-573): “The 6-MSITC concentration of 40 μM, which promotes mineralisation, aligns with the EC₅₀ value (29.271 μM) for the concentration-dependent pH increase induced by 6-MSITC.”

16) Figure 5: why P= and P< in the same figure? Did the authors ever apply DZ

independently of 50 μ M 6-MSITC? Why the authors used here only one concentration (50 μ M) of 6-MSITC?

Response: Please see our response to the comment #14 above raised by this Reviewer, regarding the expression of *P* values in the text and figures.

In addition, regarding the 6-MSITC concentration used in Figure 6, we have already addressed this in comment #2 raised by this Reviewer.

We have data showing effect of DZ independently of 6-MSITC, which is in preparation for publication in a different journal. The results showed that DZ dose-dependently inhibited the mineralisation efficacies. The authors also greatly appreciate it if the reviewer considers this point.

17) Figure 7: in C why $P <$ and $P =$, why is that? Please make uniform. Why not to test 10 μ M KB-R7943, 100 μ M caloxin 1b1, and 1 mM dorzolamide (DZ), on their own without 50 μ M 6-MSITC?

Response: Please see our response to Comment #14 above raised by the Reviewer regarding the expression of *P* values in the text and figures.

Regarding the effect of the plasma membrane Ca^{2+} -ATPase (PMCA) inhibitor caloxin 1b1 on mineralisation under conditions without 6-MSITC, we have previously reported that caloxin 1b1 significantly inhibited mineralisation efficacy. This finding was published in Kimura et al., 2021 (Kimura et al., Plasma membrane Ca^{2+} -ATPase in rat and human odontoblasts mediates dentin mineralization. *Biomolecules* **11**, 1010, 2021) and has already been cited in the original manuscript.

In addition, our preliminary results indicated that both the Na^{+} - Ca^{2+} exchanger (NCX) inhibitor KB-R7943 and the carbonic anhydrase inhibitor dorzolamide (DZ) also inhibit mineralisation under conditions without 6-MSITC. These findings suggest that not only both types of plasma membrane Ca^{2+} extrusion mechanisms, Ca^{2+} -ATPase and the forward (Ca^{2+} extrusion) mode of NCX, but also carbonic anhydrase contribute to physiological dentin mineralisation. These data are in preparation for publication in a different journal. The authors also greatly appreciate it if the reviewer considers this point.

In response to the Reviewer's comment, we have added the following sentences to the manuscript to reflect these findings.

Figure legend for Figure 17 (Page 95; Lines 1373–1381), “Our preliminary results showed that the NCX inhibitor KB-R7943 and the carbonic anhydrase inhibitor dorzolamide both inhibited mineralisation, suggesting that the Ca^{2+} -extrusion activity of NCX and the role of carbonic anhydrase were involved in physiological dentin formation (personal communication,

YS). Together with our present findings, these results suggest that DIDS-sensitive SLC4A transporters (SLC4A1, SLC4A2, SLC4A3, SLC4A8, and SLC4A9), along with Ca²⁺-ATPase, NCX, and carbonic anhydrase activity, also contribute to physiological dentin formation."

18) Figure 8: Why the effects of 6-MSITC and DIDS are so much heterogenous in alizarin red in A and von Kossa staining in B?

Response: Thank you for your comment. We acknowledge that our previous description was insufficient. To improve clarity, we have revised the relevant sentences as follows:

(Page 30; Lines 471–478): "The application of 100 μM DIDS (a solute carrier family 4 [SLC4A] transporter and Cl⁻ channel inhibitor) completely inhibited mineralisation by odontoblasts cultured in mineralisation medium, as shown by both alizarin red staining (Figures 8A and 8C) and von Kossa staining (Figures 8B and 8D). Furthermore, DIDS significantly reduced mineralisation efficiency in odontoblasts even in the presence of 50 μM 6-MSITC in the mineralisation medium, as evidenced by both alizarin red (Figures 8A and 8C) and von Kossa staining (Figures 8B and 8D)."

19) Figure 9: Not quite convincing immunostaining to support the idea on the expression. Did the authors try RNA-seq or q-PCR?

Response: In response to this comment, as well as comment #6 from the Reviewer, we have added new results showing the mRNA expression of SLC4A family members and carbonic anhydrase (CA) using real-time RT-PCR (see additional Figure 16). Please also refer to our response to comment #6.

Response to Reviewer #2

1) This manuscript provides a very detailed mechanistic description for mineralisation of odontoblasts in response to the active ingredients of wasabi sulfinyls (6-MSITS). Based on the data, this seems to involve a functional coupling between carbonic anhydrase (CA), HCO₃--transporters (SLC4A), Na-H exchangers (NHE), plasma membrane Ca ATPase (PMCA)/Na-Ca exchange (NCX) and the TRPA1.

Overall, this is an excellent manuscript that is very impactful for the oral biology/dentistry field, and I believe will be extensively cited. The manuscript is very well written, with nicely presented data that are extensively and appropriately analyzed. All data seem to be internally consistent with the authors hypotheses. However, I have a few minor corrections/suggestions for improvement:

Response:

We would like to sincerely thank the Reviewer for the valuable comments. We greatly appreciate the feedback and apologize for the delay in resubmitting the revised manuscript as additional results were required to address the Reviewers' concerns.

Furthermore, during the revision process, the corresponding author (YS) underwent surgery for stomach cancer, was hospitalized, and is currently undergoing chemotherapy. We deeply apologize once again for the delay in resubmitting the revised manuscript.

2) Could the authors explain whether any increase/decrease in mineralization (Alizarin red or von Kossa stain) observed with any of the drugs might be influenced by effects on cell proliferation (increase) or cell death (decrease). Could these be confounding factors or do the authors have any additional data or could cite previous papers validating this technique for assessing mineralization? Should/could the Alizarin/von Kossa stain be normalized to live cell number?

Response: We have revised the manuscript to include the quantification of cell numbers following 6-MSITC treatment, as assessed via crystal violet staining (see revised Figure 11). Treatment of human odontoblasts with 6-MSITC led to a significant reduction in cell numbers at concentrations of 10 μ M and 50 μ M. Interestingly, the cell numbers increased at 500 μ M, suggesting a recovery at higher concentrations. These findings indicate that 6-MSITC exerts a bimodal effect, acting reciprocally on mineralisation and cell proliferation. At lower concentrations, enhanced mineralisation was associated with reduced cell proliferation, whereas at higher concentrations (100 μ M and 500 μ M), both mineralisation and proliferation were promoted.

We have accordingly added revised Figure 11 and incorporated these findings into the manuscript text.

Materials and Methods section (From page 18, Line 280 to page 19, line 291), **"Crystal violet staining. HOB cells were seeded on 24-well adherent plastic plates (Sumitomo Bakelite Co., Ltd) and cultured in basal medium until they reached confluence. Once confluent, increasing concentrations of 6-MSITC (0 μ M as the control, 10 μ M, 50 μ M, and 500**

µM) were applied. After 72 h of incubation, the cells were fixed with 4% PFA and stained with 0.5% crystal violet (dissolved in methanol, 038-04862, FUJIFILM Wako Pure Chemical Co.) for 20 min at room temperature. After staining, the crystal violet solution was removed, and cells were gently washed three times with distilled water. Images were captured using a microscope (BZ-X710, Keyence, Osaka, Japan). Stained cells were quantified by manual counting from the images and presented as the number of cells per superficial area of 9,846,995 µm², denoted as S (cell number/S)."

Results section (From page 31, Line 492 to page 32, line 504), "**Quantification of cell numbers following 6-MSITC treatment** To monitor cell numbers during the application of 6-MSITC, cell counts were performed using crystal violet staining (Figure 11). Stained cells were quantified by manual counting and presented as the number of cells per superficial area of 9,846,995 µm², denoted as S (cell number/S). Treatment of human odontoblasts with 6-MSITC resulted in a significant decrease in cell numbers at concentrations of 10 µM and 50 µM (4475.33 ± 341.35 cell number/S, *n* = 6, *P* < 0.0001 at 10 µM; 1803.33 ± 211.50 cell number/S, *n* = 6, *P* < 0.0001 at 50 µM; Figures 11B, 11C, and 11E). However, cell numbers recovered at a concentration of 500 µM 6-MSITC (3143.83 ± 175.89 cell number/S *n* = 6, *P* < 0.0001 at 500 µM; Figures 11D and 11E), compared with the control condition (no 6-MSITC application: 6502.33 ± 404.00 cell number/S, *n* = 6; Figures 11A and 11E)."

Figure legend section for Figure 11, "Figure 11. Quantification of cell numbers following 6-MSITC treatment. (A–D) Human odontoblast cells were cultured for 72 h in the presence of 0 µM (control), 10 µM, 50 µM, or 500 µM 6-MSITC after reaching confluence. After crystal violet staining, representative images were captured using a microscope: (A) control (0 µM), (B) 10 µM, (C) 50 µM, and (D) 500 µM 6-MSITC. (E) Quantification of stained cells was performed by manual counting from the images and presented as the number of cells per superficial area of 9,846,995 µm², denoted as S (cell number/S). Box-and-whisker plots show the lower inner limit, 25% percentile, median, 75% percentile, and upper inner limit of cell numbers for each treatment condition. The number of replicates (*n*) is indicated in parentheses. Statistically significant differences between groups are shown with solid lines and corresponding *P*-values."

Discussion section (Page 36; Lines 575–583), "Treatment of human odontoblasts with 6-MSITC resulted in a significant decrease in cell numbers at concentrations of 10 µM and 50 µM, whereas the cell numbers recovered at 500 µM. These results suggest that 6-MSITC exerted a bimodal effect, acting reciprocally on mineralisation and cell proliferation. At lower concentrations, the increased mineralisation efficacy was accompanied by reduced cell proliferation. However, at the higher concentration, both mineralisation and cell proliferation were enhanced (at 100 µM

and 500 μ M, respectively). The mechanism of the bimodal behaviour exerted on mineralisation and proliferation remains unclear, however.”

3) What is the mechanism for the biphasic response of 6-MSITC on mineralization? In the Discussion, the authors suggest that the effect of 6-MSITC-induced mineralization is independent of TRPA1 activation. However, the initial 6-MSITC-induced decrease in mineralization may be due to activation of TRPA1, but the secondary increase may involve some other mechanism. What might that be? Could the authors speculate as to the potential bimodal effect on mineralization (i.e. both the decrease and increase?)

Response: As we have already described in our response to comment #2 raised by this Reviewer (see above), 6-MSITC showed significant inhibition and a significant increase in mineralisation efficiency, depending on its concentration. Treatment of human odontoblasts with 6-MSITC, however, resulted in a significant decrease in cell numbers at concentrations of 10 μ M and 50 μ M, whereas cell numbers recovered at 500 μ M. These results suggest that 6-MSITC exerts a bimodal effect, acting reciprocally on mineralisation and cell proliferation. At lower concentrations, increased mineralisation efficacy was accompanied by reduced cell proliferation. However, at the higher concentration (i.e. at 100 μ M and 500 μ M, respectively), both mineralisation and cell proliferation were enhanced.

Additionally, as the Reviewer has also noted in this comment, AITC, a TRPA1 channel agonist, significantly decreased mineralisation efficiency in odontoblasts. Although the observed 6-MSITC-induced decrease in mineralisation at concentrations up to 20 μ M may have involved a minor contribution from TRPA1 activation, the primary cause was likely a direct effect of 6-MSITC itself. These results also suggest that the 6-MSITC-induced increase in mineralisation occurs independently of TRPA1 channel activation. The authors greatly appreciate it if the reviewer would take these points into consideration.

We have accordingly incorporated this description into the revised manuscript.

In Discussion section (Page 37; Lines 590–596), “Additionally, in the present study, AITC, a TRPA1 channel agonist, significantly decreased the mineralisation efficiency in odontoblasts. Although the 6-MSITC-induced decrease in mineralisation at concentrations up to 20 μ M may involve a minor contribution from TRPA1 activation, the primary cause is likely a direct effect of 6-MSITC itself. These results also suggest that the 6-MSITC-induced increase in mineralisation occurs independently of TRPA1 channel activation.”

4) Related to the above point, did any of the other analogues of 6-MSITC have a similar bimodal effect or were lower concentrations not tested? Could the "lack of potency" of some of these compounds be due to a rightward shift in their bimodal response on mineralization?

Response: We did not observe any such 'bimodal' effects on mineralisation efficiency with Wasabi sulfinyls or related compounds used in this study.

5) Its also worth noting in the Discussion that extracellular acidification will enhance and alkalinization will inhibit PMCA activity because PMCA is in fact an electroneutral ATP-driven H⁺/Ca²⁺-exchanger.

Response: We have added the following description to the Discussion section and included two additional references.

Discussion (Page 46; Lines 752–756), “Note that the PMCA couples Ca²⁺ efflux to H⁺ uptake (Thomas, 2009; 2011). The H⁺ incorporated into the intracellular medium via PMCA activity may subsequently be extruded by the NHE, which further reverses the transmembrane Na⁺ gradient. This reversal can enhance Ca²⁺ influx through NCX activity and promote Ca²⁺ extrusion via the PMCA.”

The following references have been added to the Reference section:

1) Thomas RC (2009). The plasma membrane calcium ATPase (PMCA) of neurones is electroneutral and exchanges 2 H⁺ for each Ca²⁺ or Ba²⁺ ion extruded. *J Physiol* **587**, 315-327.

2) Thomas RC (2011). The Ca²⁺: H⁺ coupling ratio of the plasma membrane calcium ATPase in neurones is little sensitive to changes in external or internal pH. *Cell Calcium* **49**: 357-364.

6) Why is there clustering of cells (or stain) in the center of the wells in Figure 6?

Response: We have replaced the relevant images in Figure 6 in the revised manuscript with clearer images.

7) There is no control in Figure 6. Is this because the mean value for 6-MSITC (approx. 1.5) is very close to the control value (1.3, instead of 2.3 in Fig 4)? If this is the case, then why. Moreover, in this experiments the CA inhibitor (dorzolamide) appears to reduce the values to less than the control values (without 6-MSITC). Why is this?

Response: The overall average values for mineralisation (I/I_0) under the control condition were 1.32 ± 0.28 (n = 76) as measured by alizarin red staining and 1.09 ± 0.069 (n = 70) as measured

by von Kossa staining. In Figure 6, the administration of 6-MSITC to the odontoblast culture medium increased the mineralisation efficiency in both alizarin red and von Kossa staining compared with the condition without 6-MSITC (based on the overall average values for mineralisation). As the Reviewer noted, the carbonic anhydrase (CA) inhibitor, dorzolamide, reduced the mineralisation (I/I_0) values to below those of the control (without 6-MSITC). Our preliminary results indicate that dorzolamide inhibits mineralisation, suggesting that carbonic anhydrase activity plays a role in physiological dentin formation (personal communication, YS).

We have thus added the following description to the legend of Figure 17. We sincerely appreciate the reviewer's understanding of these points.

Figure 17 legend (Page 95; Lines 1373–1381), “Our preliminary results showed that the NCX inhibitor KB-R7943 and the carbonic anhydrase inhibitor dorzolamide both inhibited mineralisation, suggesting that the Ca^{2+} -extrusion activity of NCX and the role of carbonic anhydrase were involved in physiological dentin formation (personal communication, YS). Together with our present findings, these results suggest that DIDS-sensitive SLC4A transporters (SLC4A1, SLC4A2, SLC4A3, SLC4A8, and SLC4A9), along with Ca^{2+} -ATPase, NCX, and carbonic anhydrase activity, also contribute to physiological dentin formation.”

8) MTA not defined on page 11

Response: We have added the definition of MTA, accordingly.

Dear Dr Shibukawa,

Re: JP-RP-2025-287809R1 "6-Methylsulfinylhexyl isothiocyanate activates carbonic anhydrase-dependent HCO₃⁻/H⁺/Na⁺/Ca²⁺ transport via SLC4As-NHE-NCX-PMCA axis in odontoblasts" by Yoshiaki Furusawa, Maki Kimura, Isao Okunishi, Takehito Ouchi, Ryuya Kurashima, Tomoe Katou-Yamada, Hidetaka Kuroda, Makoto Sugita, Masahiro Furusawa, and Yoshiyuki Shibukawa

Thank you for submitting your manuscript to The Journal of Physiology. It has been assessed by a Reviewing Editor and by 2 expert referees and we are pleased to tell you that it is acceptable for publication following satisfactory revision.

REVISION CHECKLIST:

Please upload two versions of your manuscript text: one with all relevant changes highlighted and one clean version with no changes tracked. The manuscript file should include all tables and figure legends, but each figure/graph should be uploaded as separate, high-resolution files. The journal is now integrated with Wiley's Image Checking service. For further details,

see: <https://www.wiley.com/en-us/network/publishing/research-publishing/trending-stories/upholding-image-integrity-wileys-image-screening-service>

We look forward to receiving your revised submission.

Yours sincerely,

Kim Barrett
Senior Editor
The Journal of Physiology

EDITOR COMMENTS

Reviewing Editor:

Thank you for submitting the revised manuscript. Both Referees agree that the manuscript has improved substantially and that all major and minor comments have been satisfactorily addressed.

Please ensure that the remaining suggestion from Referee #2 has been fully implemented. In addition, I kindly ask you to address the following two minor points:

- Please specify the source of the animals (purchased and supplier, or bred in-house)
- Please state explicitly whether postoperative analgesia was provided and briefly describe how animals were monitored during recovery
- Please indicate how data normality was tested

There is no issue regarding the delay in revision. I would also like to extend my best wishes for continued recovery and good health to the Corresponding Author.

REFEREE COMMENTS

Referee #1:

The revised manuscript shows substantial improvement, and the additional work undertaken has strengthened the overall study. The investigation provides valuable insight into the coordinated roles of SLC4A family members, carbonic anhydrase activity, and Ca^{2+} -handling mechanisms during dentin mineralisation. The integration of ionic transport, electrophysiology, and functional mineralisation assays makes this a meaningful and original contribution to the field of odontoblast physiology and dentinogenesis.

The new quantitative details, including $\text{IC}_{50}/\text{EC}_{50}$ values, means {plus minus} SD or SEM, n-numbers, and precise P-values, greatly improve the transparency and interpretability of the data. The clarification of pharmacological concentrations, along with appropriate justification from the literature, enhances the robustness of the experimental design. The addition of RT-qPCR analysis, alongside the revised immunofluorescence figures, provides convincing molecular evidence to support the expression patterns of SLC4A transporters and carbonic anhydrases in human odontoblasts. These data significantly reinforce the mechanistic conclusions.

The electrophysiological descriptions have also benefited from clearer terminology and the inclusion of current density values, aligning the findings with known channel expression profiles in odontoblasts. The in vivo results are now presented with appropriate statistical detail and clearer methodological context, improving confidence in the physiological relevance of the observations.

Overall, the manuscript now presents a coherent and mechanistically insightful study that advances understanding of bicarbonate transport, Ca^{2+} extrusion mechanisms, and their integration in odontoblast-mediated mineralisation. The work is technically solid, conceptually well developed, and of interest to researchers studying ion transport, mineralised tissue biology, and dental physiology.

Referee #2:

I would like to thank the authors for a valiant attempt at addressing all of my concerns with the original manuscript with robust arguments and new data. The additional crystal violet experiments, as a method to assess the effects of 6-MSITC treatment on cell number, helps to reconcile at least in part, the interpretation of the "bimodal" response on mineralization (Alizarin/von Kossa stain). However, in the revised manuscript the authors fail to acknowledge that the observed "apparent" decrease in mineralization (Alizarin/von Kossa stain) with lower concentrations of 6-MSITC are likely caused by a direct reduction in cell number (most likely due to inhibition of cell proliferation, but also potentially cell toxicity or reduced cell adhesion), rather than a decrease in mineralization per se. However, at higher concentrations of 6-MSITC there appears to be a recovery in cell number and corresponding increase in mineralization. In other words, 6-MSITC causes a concentration dependent reduction in cell proliferation (at low concentrations, up to 50 μM), whilst on top of this a concentration-dependent increase in mineralization (at higher concentrations, 40 - 100 μM). An important caveat to this interpretation is that these experiments were performed over a different time course; with the mineralization experiment the cells were cultured for 28 days, whereas for the crystal violet experiments the cells were cultured for 72 hours. Nevertheless, interpretation of the data in this way provides more clarity and is thus less complex or ambiguous. With this in mind, I suggest that the initial mineralization data (Figure 2) and crystal violet data (Figure 11) should be described together, perhaps by combining these data as a single figure or at least presenting these data together as Figure 2 and 3. This then becomes more obvious that the initial decrease in Alizarin/von Kossa stain with lower concentrations is due to reduced cell number (cell proliferation). This does not take anything away from the remaining data and interpretation but instead provides a bit more clarity for the reader.

All other issues have now been adequately addressed

END OF COMMENTS

Response to Reviewers

MS ID#: JP-RP-2024-287809R2

MS Title: 6-Methylsulfinylhexyl isothiocyanate activates carbonic anhydrase-dependent $\text{HCO}_3^-/\text{H}^+/\text{Na}^+/\text{Ca}^{2+}$ transport via SLC4As-NHE-NCX-PMCA axis in odontoblasts

Format: Research Article

Authors: Yoshiaki Furusawa, Maki Kimura, Isao Okunishi, Takehito Ouchi, Ryuya Kurashima, Tomoe Katou-Yamada, Hidetaka Kuroda, Makoto Sugita, Masahiro Furusawa, Yoshiyuki Shibukawa

Response to Reviewing Editor

Thank you for submitting the revised manuscript. Both Referees agree that the manuscript has improved substantially and that all major and minor comments have been satisfactorily addressed.

Please ensure that the remaining suggestion from Referee #2 has been fully implemented. In addition, I kindly ask you to address the following two minor points:

There is no issue regarding the delay in revision. I would also like to extend my best wishes for continued recovery and good health to the Corresponding Author.

We would like to sincerely thank the Editor and the Reviewers for their valuable comments and kind words. The authors greatly appreciate the time and effort of the Editor, the Reviewers, and the editorial staff in supporting the revision process. In response to their suggestions, we have revised the manuscript as follows.

Response:**1. Please specify the source of the animals (purchased and supplier, or bred in-house)**

We have added information regarding the source of the animals to the Ethical Approval section of the Materials and Methods as follows: “The animals used in this study [Wistar rats (Slc:Wistar; n = 7; both sexes; 12–13 weeks old; weight 190–210 g; Sankyo Lab Service Co. INC., Tokyo Japan)] were...”

2. Please state explicitly whether postoperative analgesia was provided and briefly describe how animals were monitored during recovery

We have added the relevant information regarding postoperative treatment to the Ethical Approval section of the Materials and Methods as follows: "For the animal experiments, cavities were prepared in the dentin without pulpal exposure on the occlusal surfaces of mandibular first molars in Wistar rats. The cavities were tightly sealed with a dental adhesive resin cement (see *In vivo* Experiments section) to completely prevent postoperative pain. To prevent abrasion and dislodgement of the restorations due to occlusal contact, the cusps of the maxillary teeth were also drilled. The animals were monitored daily for oral and dental conditions, such as dislodgement of the restorations, and all animals used in this study remained healthy and showed no complications throughout the experimental period."

3. Please indicate how data normality was tested

We have added information regarding the normality test to the Statistics and Offline Analysis section of the Materials and Methods as follows: "Shapiro-Wilk test or Kolmogorov-Smirnov test was used to test for normality."

Response to Reviewer #1

The revised manuscript shows substantial improvement, and the additional work undertaken has strengthened the overall study. The investigation provides valuable insight into the coordinated roles of SLC4A family members, carbonic anhydrase activity, and Ca²⁺-handling mechanisms during dentin mineralisation. The integration of ionic transport, electrophysiology, and functional mineralisation assays makes this a meaningful and original contribution to the field of odontoblast physiology and dentinogenesis.

The new quantitative details, including IC₅₀/EC₅₀ values, means {plus minus} SD or SEM, n-numbers, and precise P-values, greatly improve the transparency and interpretability of the data. The clarification of pharmacological concentrations, along with appropriate justification from the literature, enhances the robustness of the experimental design. The addition of RT-qPCR analysis, alongside the revised immunofluorescence figures, provides convincing molecular evidence to support the expression patterns of SLC4A transporters and carbonic anhydrases in human odontoblasts. These data significantly reinforce the mechanistic conclusions.

The electrophysiological descriptions have also benefited from clearer terminology and the inclusion of current density values, aligning the findings with known channel expression profiles in odontoblasts. The *in vivo* results are now presented with appropriate statistical detail and clearer methodological context, improving confidence in the physiological relevance of the observations.

Overall, the manuscript now presents a coherent and mechanistically insightful study that advances understanding of bicarbonate transport, Ca²⁺ extrusion mechanisms, and their integration in odontoblast-mediated mineralisation. The work is technically solid, conceptually well developed, and of interest to researchers studying ion transport, mineralised tissue biology, and dental physiology.

Response:

We would like to sincerely thank the Editor and the Reviewers for their valuable comments and kind words. The authors greatly appreciate the time and effort of the Editor, the Reviewers, and the editorial staff in supporting the revision process.

Response to Reviewer #2

I would like to thank the authors for a valiant attempt at addressing all of my concerns with the original manuscript with robust arguments and new data. The additional crystal violet experiments, as a method to assess the effects of 6-MSITC treatment on cell number, helps to reconcile at least in part, the interpretation of the "bimodal" response on mineralization (Alizarin/von Kossa stain). However, in the revised manuscript the authors fail to acknowledge that the observed "apparent" decrease in mineralization (Alizarin/von Kossa stain) with lower concentrations of 6-MSITC are likely caused by a direct reduction in cell number (most likely due to inhibition of cell proliferation, but also potentially cell toxicity or reduced cell adhesion), rather than a decrease in mineralization per se. However, at higher concentrations of 6-MSITC there appears to be a recovery in cell number and corresponding increase in mineralization. In other words, 6-MSITC causes a concentration dependent reduction in cell proliferation (at low concentrations, up to 50 μ M), whilst on top of this a concentration-dependent increase in mineralization (at higher concentrations, 40 - 100 μ M). An important caveat to this interpretation is that these experiments were performed over a different time course; with the mineralization experiment the cells were cultured for 28 days, whereas for the crystal violet experiments the cells were cultured for 72 hours. Nevertheless, interpretation of the data in this way provides more clarity and is thus less complex or ambiguous. With this in mind, I suggest that the initial mineralization data (Figure 2) and crystal violet data (Figure 11) should be described together, perhaps by combining these data as a single figure or at least presenting these data together as Figure 2 and 3. This then becomes more obvious that the initial decrease in Alizarin/von Kossa stain with lower concentrations is due to reduced cell number (cell proliferation). This does not take anything away from the remaining data and interpretation but instead provides a bit more clarity for the reader.

Response:

1. However, in the revised manuscript the authors fail to acknowledge that the observed "apparent" decrease in mineralization (Alizarin/von Kossa stain) with lower concentrations of 6-MSITC are likely caused by a direct reduction in cell number (most likely due to inhibition of cell proliferation, but also potentially cell toxicity or reduced cell adhesion), rather than a decrease in mineralization per se. However, at higher concentrations of 6-MSITC there appears to be a recovery in cell number and corresponding increase in mineralization.

We sincerely thank the Reviewer for the valuable comments and fully agree with the points raised. In our preliminary data, treatment of human odontoblasts with 6-MSITC decreased the expression of cell cycle-dependent mRNAs involved in proliferation-related signaling pathways, while increasing the expression of mineralization-related signals (personal communication from T.O. and Y.S.). We are currently conducting experiments to elucidate the detailed regulatory mechanisms of both proliferation- and mineralization-related signaling pathways, as well as their potential interactions. These data will be submitted separately in the near future and, therefore, have not been included in the

revised manuscript. We greatly appreciate the Reviewer's understanding.

Accordingly, we have revised the wording of the relevant sentences in the Discussion section as follows: "At lower concentrations of 6-MSITC (10 and 20 μ M), the observed decrease in mineralization was likely attributable to a reduction in cell number, resulting from decreased cell proliferation rather than a direct reduction in mineralization. At higher concentrations, both cell number and mineralization capacity recovered, with mineralization and cell proliferation specifically enhanced at 100 μ M and 500 μ M, respectively."

2. With this in mind, I suggest that the initial mineralization data (Figure 2) and crystal violet data (Figure 11) should be described together, perhaps by combining these data as a single figure or at least presenting these data together as Figure 2 and 3. This then becomes more obvious that the initial decrease in Alizarin/von Kossa stain with lower concentrations is due to reduced cell number (cell proliferation). This does not take anything away from the remaining data and interpretation but instead provides a bit more clarity for the reader.

In accordance with the Reviewer's comment, we have moved the crystal violet data (originally Figure 11) to Figures 2E1–F, presenting the initial mineralization data, along with the corresponding figure legends. Figure numbers have been revised accordingly throughout the manuscript.

Dear Professor Shibukawa,

Re: JP-RP-2026-287809R2 "6-Methylsulfinylhexyl isothiocyanate activates carbonic anhydrase-dependent HCO₃⁻/H⁺/Na⁺/Ca²⁺ transport via SLC4As-NHE-NCX-PMCA axis in odontoblasts" by Yoshiaki Furusawa, Maki Kimura, Isao Okunishi, Takehito Ouchi, Ryuya Kurashima, Tomoe Katou-Yamada, Hidetaka Kuroda, Makoto Sugita, Masahiro Furusawa, and Yoshiyuki Shibukawa

We are pleased to tell you that your paper has been accepted for publication in The Journal of Physiology.

Yours sincerely,

Kim Barrett
Senior Editor
The Journal of Physiology

IMPORTANT POINTS TO NOTE FOLLOWING ACCEPTANCE OF YOUR PAPER:

- **IMPORTANT NOTICE ABOUT OPEN ACCESS:** To assist authors whose funding agencies mandate immediate public access to published research findings, The Journal of Physiology allows authors to pay an Open Access (OA) fee to have their papers made freely available immediately on publication.

- You can help your research get the attention it deserves! Check out Wiley's free Promotion Guide for best-practice recommendations for promoting your work at: www.wileyauthors.com/eeo/guide. You can learn more about Wiley Editing Services which offers professional video, design, and writing services to create shareable video abstracts, infographics, conference posters, lay summaries, and research news stories for your research at: www.wileyauthors.com/eeo/promotion.

- If you would like to receive our 'Research Roundup', a monthly newsletter highlighting the cutting-edge research published in The Physiological Society's family of journals (The Journal of Physiology, Experimental Physiology, Physiological Reports, The Journal of Nutritional Physiology and The Journal of Precision Medicine: Health and Disease), please click this link, fill in your name and email address and select 'Research Roundup': <https://www.physoc.org/journals-and-media/membernews>

EDITOR COMMENTS

Reviewing Editor:

Thank you for the additional revisions and for addressing the remaining comments.